# Ultrafast THz probing of nonlocal orbital current in transverse multilayer metallic heterostructures

Sandeep Kumar [1] & Sunil Kumar [1]✉

THz generation from femtosecond photoexcited spintronic heterostructures has become a versatile tool for investigating ultrafast spin-transport and transient charge-current in a non-contact and non-invasive manner. The equivalent effect from the orbital degree of freedom is still in the primitive stage. Here, we experimentally demonstrate orbital-to-charge current conversion in metallic heterostructures, consisting of a ferromagnetic layer adjacent to either a light or a heavy metal layer, through detection of the emitted THz pulses. Our temperature-dependent experiments help to disentangle the orbital and spin components that are manifested in the respective Hall-conductivities, contributing to THz emission. NiFe/Nb shows the strongest inverse orbital Hall effect with an experimentally extracted value of effective intrinsic Hall-conductivity, $(\sigma_{SOH}^{\text{int}})^{eff} \sim 195\,\Omega^{-1}cm^{-1}$, while CoFeB/Pt shows maximum contribution from the inverse spin Hall effect. In addition, we observe a nearly ten-fold enhancement in the THz emission due to pronounced orbital-transport in W-insertion heavy metal layer in CoFeB/W/Ta heterostructure as compared to CoFeB/Ta bilayer counterpart.

Efficient generation and detection of spin current are the key requirements in spintronic devices for their different potential applications[1,2]. These include, the spin-orbit torque (SOT), spin-pumping, magnetic memories, excitation of magnons, manipulating the magnetic damping, etc. Mainly, the spin Hall effect[3,4] (SHE) and Rashba-Edelstein effect[5,6] (REE), governed by the spin angular momentum (S) transfer, have been invoked commonly in the generation of a spin current from a charge current. It has become clear from a few recent studies that in certain solids[7–9], the transport of electron's orbital angular momentum (L) and hence the associated magnetic moment, is also responsible for various interesting phenomena in the emerging field of orbitronics. Orbital Hall effect (OHE), which was conceived[10,11] just after the SHE[3], has often been neglected due to orbital quenching in the periodic solids[12,13]. Bearing many similarities with the SHE, in the OHE, a transverse flow of orbital angular momentum occurs in response to a longitudinally applied electric field. In fact, fundamentally, SHE has been proposed to originate from OHE only[14–16]. Therefore, it has helped in resolving the sign and magnitude of the reported values of spin Hall conductivities for certain materials. A few theoretical and experimental studies in the recent literature[13,14,17] have indicated a gigantic OHE in the light as well as several heavy metals and therefore, it has necessitated more careful investigations of SHE-based phenomena and devising new schemes to disentangle the OHE.

As indicated above, OHE is a fundamental phenomenon, which can be observed in a variety of materials, including transition metals[13–15,17–20], semiconductors[11], two-dimensional materials[21–23], etc[24]. Unlike the SHE, whose strength greatly depends on the spin-orbit coupling (SOC) in the material, OHE, on the other hand, can be found even in the light materials, having very weak SOC[13,20]. After a few theoretical reports[13,14,17] on the large orbital Hall conductivity ($\sigma_{OH}$) in some strong as well as weak SOC-type materials, detailed experimental demonstrations are required to obtain further insights for harnessing the same in practical applications. For instance, a high value of the Hall

[1]Femtosecond Spectroscopy and Nonlinear Photonics Laboratory, Department of Physics, Indian Institute of Technology Delhi, New Delhi, India.
✉e-mail: kumarsunil@physics.iitd.ac.in

conductivity is always advantageous as it helps in the enhancement of spin-orbit torque (SOT), which is technologically relevant to memory applications. Conventionally, in SHE induced torque (SHT), the spin angular momentum transfer exerts a torque directly on the local magnetization of the material. However, due to the lack of direct exchange coupling[7] of orbital angular momentum with the local magnetization, a similar direct realization of the OHE induced torque or the orbital Hall torque (OHT) was lacking. This pertinent issue has found a regenerated interest among researchers[7,8,17,19] to make OHT based applications viable[9,25] by devising novel orbital-spin (L-S) conversion schemes and suitable material combinations[19,26]. It follows from the magnetization manipulation through the exerted net torque, dominated by either SHT or OHT, and acts as a key in distinguishing the orbital character indirectly from the pure spin transport, yet not in an unambiguous manner. Choi et al.[16], have used magneto-optical Kerr effect (MOKE) in addition to the orbital torque measurement technique for direct detection of orbital magnetic moment accumulation created by the charge current flow in a light metal, Ti. A non-contact method is always promising to non-invasively measure the spin and orbital transport in materials.

Like the toque method to detect the charge to spin or orbital conversion, the inverse of the SHE (ISHE) and REE (IREE) are routinely used for the detection of spin transport through the spin-charge conversion, where the spin source can be one from either spin pumping or spin Seebeck current or optical excitation, etc. The Onsager reciprocity[27] allows an interconversion between the orbital and charge currents, where, similar to the ISHE and IREE for the spin counterpart, here, inverse orbital Hall effect (IOHE) and inverse orbital Rashba-Edelstein effect[25] (IOREE) are both in play. In several studies[28–37] in the last decade or so, ISHE and IREE have been utilized in the THz electromagnetic pulse generation from ultrafast photoexcited magnetic/nonmagnetic (NM) multilayer systems. Consequently, the scheme, in conjunction with the multilayers, is not only recognized as a source of effective THz radiation[38], but also a highly sensitive contactless optical probing tool[39–46] for the detection and control of the ultrafast processes at femtosecond time scale, spin-charge conversion mechanisms, demagnetization dynamics and transport, interfacial properties, etc. For the case of the ISHE based spintronic THz emitters, spin current from the ferromagnetic (FM) or antiferromagnetic layer is injected into the NM layer, where it gets converted to charge current. Therefore, heavy metal layer with large SOC is desirous for efficient THz generation from a bilayer[47]. Similar effects are envisaged to exist for the orbital counterpart too[48,49]. For the THz emission utilizing the orbital transport properties, i.e., orbital-charge conversion through IOHE, material candidates capable of generating an orbital current ($J_L$) and subsequently converting the same into charge current, are desired.

In the current work, existence of nonlocal orbital transport is experimentally detected through IOHE mediated efficient THz emission from femtosecond NIR (near-infrared) pulse excited bi- and tri-layer metallic heterostructures using temperature-dependent THz time-domain spectroscopy. Since the orbital degree of angular momentum is strongly correlated with the crystal field potential, therefore, the temperature-dependency of the phonon scattering would severely affect the OHE and the related phenomena, microscopically. The specifically chosen heterostructures, in this work, consist of FM and NM material combinations, where the choice of the FM is from either CoFeB or NiFe, whereas the NM is from both the light metal (Nb) as well as the heavy metals (Pt, Ta, and W). While the THz emission from CoFeB/Pt, CoFeB/Ta, CoFeB/W, and Fe/Ta bilayers is shown to originate principally from the ISHE in the heavy metal layers therein, the same from NiFe/Nb arises primarily via the IOHE in the light metal layer of Nb. In the prior, strong ultrafast photoinduced spin current is generated in the FM layer to be further injected into the NM heavy metal layer whereas in the latter case, efficient spin-orbital

conversion within NiFe layer facilitates strong ultrafast orbital current injection into the Nb layer. The temperature-dependence of the THz amplitude vis-a-vis the Hall conductivities are used to distinguish spin-to-charge and orbital-to-charge signatures in the NM layers. The wide-range temperature-dependent THz results and analysis also help to distinguish dominating extrinsic and intrinsic contributions to IOHE in different resistivity regions. For the observation of IOHE mediated THz emission from structures consisting of heavy metal layer, we fabricated a tri-layer system of CoFeB/W/Ta and measured the temperature-dependent THz amplitude and the Hall conductivities. The THz emission from such a tri-layer, having the W-insertion layer, interfaced with another heavy metal layer of same sign of the spin Hall angle and placed side-by-side, is nearly one-order stronger than the CoFeB/Ta bilayer counterpart. Such an enhancement in the THz emission is associated with efficient spin to orbital conversion due to strong SOC and long diffusion length of the orbital current in the W-insertion layer.

## Results and discussion
### NiFe/Nb: Probing inverse orbital Hall effect in light metal through generation of THz pulses

Figure. 1a schematically illustrates the emission of THz pulses from the NiFe/Nb bilayer following optical excitation by linearly polarized femtosecond NIR pulses whereas, the same for bare NiFe sample is presented in Supplementary Section S12. The thicknesses of the layers were kept at 5 nm for NiFe and 10 nm for Nb layer. The full THz bandwidth of the signals from NiFe/Nb sample as obtained by fast Fourier transform is shown in Supplementary Section S6. A constant external magnetic field (**B**), having a value just above the saturation (~200 Oe), is applied along the y-direction. A few consistency checks were recorded, to initially validate the origin for the generation of THz pulses, for four geometries of the direction of the optical excitation and the magnetic field, as presented in Fig. 1b, c, respectively. NiFe is a popular FM material for spintronic applications. Dominant presence (≥90%) of Ni (light element) in NiFe makes it possess a large and positive value of spin-orbit correlation factor[8], $\langle L.S \rangle = \eta > 0$. As shown in Fig. 1d, in positive spin-orbit correlation materials, transverse orbital and spin Hall effects are induced in response to the flow of a longitudinal charge current, $J_C$ such that the polarization direction of the accumulated orbital and spin magnetic moments is the same[8]. An optimal material composition and growth of NiFe enriched with Ni can provide a better value of $\eta$ as compared to that in Ni[50]. Moreover, a transient change in the spin-orbit coupling in Ni, triggered by the high energy ultrafast excitation, has been seen to enhance the $\eta$ value significantly[51].

Following the optical pulse excitation of NiFe/Nb (Fig. 1a), ultrafast demagnetization[52] in the NiFe layer stimulates flow of a spin current with density $J_S$. Due to a large positive value of $\eta$ in the NiFe, a fraction of the ultrafast spin current is converted into an ultrafast orbital current ($J_L$) of same polarity through the L-S conversion, given[8,48] as, $J_L = \eta_{L-S} \cdot J_S$. Therefore, an ultrafast optically induced orbital current sets in[48,53], which possess similar symmetry properties to the spin current but can exhibit relatively different transport dynamics[9,54,55]. Furthermore, as the ultrafast excitation of spin and orbital magnetization has been reported[51,56–58] to exhibit a similar evolution, the emergence of orbital current can also be comprehended through the analogy with the already established spin current formation[53]. Consequently, as indicated in Fig. 1a, both the spin and orbital currents through their interconversion within the NiFe layer are now launched into the adjacent Nb layer. A very weak negative SOC strength and about an order larger orbital Hall conductivity than the spin Hall conductivity ($\sigma_{OH}$) in Nb make it a suitable candidate[59,60] for realizing orbital transport phenomena. In case of heavy metals, although the value of $\sigma_{OH}$ is typically much larger than the $\sigma_{SH}$[14], but, OHE is greatly suppressed by the inherently present SOC owing to the

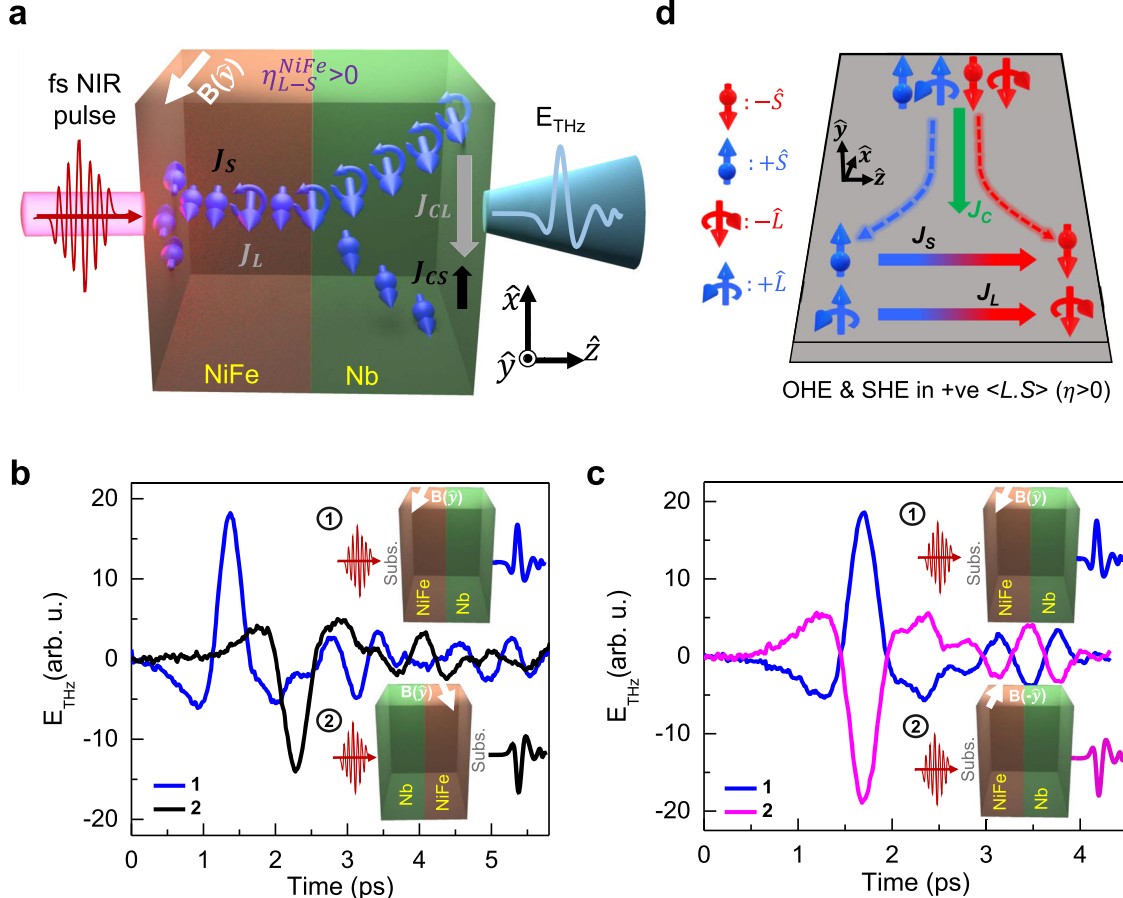

**Fig. 1 | Orbital-to-charge current conversion mediated THz emission from NiFe/Nb bilayer. a** Schematic illustration of ultrafast optically induced spin current $J_S$ and orbital current $J_L$ in the NiFe and their injection into the Nb layer producing ultrafast charge current and hence emission of THz pulses. The positive value of $\eta_{NiFe}$ represents the spin-to-orbital current conversion efficiency in NiFe. $J_{CS}$ and $J_{CL}$ represent charge currents converted from $J_S$ and $J_L$ through the ISHE and IOHE, respectively. $B$ represents the externally applied magnetic field. Temporal profiles of THz signal obtained from NiFe/Nb under four experimental geometries with the identical macroscopic geometries between OHE and SHE in response direction of the optical excitation and the magnetic field: **b** 1. substrate side excitation; 2. Nb film side excitation, while keeping **B** fixed along $\hat{y}$-direction, **c** 1. $\mathbf{B}(+\hat{y})$; 2. $\mathbf{B}(-\hat{y})$, while keeping the optical excitation from the substrate side. **d** Schematic of OHE and SHE in a prototype material. A charge current, $J_C$ in the $-\hat{x}$-direction induces accumulation of spin ($S$) and orbital ($L$) angular momenta in the transverse $\hat{z}$-direction. For positive spin-orbit correlation, $\eta > 0$, both the spin and orbital polarizations are parallel along either $+\hat{y}$-direction or $-\hat{y}$-direction.

identical macroscopic geometries between OHE and SHE in response to the applied electrical current in steady operations[20]. Therefore, to overcome the effect of spin transport in the realization of OHE, selection of an appropriate light element material is regarded as one of the key solutions.

The spin and orbital currents injected into the Nb layer convert to respective in-plane transient charge currents. Charge current, $J_{CS}$ is produced via ISHE and $J_{CL}$ is produced via IOHE. The magnitude and polarity of the emitted THz radiation are finally dependent on the net transient charge current in the Nb layer. Typically, the orbital diffusion length in Nb is comparable to that of spin diffusion length (see Supplementary Section S13). Due to the negligible SOC and correspondingly smaller spin Hall angle in Nb, the ISHE signal is nearly quenched (depicted by small black vector within the Nb layer in Fig. 1a) as against the IOHE. The net charge current in the Nb layer is the vectorial sum of the two charge currents, i.e., $J_C = J_{C-ISHE} + J_{C-IOHE}$, and it can be further expressed as

$$J_C = \theta_{SH} \cdot J_S + \theta_{OH} \cdot J_L = \theta_{SH}^{Nb} \cdot J_S + \theta_{OH}^{Nb} \cdot \eta_{L-S}^{NiFe} \cdot J_S \qquad (1)$$

Here, $\theta_{SH}$ and $\theta_{OH}$ are the spin and orbital Hall angles of Nb, which are negative and positive, respectively[15,18]. Since the spin-orbit correlation factor, $\eta_{L-S}^{NiFe}$ is positive for the NiFe, therefore, the charge current contribution from the individual components would have opposite signs, as represented in Fig. 1a. The net charge current is, therefore, the difference of the two, and the dominant one among them would control the amplitude and polarity of the emitted THz radiation. If the THz signal from the NiFe/Nb bilayer structure is a result of the inverse spin Hall effect (ISHE), its polarity would be expected to be the same as that of the Fe/Ta bilayer but opposite to that of the CoFeB/Pt bilayer structures. This distinction arises because Nb and Ta both have the negative sign of $\theta_{SH}^{Ta}$, while Pt has a positive sign of $\theta_{SH}^{Pt}$. However, the observed polarity of the THz signals in the NiFe/Nb bilayer structure is opposite to that of Fe/Ta bilayer structure but coincides with the CoFeB/Pt bilayer structure, as schematically depicted in Figs. 1a, 3a, b, respectively. In fact, the polarity of the THz signals emitted from NiFe/Nb bilayer structure aligns with the sign of $\theta_{OH}^{Nb} \cdot \eta_{L-S}^{NiFe}$ implying that the resultant THz emission from NiFe/Nb takes place via IOHE, mainly. A few consistency checks are carried out in Fig. 1b, c to reveal the magnetic origin of the THz signal. The THz signal polarity is inverted by reversing either the direction of the optical excitation while keeping the magnetic field unchanged or the direction of the external magnetic field while keeping the direction of optical excitation unchanged. In the first case, the THz sign reversal is due to the change in the direction of the spin current and consequently the flow of orbital current via spin to orbit conversion in NiFe, whereas, in the second case, it is due to the change in the spin current polarization and

## a

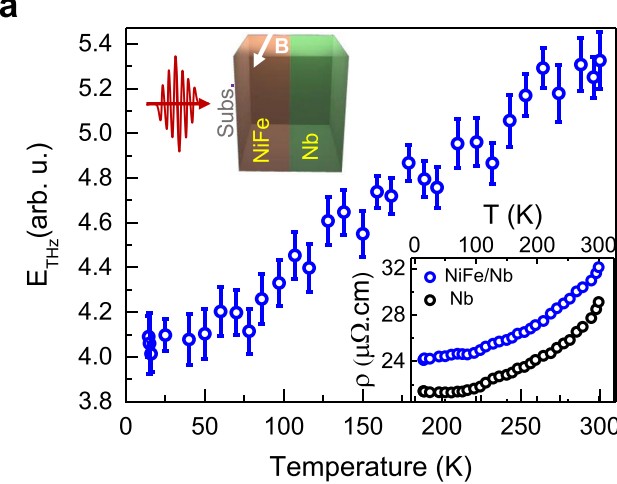

## b

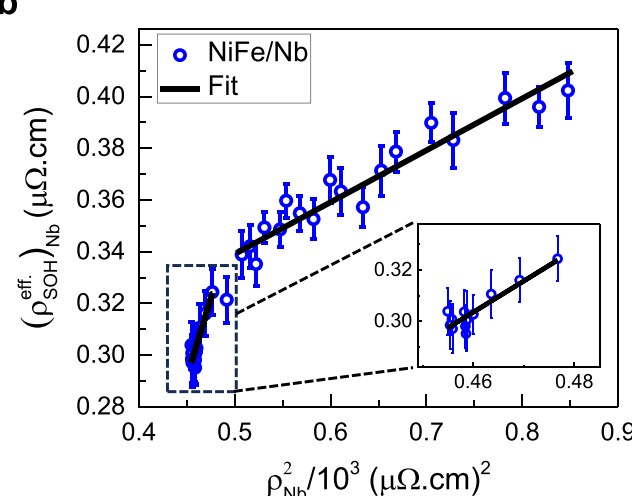

**Fig. 2 | Temperature-dependent THz emission from NiFe/Nb (Nb = light metal).** **a** Peak-to-peak value of THz amplitude as a function of varying sample temperature. The error bars at each temperature correspond to the largest absolute deviation of the peak-to-peak THz amplitude from the mean of three measurements. Inset: The optical excitation from the substrate side; temperature-dependent resistivity ($\rho$) variations of NiFe/Nb and Nb samples measured by the four-point van der Pauw method. **b** Effective spin-orbital Hall resistivity as a function of the squared longitudinal resistivity for the Nb film. Inset: Zoomed-in view of the data below $\rho_{Nb}^2 \sim 0.5 \times 10^3 \ (\mu\Omega \cdot cm)^2$. The solid lines are fit to the data using Eq. (4).

subsequently the polarization of orbital current via spin-orbit conversion as both are constrained by positive $\eta_{L-S}^{NiFe}$.

Temperature-dependent studies[61–63] have been proven to be important in identifying various scattering mechanisms in the materials and subsequent determination of respective Hall conductivities. To validate the IOHE origin of the THz emission from NiFe/Nb bilayer, we have performed temperature-dependent experiments by monitoring the changes in the peak-to-peak amplitude of the THz signal as a function of the sample temperature (T) from 10 K to 300 K. The temperature sensitivity of the phonon scattering would have a significant impact on the OHE and associated phenomena because the orbital degree of angular momentum is strongly coupled with the crystal field potential[13,20]. The result is presented in Fig. 2a, where the error bars represent the largest absolute deviation from the mean of three THz signal outputs at each temperature. As indicated in the inset of the figure, the sample is optically excited from the substrate side and the same orientation for sample excitation is followed in all the results presented below in the paper. The substrate side optical

excitation geometry does not create any temperature dependent contributions (see Supplementary Section S15). Since, the sign and magnitude of the emitted THz pulse is directly related with the $\sigma_{SH}$ or $\sigma_{OH}$, hence, by looking at the THz signal's amplitude and phase, a qualitative information about the $\sigma_{SH}$ or $\sigma_{OH}$ can be obtained[32]. For exactly determining the dominating role of either the $\sigma_{SH}$ or $\sigma_{OH}$, detailed analysis is presented below.

By employing the four-point van der Pauw method, electrical longitudinal resistivity ($\rho$) was also determined for the Nb and NiFe/Nb films in the entire experimental temperature range (see inset of Fig. 2a). The resistivity information, together with the THz amplitude data in Fig. 2a, are used to obtain the results presented in Fig. 2b, where we have plotted the behavior of extracted effective spin-orbital Hall resistivity ($\rho_{SOH}^{eff}$) with respect to the squared longitudinal resistivity ($\rho_{NM}^2$) of NM = Nb. See Supplementary Section S10 for details to obtain $\rho_{SOH}^{eff}$. In the light of a temperature scaling relation[61–64] the spin Hall resistivity ($\rho_{SH}$) is expressed as,

$$\rho_{SH}(T) = \sigma_{SH}^{int} \cdot \rho_{NM}^2(T) + \sigma_{SJ} \cdot \rho_{0,NM}^2 + \alpha_{ss} \cdot \rho_{0,NM} \quad (2)$$

Here, $\sigma_{SH}^{int}$, $\sigma_{SJ}$, $\rho_{0,NM}$, $\alpha_{ss}$ are intrinsic spin Hall conductivity, side-jump spin Hall conductivity, residual resistivity, and skew scattering angle of the NM layer, respectively. The second and the third terms on the right-hand side of Eq. (2) represent the extrinsic contributions to scattering. Because of the correspondence between the SHE and OHE, a similar phenomenological equation for the orbital Hall resistivity ($\rho_{OH}$) can be constructed by accounting for the intrinsic and extrinsic orbital scattering processes[8,15,20]. Hence, temperature-dependence of $\rho_{OH}$ can be expressed as,

$$\rho_{OH}(T) = \eta_{L-S} \cdot \sigma_{OH}^{int} \cdot \rho_{NM}^2(T) + Ex \quad (3)$$

On the right side of the Eq. 3, the first term, consisting of the intrinsic orbital Hall conductivity, $\sigma_{OH}^{int}$, represents the intrinsic scattering, while the extrinsic contributions to orbital Hall resistivity are represented by the second term, $Ex$. The factor, $\eta_{L-S}$, takes care of the spin-orbit interconversion arising due to SOC. The extrinsic contribution to the spin or the orbital Hall resistivity in the above equations, is usually neglected for pure materials[63,65] while, it can be significantly high for materials with high impurity concentration. For capturing the temperature-dependence of the effective spin-orbital Hall resistivity, Eqs. (2) and (3) can be combined and $\rho_{SOH}^{eff}$ is expressed at each temperature $T$ as following,

$$\rho_{SOH}^{eff}(T) = (\sigma_{SH}^{int} + \eta_{L-S} \cdot \sigma_{OH}^{int}) \cdot \rho_{NM}^2(T) = (\sigma_{SOH}^{int})_{NM}^{eff} \cdot \rho_{NM}^2(T) \quad (4)$$

In the above, we have used $(\sigma_{SOH}^{int})^{eff} = (\sigma_{SH}^{int} + \eta_{L-S} \cdot \sigma_{OH}^{int})$ to represent the effective intrinsic spin-orbital Hall conductivity and any weak temperature variations in the extrinsic terms in both the spin and the orbital Hall resistivities have been ignored. As shown in Section S10 of the Supplementary Information, the effective spin-orbital Hall resistivity at given temperature $T$ is related to the amplitude of the THz signal[61,64], and it is given by the relation,

$$\rho_{SOH}^{eff}(T) \sim E_{THz}(T) \left( \frac{\rho_{xx}^{NM}}{\rho_{FM/NM}} \right) \left( \frac{d}{\lambda_{LS}} \right) \frac{1}{e \cdot J_S} \quad (5)$$

where, $\lambda_{LS}$ is the effective spin-orbit diffusion length[17] and $J_S$ still represents the maximum instantaneous spin current produced in the FM layer by the ultrafast optical excitation. The extracted data for $\rho_{SOH}^{eff}$ vs $\rho_{NM}^2$ at each temperature, are presented in Fig. 2b. From the behavior of the data, two distinct resistivity regions, above and below $\rho_{Nb}^2 \sim 0.5 \times 10^3 \ (\mu\Omega \cdot cm)^2$, are evident. Such an observation with respect to the squared longitudinal resistivity has been found helpful

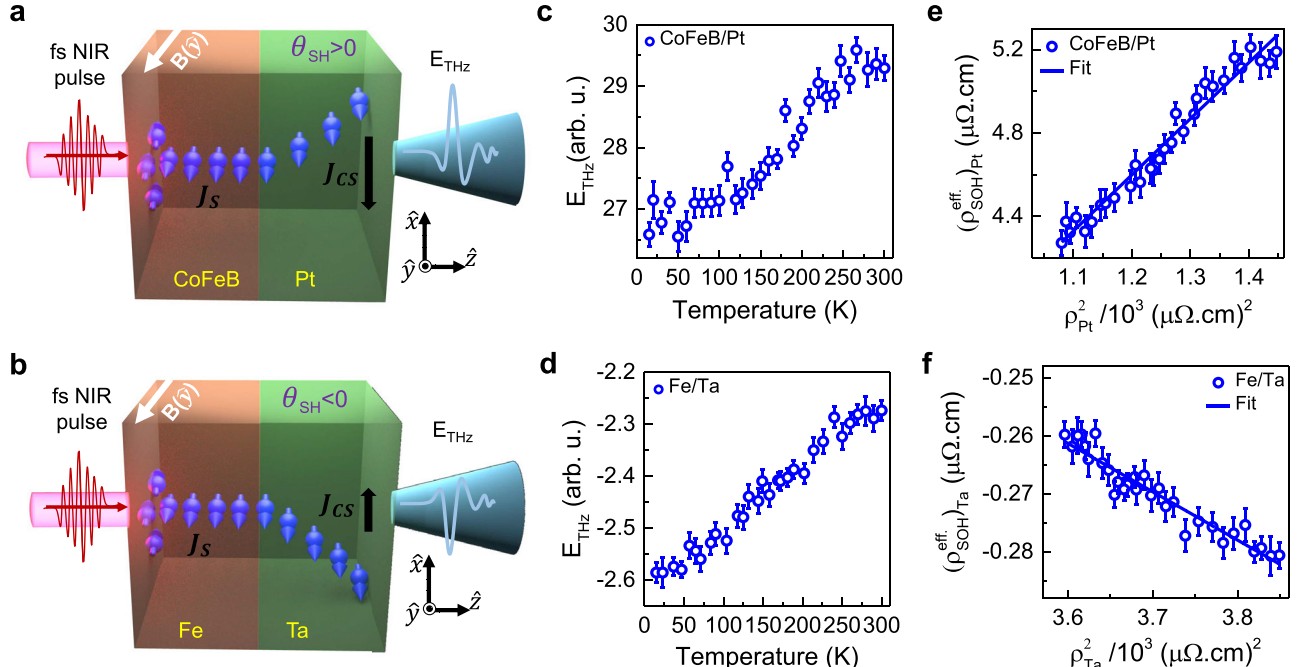

**Fig. 3 | Temperature-dependence of THz signal and effective Hall resistivity for CoFeB/Pt and Fe/Ta bilayers (Pt, Ta = heavy metal).** Schematic illustration of the ultrafast optical excitation, spin magnetic moment transport and its conversion to transient charge current through ISHE to emit a THz pulse from **a** CoFeB/Pt and **b** Fe/Ta. The spin Hall angle (θ$_{SH}$) and the direction of the external magnetic field are indicated. Temperature-dependent variation of the THz signal peak-to-peak amplitude for **c** CoFeB/Pt, and **d** Fe/Ta. The error bars at each temperature correspond to the largest absolute deviation of the peak-to-peak THz amplitude from the mean of three measurements. Effective spin-orbital Hall resistivity, $\rho_{SOH}^{eff.}$ as a function of the squared longitudinal resistivity, $\rho_{NM}^2$ for **e** Pt and **f** Ta. The continuous curves in **e**, **f** are linear fits to the data using Eq. (4).

in probing intrinsic and extrinsic contribution dominated resistivity regions for AHE[66-69] and SHE[63]. However, the same for the case of OHE, though anticipated[17], but has not been reported experimentally so far. According to Eqs. (4) and (5), linear fits to the data in Fig. 2b yield $(\sigma_{SOH}^{int})_{Nb}^{eff} \sim +195(\hbar/e)\,\Omega^{-1}cm^{-1}$ in the low-resistivity region and $(\sigma_{SOH}^{int})_{Nb}^{eff} \sim +1210(\hbar/e)\,\Omega^{-1}cm^{-1}$ in the high resistivity region. From the theoretically known values of different intrinsic parameters for NiFe/Nb sample, i.e., $\sigma_{SH}^{int}((\hbar/e)\Omega^{-1}cm^{-1}) \cong -100^{14,18}$, $\eta_{L-S} \cong 0.045^8$, and $\sigma_{OH}^{int}((\hbar/e)\Omega^{-1}cm^{-1}) \cong 6000^{18}$, we obtain, $(\sigma_{SOH}^{int})_{Nb}^{eff} = (\sigma_{SH}^{int} + \eta_{L-S}.\sigma_{OH}^{int}) \sim +170(\hbar/e)\Omega^{-1}cm^{-1}$, a value matching with our experimental value. The excellent agreement between the two values clearly indicates dominance of orbital transport with majorly intrinsic contribution to OHE in the high resistivity region of the NiFe layer. Therefore, we conclude from here that dominating orbital current in the NiFe layer governs the THz emission from NiFe/Nb via intrinsic IOHE in the light metal Nb layer. The much higher value of $(\sigma_{SOH}^{int})_{Nb}^{eff} \sim +1210(\hbar/e)\,\Omega^{-1}cm^{-1}$ in the low resistivity region must originate from extrinsic reasons, analogous to what is known for AHE[66-69] and SHE[63]. With the information at hand that for Nb, $\sigma_{SH}^{int}$ is negative and $\eta_{L-S}$ is positive, a positive value of $(\sigma_{SOH}^{int})_{Nb}^{eff}$ from the fit clearly reveals that $\eta_{L-S}.\sigma_{OH}^{int}$ for Nb is a large positive value and dominates over $\sigma_{SH}^{int}$. In fact, theoretical predictions of a gigantic +ve value of $\sigma_{OH}$ over a small -ve value of $\sigma_{SH}$ in Nb[70], are well aligned with our experimental observations.

## FM(CoFeB,Fe)/NM(Pt,Ta): ISHE mediated THz emission from heavy metal layers based FM/NM heterostructures

Experimental results are now presented and discussed on Fe/Ta and CoFeB/Pt, in conjunction with those on CoFeB/Ta, presented elsewhere[61], and re-examined again as presented in Supplementary

Section S8. These bilayer heterostructures contain a heavy metal layer, either Pt or Ta, and are chosen selectively for their opposite sign of the spin Hall angle[32,34]. Moreover, CoFeB and Fe are selected because of their negligible[8,48] spin-orbit correlation factor, i.e., $\eta_{L-S}^{CoFeB}$ and $\eta_{L-S}^{Fe} \sim 0$ as compared to Ni or NiFe presented earlier. By these choices, we initially ensure that there is negligible fractional conversion of the spin current into orbital current within the FM layer. Consequently, from the femtosecond pulse excited FM/NM bilayers, THz generation takes place via ISHE only. For the CoFeB/Pt with the Pt layer having positive spin Hall angle ($\theta_{SH}^{Pt} > 0$), the experimental configuration is depicted in Fig. 3a, where the directions of the external magnetic field, spin current, charge currents and the THz signal polarities are indicated. Owing to the opposite sign of the spin Hall angle in Ta ($\theta_{SH}^{Ta} < 0$), emission of opposite polarity THz signal from Fe/Ta, under the same experimental configuration, is shown in Fig. 3b. In both cases, the THz signal polarity dependence on the experimental configuration, including the direction of the external magnetic field and the optical excitation, are found to be as expected from the ISHE mediated THz emission[29,32]. Moreover, ultrafast demagnetization mechanism is mainly responsible for the THz emission from the bare FM layer (see Supplementary Section S12). Variations of the peak-to-peak THz signal amplitude as a function of the sample temperature for the CoFeB/Pt and Fe/Ta bilayers, are presented in Fig. 3c, d, respectively. The temperature-dependent longitudinal resistivities for both the samples is provided in Supplementary Section S7. From the experimentally measured THz amplitude and resistivities, we have derived the effective spin-orbital Hall resistivity, $\rho_{SOH}^{eff}$ at each temperature using the procedure discussed in Section S10 of Supplementary Information, and the results for the same as a function of squared longitudinal resistivity, $\rho_{NM}^2$ (NM = Pt, Ta) are presented in Fig. 3e, f for CoFeB/Pt and Fe/Ta, respectively. In the next paragraph, we establish sole ISHE origin of the THz signal generation through the dominance of the extracted respective spin Hall conductivities in CoFeB/Pt and Fe/Ta bilayers.

We fit the results in Fig. 3e, f using Eq. (4) to obtain $(\sigma_{SOH}^{int})^{eff}$ from the slope of the linear fit. Firstly, in the case of Fe/Ta, a negative slope value, i.e., $(\sigma_{SOH}^{int})^{eff} = (\sigma_{SH}^{int} + \eta_{L-S} \cdot \sigma_{OH}^{int}) < 0$, is obtained. As it is well known in the literature that for Ta, $\sigma_{SH}^{int}$ is negative, while, $\sigma_{OH}^{int}$ is positive[14,18]. Also, the spin-orbit correlation factor for Fe, ($\eta_{L-S}$), is known to be insignificantly positive[8]. Therefore, a negative value of $(\sigma_{SOH}^{int})^{eff}$ from the experiments directly implies that $\sigma_{SH}^{int} > \eta_{L-S} \cdot \sigma_{OH}^{int}$. Moreover, positive $\sigma_{OH}$ value of Ta demands the slope to be positive in Fe/Ta sample, which is not the case here. Hence, it can be concluded that the THz emission from Fe/Ta is entirely due to the spin-to-charge conversion via ISHE in Ta. On the other hand, a positive slope of the fit, i.e., $(\sigma_{SOH}^{int})^{eff} > 0$ is obtained in the case of CoFeB/Pt. Since, $\sigma_{SH}^{int}$, $\sigma_{OH}^{int}$ and $\eta_{L-S}$, all are positive for Pt[14,18], the positive valued effective conductivity, $(\sigma_{SOH}^{int.})^{eff} = (\sigma_{SH}^{int} + \eta_{L-S} \cdot \sigma_{OH}^{int})$ is as per the expectation for dominating spin Hall conductivity. Thanks to the negligible value of $\eta_{L-S}^{CoFeB}$, the contribution, $\eta_{L-S} \cdot \sigma_{OH}^{int}$ becomes far smaller than $\sigma_{SH}^{int}$, and this would results into the condition $\sigma_{SH}^{int} > \eta_{L-S} \cdot \sigma_{OH}^{int}$, same as in the case of Ta. The dominating $\sigma_{SH}^{int}$ dictates flow of majorly a spin current from CoFeB and its conversion to charge current takes place in Pt via ISHE. Therefore, it can be concluded from here that ISHE is the origin for THz pulse emission takes place in the CoFeB/Pt and Fe/Ta bilayers.

### CoFeB/W/Ta: A heavy metal insertion layer enhances orbital transport and hence the THz generation efficiency

In the previous two sections, we have established role of IOHE in NiFe/Nb and ISHE in CoFeB/Pt and Fe/Ta, as the principal reason for the generation of THz pulses from them. Due to comparable or even larger value of $\sigma_{OH}$ than $\sigma_{SH}$ in some heavy metals, the existence of significant OHE or IOHE is also expected in them. But to harness the effect for its direct observation, one needs to choose their appropriate combinations with the FM layers. To launch an orbital current into the heavy metal layer of a FM/NM structure, use[54] of Ni or NiFe, or similar other FM materials, would be the proper choice; otherwise, despite of large

$\sigma_{OH}$ in some heavy metals, for example, nearly an order higher $\sigma_{OH}$ than $\sigma_{SH}$ in Ta, it is very difficult to observe orbital transport in either Fe/Ta or CoFeB/Pt, the two cases described in the previous section. In this section, we show that by adding/interfacing a heavy metal W-insertion layer in technologically relevant CoFeB/Ta heterostructure, the orbital transport gets pronounced. For this study, CoFeB(2)/W(2)/Ta(2) and CoFeB(2)/W(1)/Ta(2), where, the integers inside small parentheses represent layer thickness in nm, were fabricated. The thickness and good interface quality of the W-insertion layer is confirmed by analysing the elemental stack using the secondary ion mass spectroscopy (SIMS) technique, as shown in Supplementary Section S2. The thickness constraint on the W-insertion layer is motivated by the previous studies[71,72]. We may emphasize that by adding the W-insertion layer, the magnetic properties of the trilayers are nearly unchanged from the bilayer counterparts, as confirmed from both the MH and MT measurements (see Sections S3 and S4 of Supplementary Information).

Figure 4a schematically shows the ultrafast optically excited spin current in the CoFeB layer and subsequent injection into the W-insertion layer. Heavy metal W supports efficient spin-orbit conversion[54]. Thus, within the W-insertion layer, in addition to a fractional spin-charge conversion via ISHE, an orbital current is also produced from the incoming spin current. Due to the high and negative value of the spin-orbit correlation factor $\eta_{L-S}^{W}$ of W, as explained in Fig. 4b, the spin and orbital moments are oppositely polarized during their propagation in the W-insertion layer. Subsequently, both the spin and orbital currents are injected from the W-insertion layer into the adjacent heavy metal Ta layer. Ta is known to possess[8,14] nearly an order higher and positive value of $\sigma_{OH}$ than the $\sigma_{SH}$ (see Table 1), and the corresponding spin-Hall angle is also much smaller in the α-phase Ta. Therefore, maximal charge current conversion from the orbital current via IOHE and comparatively negligible from the spin current via ISHE takes place in the Ta layer. The net charge current produced in

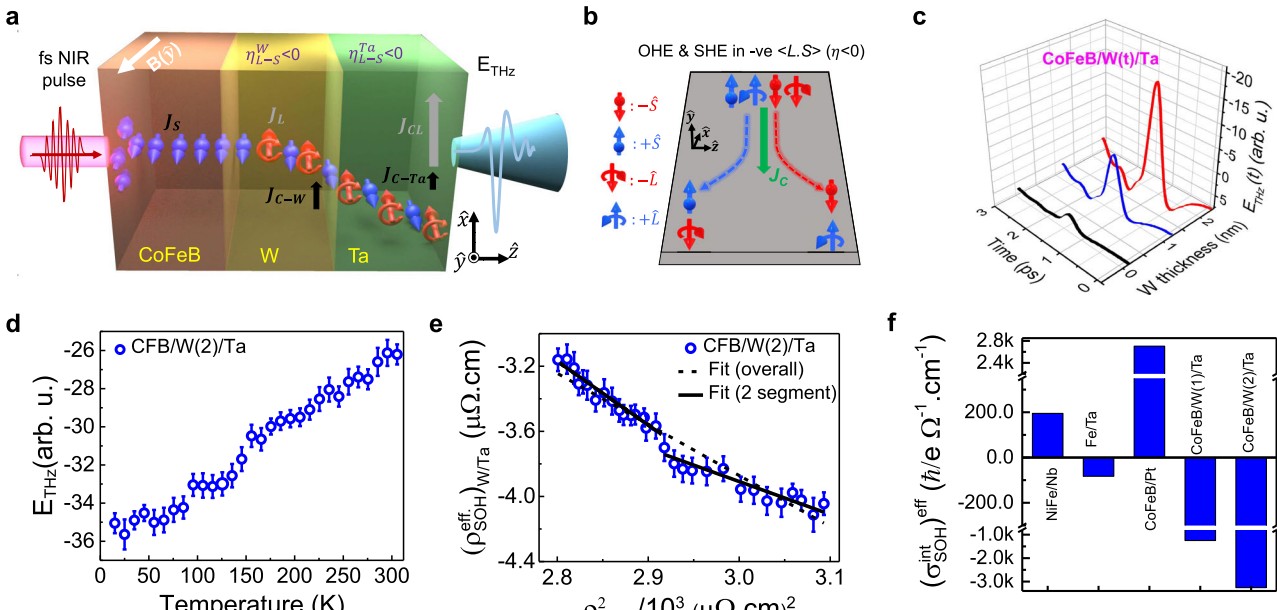

**Fig. 4 | Enhanced spin-to-orbital current conversion by W-insertion layer in CoFeB/W/Ta. a** Schematic to illustrate spin to orbital current conversion due to large negative spin-orbit correlation factor, $\eta_{L-S}^{W}$ (<0) in the W-insertion layer of the CoFeB/W/Ta heterostructure which is ultrafast optically excited from the substrate side. The transient charge currents due to spin-charge conversion in W and Ta layers are labeled as $J_{C-W}$ and $J_{C-Ta}$, respectively, while the charge current in Ta layer due to orbital-charge conversion is labeled as $J_{CL}$. **b** Schematic illustration of transport of spin and orbital moments in a material having negative spin-orbit correlation ($\eta_{L-S}$<0) such as W and Ta. **c** Time-domain traces of the THz radiation emitted by CoFeB/W(t)/Ta heterostructure with varying thickness of the W-insertion layer, t = 0, 1, 2 nm. **d** Peak-to-peak THz signal amplitude variation with respect to the sample temperature for CoFeB/W(2)/Ta. The error bars at each temperature correspond to the largest absolute deviation of the peak-to-peak THz amplitude from the mean of three measurements. **e** Effective spin-orbital Hall resistivity, $\rho_{SOH}^{eff}$ as a function of the squared longitudinal resistivity, $\rho_{W/Ta}^{2}$. The solid and dashed lines represent linear fits as discussed in the text. **f** Extracted values of effective intrinsic spin-orbital Hall conductivity, $(\sigma_{SOH}^{int})^{eff}$ for different bi- and trilayer heterostructures used in the current study.

**Table 1 | Comparison of the spin ($\sigma_{SH}$), orbital ($\sigma_{OH}$), and effective Hall $\left(\sigma_{SOH}^{int.}\right)^{eff}$ conductivities in different materials and heterostructures**

| Materials and heterostructures | $\sigma_{SH}$ ($\Omega^{-1}\cdot cm^{-1}$) | $\sigma_{OH}$ ($\Omega^{-1}\cdot cm^{-1}$) | $\left(\sigma_{SOH}^{int.}\right)^{eff}$ ($\Omega^{-1}\cdot cm^{-1}$) | $\eta_{L-S}$ |
|---|---|---|---|---|
| Pt | +2152[8]<br>+815[14]<br>+1739[18] | +2919[8]<br>+1918[14]<br>+4330[18] | | +ve |
| CoFeB/Pt | — | — | +2208[8]<br>+2140[8]<br>+2704 (this work) | — |
| Ta | −274[8]<br>−286[14]<br>−7[18] | +6803[8]<br>+3820[14]<br>+3950[18] | | -ve |
| Co/Ta | — | — | −98[8] | — |
| FeB/Ta | — | — | −359[8] | — |
| Fe/Ta | — | — | −211[8]<br>−83 (this work) | — |
| W | −324[14]<br>−83[18] | +3293[14]<br>+4490[18] | | -ve |
| CoFeB/W(1)/Ta | — | — | −1250 (this work) | — |
| CoFeB/W(2)/Ta | — | — | −3256 (this work) | — |
| Nb | −117[14]<br>−48[18] | +3641[14]<br>+5930[18] | | -ve |
| Ni/Nb | — | — | +293[59] | — |
| Co/Nb | — | — | +149[60] | — |
| NiFe/Nb | — | — | +195 (this work) | — |

Values of the spin-orbit correlation factor, $\eta_{L-S}$, wherever available, are also listed.

the CoFeB/W/Ta trilayer that is responsible for THz emission from it can be expressed by a relation like Eq. (1) as follows,

$$J_C = J_{C-ISHE} + J_{C-IOHE} = (\theta_{SH}^{Ta}\cdot J_S + \theta_{SH}^W\cdot J_S) + \theta_{OH}^{Ta}\cdot\eta_{L-S}^W\cdot J_S \quad (6)$$

Here, the signs of $\theta_{SH}^{Ta}$, $\theta_{SH}^W$, $\eta_{L-S}^W$ are negative[54,73], while it is positive for $\theta_{OH}^{Ta}$. Therefore, the transient charge currents add up constructively to generate enhanced coherent THz radiation.

Figure 4c presents time-domain traces of the THz radiation emitted from CoFeB(2)/W(t)/Ta(2) trilayer samples of varying W-insertion layer thickness from t = 0 to 2 nm. While the THz amplitude for t = 2 nm is nearly 10 times stronger than that from t = 0 nm, i.e., the CoFeB(2)/Ta(2) bilayer, the same is nearly 4 times higher than from the CoFeB(2)/W(2) bilayer (see Supplementary Section S14). The monotonic increase in the THz signal amplitude with the thickness of the W-insertion layer in Fig. 4c is in contradiction to a previous study[39] where, irrespective of the type of insertion layer material, a decrease in the THz emission efficiency with the increasing insertion layer thickness is reported. There, the results were interpreted in the context of increased spin memory loss in the insertion layer. At the same time, in few other reports in the recent literature[71,72,74,75], significant enhancement in the spin current flow due to an insertion layer was attributed to the atomically thin nature of the insertion layer. We argue that such inconsistency in the literature is arising because of the fact that conventionally, the experimental outcomes have been interpreted solely in terms of the spin current, neglecting the effect of orbital contribution altogether, though the latter can be gigantic even in the light metals with weak SOC[13,14]. The orbital current diffusion length can be several times larger than the spin counterpart in some heavy metals[7,25,48,53,59]. The orbital current due to spin-orbit conversion within the heavy metal W-insertion layer is given as[48] $J_L \propto \int_0^d J_S\eta_{L-S}^W dt$, according to which, higher thickness and stronger $\eta$, both contribute to enhance the orbital current (see Supplementary Section S16). Here, the orbital current diffusion length[54] in W is much larger than the thicknesses of the insertion layer used currently. These reasons clearly

justify our observation of W-insertion layer thickness dependent enhancement in the THz emission via IOHE in CoFeB/W/Ta. The nearly one order increase in the THz generation efficiency of CoFeB/W/Ta relative to CoFeB/Ta, as shown in Fig. 4c can be clearly attributed to the strong orbital current injection from the W layer into the Ta layer and subsequently large orbital-charge conversion via IOHE in it.

To strengthen our point that indeed the orbital current within the W-insertion layer contributes to enhance the THz pulse emission from CoFeB/W/Ta as compared to CoFeB/Ta bilayer counterpart, we now present temperature-dependent results on them. The governing role of IOHE for THz emission from CoFeB/W/Ta, due to the orbital current within the W-insertion layer that increases in proportion with its thickness, is brought out clearly. In Fig. 4d, we have plotted the variation in the THz signal amplitude as a function of CoFeB/W(2)/Ta sample temperature. The corresponding longitudinal resistivities are also recorded at each sample temperature and presented in Supplementary Section S7. The resistivity information, together with the THz amplitude data in Fig. 4d, are used to obtain the results presented in Fig. 4e, where the extracted effective Hall resistivity ($\rho_{SOH}^{eff}$) with respect to the squared longitudinal electrical resistivity ($\rho_{W/Ta}^2$) is plotted at each temperature following the same procedure as discussed earlier in the paper. Additional temperature-dependent results obtained on CoFeB/W(1)/Ta sample for the different thickness of the W-insertion layer are included in Supplementary Section S9, where more or less same temperature trends have been obtained. We have fitted the experimental data for CoFeB/W(2)/Ta in Fig. 4e using Eq. (4) to obtain the effective intrinsic spin-orbital Hall conductivity $(\sigma_{SOH}^{int.})^{eff}$ from the slope of the linear fit. The behavior of the data suggests that a two-segment linear curve fitting would be more appropriate here. Though, the degree of linearity is enhanced with two-segment linear curve fitting as compared to the single linear curve fitting, yet both approaches yield nearly the same value of the slope and there is no clear distinction in terms of low and high-resistivity regions as was the case with NiFe/Nb, discussed earlier. Therefore, similar to the cases of AHE[66–69] and SHE[63], the OHE here is expected to be dominated by the intrinsic contribution in the entire

temperature range of the Hall resistivity. From the linear fitting, we obtain $(\sigma_{SOH}^{int})^{eff} = (\sigma_{SH}^{int} + \eta_{L-S} \cdot \sigma_{OH}^{int}) \sim -3256(\hbar/e)\Omega^{-1}cm^{-1}$. Given that the values of the intrinsic spin Hall conductivities[15] of both α-phase W[76] and Ta[73] are $(\sigma_{SH}^{int})^{Ta} \sim -700(\hbar/e)\,\Omega^{-1}cm^{-1}$, and $(\sigma_{SH}^{int})^{W} \sim -103(\hbar/e)\Omega^{-1}cm^{-1}$, therefore, the large and negative value of $(\sigma_{SOH}^{int})^{eff}) \sim -3256(\hbar/e)\Omega^{-1}cm^{-1}$ clearly suggests that the term, $\eta_{L-S} \cdot \sigma_{OH}^{int}$, must be of a relatively much larger negative value. From the available literature, we find that indeed the intrinsic orbital Hall conductivity in Ta dominates[8,18] over the intrinsic spin Hall conductivity ($\sigma_{OH}^{int} \gg \sigma_{SH}^{int}$), thus, signifying the pronounced orbital current injection from W-insertion layer and its conversion to transient charge current in Ta layer of CoFeB/W/Ta heterostructure. Nearly half the value of $(\sigma_{SOH}^{int})^{eff} \sim -1250(\hbar/e)\Omega^{-1}cm^{-1}$ is obtained by analyzing the data for CoFeB/W(1)/Ta (see Supplementary Section S9), where the magnitude of the THz signal gets reduced in proportion with the thickness of the W-insertion layer. Therefore, the temperature dependent analysis of CoFeB/W/Ta samples strengthens our argument that the enhanced THz signal from CoFeB/W/Ta as compared to the CoFeB/Ta bilayer is due to the strong spin-to-orbit conversion within the W-insertion layer and subsequently efficient orbit-charge conversion via IOHE in the Ta layer of the CoFeB/W/Ta heterostructure.

Figure 4f summarizes our results on the effective intrinsic spin-orbital Hall conductivity of different heterolayer systems that are experimentally determined in a non-contact and non-invasive manner by time-domain THz emission spectroscopy. To have a ready comparison of the values for different materials and heterostructures, available from different theoretical and experimental studies in the literature, Table 1 lists them together with the values of the spin-orbit correlation factor, wherever applicable.

In summary, we have experimentally demonstrated the ultrafast optically induced orbital current and its conversion to transient charge current via IOHE by wide-range temperature-dependent THz emission measurements on multiple FM/NM heterostructures. To show the role of the spin and orbital currents exclusively, we have chosen the material combinations in the layered heterostructures such that they comprise either a heavy or a light metal layer. THz pulses emitted from them have been measured as a function of the sample temperature to disentangle the contributions from the spin and the orbital transports. Majorly the orbital-charge conversion via IOHE in NiFe/Nb and the spin-charge conversion via ISHE in CoFeB/Pt, CoFeB/W, CoFeB/Ta, and Fe/Ta, are manifested from the extracted values of effective intrinsic Hall conductivities. Further analysis of NiFe/Nb system reveals signature of different resistivity regimes dominated by either intrinsic or extrinsic contributions to OHE, which has not been seen before in any orbital Hall systems. We also find that an insertion layer of heavy metal W in the CoFeB/W/Ta heterostructure, provides a pathway to constitute an ultrafast orbital current within it and subsequently its conversion to transient charge current in the Ta layer via IOHE that significantly enhances the THz generation efficiency as compared to the CoFeB/Ta counterpart. These findings will be proven to be highly useful in efforts towards realizing ultrafast orbitronic devices as well as adding new knowledge of the underlying physics.

## Methods

Heterostructures comprising of ferromagnetic $Co_{20}Fe_{60}B_{20}$ (CoFeB) and $Ni_{90}Fe_{10}$ (NiFe) layers, and nonmagnetic Pt (platinum), Ta (tantalum), W (tungsten), Nb (niobium) material layers were created by using ultra high vacuum radio frequency magnetron sputtering technique. Bilayer systems, Sub./CoFeB(2)/Pt(3), Sub./CoFeB(2)/Ta(2), Sub./Fe(2)/Ta(3) and Sub./NiFe(5)/Nb(10), and trilayer systems, Sub./CoFeB(2)/W(2)/Ta(2) and Sub./CoFeB(2)/W(1)/Ta(2), were deposited layer by layer on 1 mm thick quartz substrates (Sub.). A nearly optimized thickness and phase of the individual layers in the heterostructures has been used. The W and Ta layers are in their α-phase. The information

related to the film thickness, roughness, and phase is obtained by using various structural and topographical measurements techniques, such as X-ray diffraction (XRD), X-ray reflectivity (XRR), and atomic force microscopy (AFM), and can be found in Supplementary Section S1. SIMS depth profile experiments reconfirmed the elemental stack and the quality of interfaces in the heterostructures (see Supplementary Section S2). The magnetic measurements shown in Supplementary Sections S3 and S4, were performed in a magnetic properties measurements system (MPMS3, Quantum Design). We have employed a closed-cycle helium optical cryostat system (SHI-4-2-XG, Janis) operating in the temperature range of 10–450 K for all the temperature-dependent electrical transport and time-domain THz experiments. Complete details of the temperature-dependent THz setup[61] are provided in Supplementary Section S5. A regenerative femtosecond amplifier (Astrella, Coherent Inc.) providing laser pulses of ~50 fs pulse duration at 1 kHz repetition rate and centered at 800 nm wavelength, were used for the THz generation and detection. The collimated optical excitation (pump) beam diameter on the sample was kept at ~3 mm. For all the THz results reported here, a pulse energy (fluence) of ~35 μJ (0.5 mJ/cm²) is used. However, results with the varying pump fluence on different samples are shown in Supplementary Section S11. All the samples were excited from the substrate side, unless specified. The emitted THz pulses were collected from behind the sample by a set of two 90⁰ off-axis gold-coated parabolic mirrors of focal length of 15 cm. THz pulses were detected by electro-optic sampling scheme in a (110)-oriented ZnTe crystal of thickness 0.5 mm by using a combination of a quarter wave plate, a Wollaston prism, a balanced photodiode, and a lock-in amplifier[61]. The THz setup was under the normal conditions of the room temperature and humidity.

## Data availability

The data that support the plots in the main text of this paper are available in Figshare at https://figshare.com/s/9a7da8c176ee1df1c00d. The data that support other findings of this study are presented in supplementary information and available from the corresponding author on request.

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

## Acknowledgements

Su.K. acknowledges the Science and Engineering Research Board (SERB), Department of Science and Technology, Government of India, for financial support through project no. CRG/2020/000892, Joint Advanced Technology Center, IIT Delhi, is also acknowledged for support through EMDTERA#5 project. We acknowledge CRF, IIT Delhi for SIMS facilities. Sa.K. acknowledges the University Grants Commission, Government of India, for Senior Research Fellowship.

## Author contributions

Su.K. supervised the work. Sa.K. conceived and carried out the experiments. Su.K. and Sa.K. analyzed the data and wrote the manuscript.

## Competing interests

The authors declare no competing interests.
