## [Peer Review File · Nature Communications]

Reviewers' Comments:

Reviewer #1:

Remarks to the Author:

Orbitronics exploits the potential of the orbital degree of freedom for information technology. Recently this topic has received increasing attention from the community. In this manuscript, the authors studied the THz generation from orbital currents in ferromagnet/nonmagnet heterostructures, including NiFe/Nb, CoFeB/Pt, CoFeB/Ta, and CoFeB/W/Ta. They concluded that the THz emission is driven by orbital effects for NiFe/Nb and CoFeB/W/Ta.

I think the arguments and reasoning presented for NiFe-based samples are not solid, undermining their conclusion's validity. Moreover, it is very likely that the THz emission from CoFeB/W/Ta arises from the inverse spin Hall effect instead of any orbital effects, as claimed by the authors.

Major issues are listed in the following.

1. The terahertz emission from anomalous Hall effect has been reported in FeMnPt by Zhang et al. (Phys. Rev. Applied 12, 054027 (2019).) and more recently in CoFeB by Liu et al. (Physical Review B 104, 064419 (2021).) and Mottamchetty et al. (arXiv:2302.07398). A net charge current perpendicular to the film can also be excited by femtosecond laser excitation. Due to the anomalous Hall effect, the net charge current is converted into a transverse transient charge current, generating THz radiation. Judging from the low signal-to-noise ratio, the signals from NiFe/Nb in this manuscript appear quite weak. The weak emission probably just comes from the anomalous Hall effect (AHE) of NiFe layers. Such a mechanism needs to be considered or excluded carefully. Otherwise, all the remaining discussions will be unreliable for readers.

2. The authors studied the temperature dependence of THz emission and temperature dependence of Hall resistivity. The authors argued that the spin Hall resistivity is proportional to the terahertz electric field according to Eqn 5. A similar equation has been used to interpret the terahertz emission from the inverse spin Hall effect, but a simple extension to the case of orbital current like Eqn 5 is much too risky. In the derivation of Eqn 5, it was implicitly assumed the orbital diffusion length is the same as the spin diffusion length, which is not supported by recent observations (Commun Phys 6, 1 (2023).). It has been well established that the diffusion length orbital current is more than one order of magnitude higher than that of the spin current. In fact, it was argued that the orbital angular momentum and spin angular momentum are fundamentally different (arXiv:2106.07928). The derivation of Eqn 5 needs to be revised. In the manuscript, the fittings using Eqn 5 were not convincing enough.

3. There is a systematic temperature difference for the two experiments, i.e., laser-induced THz emission and Hall measurement. For the experiment of the temperature dependence of THz emission, it is well known that the laser results in an average temperature increase due to the absorption of laser power. The average temperature is higher than the environmental temperature of the cryostat. But for the experiment on Hall resistivity, the temperature is expected to be close to the environmental temperature. I wonder how the temperatures are calibrated and treated in the two experiments.

4. As mentioned above, the THz emission might be from AHE, which also depends on the temperature. Thus, the temperature dependence of AHE will modify the temperature dependence of THz emission. The authors should consider the temperature dependence of AHE in the related discussion about Figures 2, 3, and 4.

5 To derive Eqn 5, it was assumed that the spin current stayed the same for all the temperatures. Matthiesen et al. (Appl. Phys. Lett. 116, 212405 (2020).) measured the temperature dependence of THz emission. When the Curie temperature of Co was much higher than the temperature under consideration, the demagnetization and spin current can be taken as constant below room temperature. However, the situation for NiFe is different because the Curie temperature of NiFe is much lower than Co. Furthermore, the Curie temperature decreases when the thickness of NiFe is reduced. In this case, one must pay attention to the temperature dependence of the spin current below room temperature.

6. After inserting a W layer, the terahertz emission increased significantly. The authors attribute the enhancement of THz emission to the generation of orbital current in the W layer. A big assumption here is that the W layer would convert spin current to orbit current. In literature, Gd or Pt was used to convert orbit current to spin current, but not W. As it has not been reported before, the role of W needs to be proved more rigorously.

The spin Hall angle of W is much greater than that of Ta, and thus the THz emission from W/FM is much stronger than Ta/FM. For example, Wu et al. have shown that the THz emission from W/Co is about ten times higher than that from Ta/Co (Advanced Materials 29, 1603031 (2017)). The authors assumed that the spin currents are the same with and without the insertion W layer, but the strength of the spin current that flows into the nonmagnetic layers depends on the thickness of the nonmagnetic layer. The calculation by Torosyan et al. (Scientific Reports 8, 1311 (2018)) shows that if the total thickness of the nonmagnetic layer is less than the spin diffusion length, the spin current will increase as the thickness of the nonmagnetic layer increases.

After inserting a W layer, both the spin Hall angle and the spin current will increase, leading to a significant enhancement of terahertz emission. The orbital effects are likely to be irrelevant for CoFeB/W/Ta, and the enhancement can be explained using the inverse spin Hall effect only. I think the conclusion drawn by the authors does not hold.

Due to the critical issues listed above, I recommend this paper be rejected.

Reviewer #2:

Remarks to the Author:

This work by Kumar and Kumar examines the nonlocal orbital current in bilayer and trilayer heterostructures by inducing femtosecond photoexcitations and measuring the resulting THz emission pulses. This work investigates three classes of systems; NiFe/Nb, FM(CoFeB,Fe)/NM(Pt,Ta), and CoFeB/W/Ta. For the first system, NiFe/Nb, the authors concluded that the emitted THz pulses are governed by orbital-to-charge conversion (or IOHE). For the second system, FM(CoFeB,Fe)/NM(Pt,Ta), on the other hand, they concluded that the emitted THz pulses are governed by spin-to-charge conversion (or ISHE). For the third system, CoFeB/W/Ta, they found an order of magnitude enhancement of the THz emission and attributed it to the IOHE due to the emergence of long-range orbital current within the W-insertion layer.

I have comments and questions on the following technical aspects.

(1) The basic idea of this work resembles that of Xu et al. [48], although the precise material systems are different. I suggest the authors clarify the difference from this earlier work.

(2) The analysis of the signal in the bilayer, NiFe/Nb, assumes that the orbital diffusion length in Nb is much larger than the spin diffusion length. But I am not sure whether this inequality holds for the nonmagnetic light metal Nb. The previous reports on the inequality dealt with heavy metals (Pt [25], W [65]) and magnetic metals (Cr [7], Ni [53]). For heavy metals, their strong spin-orbit coupling can suppress the spin diffusion length, and thus this inequality may hold. For magnetic metals, their magnetism can suppress the spin diffusion length. But for nonmagnetic light metal Nb, neither spin-orbit coupling nor magnetism can strongly suppress the spin diffusion length. Thus, whether this inequality holds is not obvious. Moreover, considering that a few previous works reported long spin diffusion length for Nb, I recommend the authors to verify the inequality explicitly. The previous experiment reported the following numbers for the spin diffusion length of Nb; 780 nm at 9 K (slightly above the superconducting transition temperature of Nb) [Appl. Phys. Lett. 65, 1460 (1994)] and 30 nm at room temperature [Phys. Rev. Applied 10, 014029 (2018)]. The previous calculation [Physics of the Solid State 51, 2211 (2009)] for Nb reported the spin relaxation length of 160 nm at the electron excitation energy 0.3 eV and 30 nm at the excitation energy 0.9 eV.

(3) In Sec. 2.1, it was mentioned that the ultrafast excitation transiently enhances the spin-orbit

coupling based on Refs. [50,52]. However, these two experiments [50,52] used the soft x-ray, whereas the present experiment uses near-infrared pulses for excitation. Considering that near-infrared pulses have much smaller excitation energy than the soft x-ray, it is not clear whether the ultrafast excitation transiently enhances the spin-orbit coupling.

(3) In Sec. 2.3, "the nearly one order of increase in the THz generation efficiency of CoFeB/W/Ta relative to CoFeB/Ta is attributed to IOHE due to the emergence of long-range orbital current within the W-insertion layer." According to this attribution, the THz generation efficiency is expected to be similarly large also for a bilayer CoFeB/W. Unfortunately, this bilayer system is not examined by the author. If this is verified, it can be strong supporting evidence for the attribution.

(4) Line 150 reads "... are used to obtain the results presented in Fig. 2(c)". However, Fig. 2 does not have subpanel (c). I suspect "Fig. 2(c)" should be "Fig. 2(b)".

Reviewer #3:

Remarks to the Author:

Nature Comms paper review: Kumar and Kumar, "Ultrafast THz probing of nonlocal orbital current in transverse multilayer metallic heterostructures"

The authors present an experimental study of the THz emission from photoexcited ferromagnetic(FM)/non-magnetic metal (NM) heterostructures in which they seek to disentangle emission driven by the inverse orbital Hall effect (IOHE) and the inverse spin Hall effect (ISHE). Based on the temperature dependence of the THz emission amplitudes, the dominant emission process in NiFe/Nb is attributed to the inverse orbital Hall effect (IOHE), whereas THz emission from Fe/Ta and CoFeB/Pt bilayers is predominantly attributed to the inverse spin Hall effect (ISHE). These results are extracted from calculated values of the effective intrinsic Hall conductivities of each structure.

To my knowledge, the disentanglement of IOHE and ISHE mechanisms using temperature dependent THz time-domain spectroscopy has not previously been reported. The subject matter sits well within the scope of work published in Nature Communications. Many of the papers referenced in the manuscript have been published in journals within the Nature stable, including Nature Communications, Nature Photonics, Nature Electronics, Nature Materials, Nature Nanotechnology, and in other quality journals including Advanced Optical Materials, Physical Review B, Physical Review Letters, and Applied Physics Letters. As such, the work is likely to be of interest and relevance to readers of Nature Communications.

Comments on the analysis presented

(i) Lines 99-101

The statement made that "...hence the conversion to orbital current is pronounced" does not appear to be well supported by reference 52 cited by the authors. The referenced paper investigates angular momentum dissipation following femtosecond laser excitation of ferromagnetic Ni. This paper states:

"We also show that electron orbits do not act as a reservoir for angular momentum" and "In 3d transition metals, sum rules relate the integral L3 XMCD signal to a linear combination of spin, S , and orbital, L , angular momentum components along the magnetization direction as $S+3/2L$The temporal evolution of $S+3/2L$ in Fig. 3b represents the first quantitative demonstration that S is transferred to the lattice and not to L on a 100 fs timescale.....This excludes L as a reservoir for S ."

(ii) Temperature dependent transmission of quartz?

The THz emission data is collected in transmission mode. The samples are pumped on the substrate side so the 800nm excitation pulses pass through the quartz substrates before reaching the film. In analysing the temperature dependence of the emission, has the temperature dependence of the pump beam transmission through quartz been checked? Different excitation

powers will clearly impact the THz emission amplitudes, so ruling this out as a potential factor would be a useful addition to the Supporting Information. Alternatively, the authors might consider recollecting this data in reflection mode.

(iii) Fig.2b and related analysis

The linear fit presented is clearly not a good fit to the data. The reliability of the extracted experimental value of $(\sigma_{\text{SOH}}^{\text{int}})_{\text{NM}}^{\text{eff}} = +281 \Omega\text{-1cm}^{-1}$ (line 175) is thus called into question. A strong and robust justification for the use of a linear fit here needs to be made by the authors. The equivalent theoretically calculated value of $+140 \Omega\text{-1cm}^{-1}$ (line 177) does not derive from the parameters stated in lines 175-176. The quoted figures would appear to suggest a value of $+170 \Omega\text{-1cm}^{-1}$. Also, given that the claimed experimental value is twice the size of the theoretical value, I am not sure I can agree with the statement (line 177) that these "compare quite well".

(iv) Fig.4e

Again a linear fit is clearly not appropriate here. The data seems to follow a curve of decreasing negative gradient. This calls into question the reliability of the extracted value of $(\sigma_{\text{SOH}}^{\text{int}})^{\text{eff}} = -3256 (\hbar/e) \Omega\text{-1cm}^{-1}$ (line 296).

(v) Comparison between emission from CoFeB/Ta and CoFeB/W/Ta

In the second half of the manuscript, further results are presented in which the authors demonstrate a ten-fold enhancement of the THz emission from CoFeB/Ta arising from the insertion of a W interlayer. It is claimed that the enhancement seen from the CoFeB/W/Ta heterostructure arises from the enhanced orbital transport in W. This section of the work raises questions which are not currently addressed by the manuscript. It is not clear how the authors can be confident that the W is acting as an interlayer and not as the key ISHE interface for the generation of THz. I would strongly suggest that a further comparison with a CoFeB/W bilayer is required in addition to the comparison between CoFeB/Ta and CoFeB/W/Ta presented in the current submission.

Referencing

A large number of non-peer reviewed arXiv works are currently cited by the authors, particularly relating to the injection of orbital currents into neighbouring heavy metal layers. While this might be expected for recently emerging research directions such as this, it would be good to see these references converted to peer reviewed journal articles as this manuscript progresses towards publication.

Other referencing queries include:

Line 193

Is the cited reference 8 correct here? I am unable to locate a reference to a negligible spin-orbit correlation factor for CoFeB and Fe in the cited paper.

Line 269

Is the cited reference 39 the most appropriate here? The cited paper did not consider pure W, as used here, as an interlayer material.

Other general comments

The technical content is generally well set out and easy to follow, however a number of abbreviations appear in the manuscript without definition and there appears to be inconsistency between the symbols used in the main text and in Table 1:

Line 91: " $\langle L, S \rangle$ ". Neither L nor S are defined. The reviewer has taken these to be orbital angular momentum (L) and spin angular momentum (S) but this needs to be made explicit within the document.

Line 102: "OHC and SHC". These abbreviations are first introduced here but not defined. These

may stand for orbital Hall conductivity (OHC) and spin Hall conductivity (SHC), however these same terms are given different symbols in Table 1, where σ_{SH} is used to represent spin Hall conductivity and σ_{OH} is used to represent orbital Hall conductivity. It is suggested that the authors might consider additions and amendments here to achieve greater clarity and consistency.

Experimental methods are clearly described, with an adequate level of detail that would enable the work to be reproduced by the reader. The experimental work is based on standard, well-tested and robust techniques. The figures generally support the content well, with appropriate use of Supplementary Material to provide useful additional images and data.

An annotated copy of the manuscript is attached. In addition to the comments above, small typing errors and queries regarding sentence construction are highlighted within.

Ultrafast THz probing of nonlocal orbital current in transverse multilayer metallic heterostructures

Sandeep Kumar and Sunil Kumar*

Femtosecond Spectroscopy and Nonlinear Photonics Laboratory,

Department of Physics, Indian Institute of Technology Delhi, New Delhi 110016, India

*Email: kumarsunil@physics.iitd.ac.in

THz generation from femtosecond photoexcited spintronic heterostructures has recently become a versatile tool for investigating ultrafast spin-transport and transient charge-current in a non-contact and non-invasive manner. The same from the orbital effects is still in the primitive stage. Here, we experimentally demonstrate orbital-to-charge current conversion in metallic heterostructures, consisting of a ferromagnetic layer adjacent to either a light or a heavy metal layer, through detection of the emitted THz pulses. Temperature-dependent experiments help to disentangle the orbital and spin components that are manifested in the respective Hall-conductivities, contributing to THz emission. NiFe/Nb shows the strongest inverse orbital Hall effect with an experimentally extracted value of effective Hall-conductivity, $(\sigma_{SOH}^{int})^{eff} \sim 280 \Omega^{-1} cm^{-1}$, while CoFeB/Pt shows maximum contribution from the inverse spin Hall effect. In addition, we observe nearly ten-fold enhancement in the THz emission due to pronounced orbital-transport in W-insertion heavy metal layer in CoFeB/W/Ta heterostructure as compared to the CoFeB/Ta bilayer counterpart.

1. INTRODUCTION

Efficient generation and detection of spin currents is required in spintronic devices for their different potential applications.^{1,2} They include, the spin-orbit torque (SOT), spin-pumping, magnetic memories, excitation of magnons, manipulating the magnetic damping, etc. Mainly, the spin Hall effect^{3,4} (SHE) and Rashba-Edelstein effect^{5,6} (REE), governed by the spin angular momentum transfer, have been invoked commonly in the generation of a spin current from a charge current. It has become clear from a few recent studies that in certain solids,⁷⁻⁹ the transport of electron's orbital angular momentum and hence the associated magnetic moment, is also responsible for different interesting phenomena in the emerging field of orbitronics. Orbital Hall effect (OHE), which was conceived^{10,11} just after the SHE,³ has often been neglected due to orbital quenching in the periodic solids.^{12,13} Bearing many similarities with the SHE, in OHE, a transverse flow of orbital angular momentum (OAM) occurs in response to a longitudinally applied electric field. In fact, fundamentally, SHE has been proposed to originate from OHE only.¹⁴⁻¹⁶ Therefore, it helps in resolving some of the conflicts in the reported values of the spin Hall conductivities for certain materials.¹⁴ A few theoretical and experimental studies in the recent literature^{13,14,17} have indicated a gigantic OHE in the light as well as several heavy metals and therefore, it has necessitated more careful investigations of SHE-based phenomena and devising new schemes to disentangle the OHE.

As indicated above, OHE is a fundamental phenomenon, which can be observed in a variety of materials, including transition metals^{13-15,17-20}, semiconductors¹¹, two-dimensional materials²¹⁻²³, etc.²⁴ Unlike the SHE, whose strength greatly depends on the spin-orbit coupling (SOC) in the material, OHE, on the other hand, can be found even in the light materials, having very weak SOC^{13,20}. After a few theoretical reports^{13,14,17} on the large orbital-Hall conductivity in both the strong and weak SOC-type materials, detailed experiments are required to obtain further insights for harnessing the same in practical applications. For instance, a high value of the Hall conductivity is always advantageous as it helps in the enhancement of spin-orbit torque (SOT), which is technologically relevant to memory applications. Conventionally, in SHE induced torque (SHT), the spin angular momentum transfer exerts a torque directly on the local magnetization of the material. However, due to the lack of direct exchange coupling⁷ of orbital angular momentum with the local magnetization, a similar direct realization of the OHE induced torque or the orbital Hall torque (OHT) was lacking. This pertinent issue has found a regenerated interest among researchers^{7,8,17,19} to make OHT based applications viable^{9,25} by devising novel orbital-spin (L-S) conversion schemes and suitable material combinations.^{19,26} It follows from the magnetization manipulation through the exerted net torque, dominated by either SHT or OHT, and applying it as a key to distinguish the orbital character indirectly from the pure spin transport. Y.G. Choi *et al.*,¹⁶ have used magneto-optical Kerr effect (MOKE) in addition to the orbital torque measurement technique¹⁶ for direct detection of orbital magnetic moment accumulation created by the charge current flow in a light metal, Ti. A non-contact method is always promising to non-invasively measure the spin and orbital transport in materials.

Like the torque method to detect the charge to spin or orbital conversion, the inverse of the SHE (ISHE) and REE (IREE) are routinely used for the detection of spin transport through the spin-charge conversion, where the spin source can be one from either spin pumping or spin Seebeck current or optical excitation, etc. The Onsager reciprocity²⁷ allows an interconversion between the orbital and charge currents, where, similar to the ISHE and IREE for the spin counterpart, here, inverse orbital Hall effect (IOHE) and inverse orbital Rashba-Edelstein effect²⁵ (IOREE) are underplay. In several studies²⁸⁻³⁷ in the last decade or so, ISHE and IREE have been utilized in the THz electromagnetic pulse generation from ultrafast photoexcited magnetic/nonmagnetic heterolayer systems. Consequently, the scheme, in conjunction with the heterolayer, is not only

54 recognized as a source of effective THz radiation,³⁸ but also a highly sensitive contactless optical probing tool³⁹⁻⁴⁶ for the
55 detection and control of the ultrafast processes at femtosecond time scale, spin-charge conversion mechanisms, demagnetization
56 dynamics and transport, interfacial properties, etc. For the case of the ISHE based spintronic THz emitters, spin current from the
57 FM or AFM layer is injected into the NM layer, where it gets converted to charge current. Therefore, heavy metal layer with
58 large SOC is desirable for efficient THz generation from a bilayer⁴⁷. Similar effects are envisaged to exist for the orbital
59 counterpart too.⁴⁸ For the THz emission utilizing the orbital transport properties, i.e., orbital-charge conversion through IOHE, a
60 material capable of generating an orbital current (J_L) is required.

61 In the current work, existence of nonlocal orbital transport is experimentally detected through IOHE mediated efficient THz
62 emission from femtosecond NIR (near infrared) pulse excited bi- and tri-layer metallic heterostructures using temperature-
63 dependent time-domain spectroscopy that has not been reported hitherto. Since, the orbital degree of angular momentum is
64 strongly correlated with the crystal field potential, therefore, the temperature-dependency of the phonon scattering would severely
65 affect the OHE and the related phenomena, microscopically. The specially chosen heterostructures, in this work, consist of FM
66 and NM material combinations, where the choice of the FM is from either CoFeB or NiFe, whereas the NM is from both the
67 light metal (Nb) as well as the heavy metals (Pt, Ta, and W). While the THz emission from CoFeB/Pt and Fe/Ta bilayers is shown
68 to originate from the ISHE, the same from NiFe/Nb is attributed primarily to the IOHE. The temperature-dependence of the THz
69 amplitude vis a vis the Hall conductivities are used to distinguish spin-to-charge and orbital-to-charge signatures. For the
70 observation of IOHE mediated THz emission from structures consisting of heavy metal layer, we fabricated a tri-layer system of
71 CoFeB/W/Ta and measured the temperature-dependent THz amplitude and the Hall conductivity. The THz emission from such
72 a tri-layer, having the W-insertion layer, interfaced with another heavy metal layer of same sign of the spin Hall angle and placed
73 side by side, is nearly one-order stronger than the CoFeB/Ta bilayer counterpart. We have presented through experimental facts
74 that the origin to this enhancement is due to efficient orbital transport in the W-insertion layer. The next section provides our
75 results and detailed discussion on them. First, the case of a NiFe/Nb bilayer is taken up, followed by the study on CoFeB/NM
76 (Pt,Ta) bilayers and, finally, the trilayer of CoFeB/W/Ta. All these heterostructures are grown on quartz substrates by using UHV
77 RF sputtering. A nearly optimized thickness and phase of the individual layers in the heterostructures has been used. The W and
78 Ta layers are in their α -phase. For all the THz results reported here, femtosecond pulses having time-duration of ~ 50 fs, central
79 wavelength of 800nm, pulse energy (fluence) of $\sim 35 \mu\text{J}$ ($0.5 \text{mJ}/\text{cm}^2$) were used. However, results with the varying pump fluence
80 on different samples are shown in S11 of the Supplementary Information. Complete details about the experimental arrangements
81 in the THz setup, material synthesis and characterization, are provided in the Supplementary Information. Briefly, they are
82 mentioned in the Experimental Section also.

83 2. RESULTS AND DISCUSSION

84 2.1 NiFe/Nb: Probing inverse orbital Hall effect in light metal through generation of THz pulses

85 Figure 1(a) schematically illustrates the emission of THz pulses from NiFe/Nb bilayer following optical excitation by linearly
86 polarized femtosecond NIR pulses. The thicknesses of the layers were kept at 5 nm for NiFe and 10 nm for Nb layer. A constant
87 external magnetic field (B), having a value just above the saturation (~ 200 Oe), is applied along the y -direction. A few consistency
88 checks to initially validate the origin for the generation of THz pulses, they were recorded in four geometries with the direction
89 of the optical excitation and the magnetic field, as presented in Figs. 1(b) and 1(c), respectively. NiFe is a popular FM material
90 for spintronic applications. Dominant presence ($\geq 90\%$) of Ni (light element) in NiFe makes it possess a large and positive value
91 of spin-orbit correlation factor,⁸ $\langle L \cdot S \rangle = \eta > 0$. As shown in Fig. 1(d), in positive spin-orbit correlation materials, transverse
92 orbital and spin Hall effects are induced in response to the flow of a longitudinal charge current, J_C such that the polarization
93 direction of the accumulated orbital and spin magnetic moments is the same.⁸ An optimal material composition and growth of
94 NiFe enriched with Ni can provide a better value of η as compared to that in Ni.⁴⁹ Moreover, a transient change in the spin-orbit
95 coupling in Ni, triggered by an ultrafast excitation, can also enhance the η value significantly.⁵⁰ This is clearly an advantage with
96 NiFe for an efficient spin-to-orbital current conversion.

97 Following the optical pulse excitation of NiFe/Nb (Fig. 1a), ultrafast demagnetization⁵¹ in the NiFe layer stimulates flow of
98 a spin current with density J_S . Due to large positive value of η in the NiFe, a fraction of the ultrafast spin current is converted
99 into an ultrafast orbital current (J_L) of same polarity through the L-S conversion, given^{8,48} as, $J_L = \eta_{L-S} J_S$. In comparison to
100 other methods,^{7,8} the ultrafast excitation transiently enhances the spin-orbit coupling^{50,52} and hence the conversion to orbital
101 current is pronounced. Consequently, as indicated in Fig. 1(a), both the spin and orbital currents are now launched into the
102 adjacent Nb layer. A very weak negative SOC strength and about an order difference in the values of OHC and SHC in Nb make
103 it a suitable candidate⁵³ for realizing orbital transport phenomena. In case of heavy metals, although the value of OHC is typically
104 much larger than the SHC,¹⁴ however, OHE is greatly suppressed by the inherently present SOC owing to the identical
105 macroscopic geometries between OHE and SHE in response to the applied electrical current.²⁰ Therefore, to overcome the effect
106 of spin transport in the realization of OHE, selection of an appropriate light element material is regarded as one of the solutions.

Fig. 1. Orbital-to-charge current conversion mediated THz emission from NiFe/Nb bilayer. (a) Schematic illustration of ultrafast optically induced spin current J_S and orbital current J_L in the NiFe and their injection into the Nb layer producing ultrafast charge current and hence emission of THz pulses. The positive value of η_{NiFe}^{L-S} represents the spin-to-orbital current conversion efficiency in NiFe. J_{CS} and J_{CL} represent charge currents converted from J_S and J_L through the ISHE and IOHE, respectively. B represents the external applied magnetic field. (b) Temporal profiles of THz signal obtained from NiFe/Nb under four experimental geometries with the direction of the optical excitation and the magnetic field: (b) 1. substrate side excitation; 2. Nb film side excitation, while keeping B fixed along \hat{y} -direction, (c) 1. $B(+\hat{y})$; 2. $B(-\hat{y})$, while keeping the optical excitation from the substrate side. (d) Schematic of OHE and SHE in a prototype material. A charge current, J_C in the $-\hat{x}$ -direction induces accumulation of spin (S) and orbital (L) angular momenta in the transverse \hat{z} -direction. For positive spin-orbit correlation, $\eta > 0$, both the spin and orbital polarizations are parallel along either $+\hat{y}$ -direction or $-\hat{y}$ -direction.

The spin and orbital currents injected into the Nb layer convert to respective in-plane transient charge currents. Charge current, J_{CS} is produced via ISHE and J_{CL} is produced via IOHE. The magnitude and polarity of the emitted THz radiation is finally dependent on the net transient charge current in the Nb layer. Typically, the orbital diffusion length in Nb is much larger than the spin diffusion length.⁵⁴ Therefore, we chose Nb of thickness ~ 10 nm within which, the ISHE signal is nearly quenched as against the IOHE. The net charge current in the Nb layer is the vectorial sum of the two charge currents, i.e., $J_C = J_{C-ISHE} + J_{C-IOHE}$, and it can be further expressed as

$$J_C = \theta_{SH} \cdot J_S + \theta_{OH} \cdot J_L = \theta_{SH}^{Nb} \cdot J_S + \theta_{OH}^{Nb} \cdot \eta_{L-S}^{NiFe} \cdot J_S \quad (1)$$

Here, θ_{SH} and θ_{OH} are the spin and orbital Hall angles of Nb, which are negative and positive, respectively.^{15,18} Since the spin-orbit correlation factor, η_{L-S}^{NiFe} is positive for the NiFe, therefore, the charge current contribution from the individual components would have opposite signs, as represented in Fig. 1(a). The net charge current is, therefore, the difference of the two and hence the dominant one among them would control the emission of THz radiation. Because the spin diffusion length and spin Hall angle in Nb are much smaller, the net current is dictated by the orbital current, and hence THz emission from NiFe/Nb takes place via IOHE mainly. A few consistency checks are carried out in Figs. 1(b) and 1(c) to reveal the magnetic origin of the THz signal. The THz signal polarity is inverted by reversing either the direction of the optical excitation while keeping the magnetic field unchanged or the direction of the external magnetic field while keeping the direction of optical excitation unchanged. In the first case, the THz sign reversal is due to the change in the direction of the spin current and consequently the flow of orbital current, whereas, in the second case, it is due to the change in the spin current polarization and subsequently the polarization of orbital current as both are constrained by positive η_{L-S}^{NiFe} .

Temperature-dependent studies⁵⁵⁻⁵⁷ have been proven to be important in identifying various scattering mechanisms in the materials and subsequent determination of respective Hall conductivities. To validate the IOHE origin behind the THz emission from NiFe/Nb bilayer, we have performed temperature-dependent experiments by monitoring the changes in the peak-to-peak amplitude of the THz signal as a function of the sample temperature (T) from 10 K to 300 K. The temperature sensitivity of the phonon scattering would have a significant impact on the OHE and associated phenomena because the orbital degree of angular momentum is tightly coupled with the crystal field potential.^{13,20} The result is presented in Fig. 2(a), where the error bars represent the largest absolute deviation from the mean of three THz signal outputs at each temperature. As indicated in the inset of the figure, the sample is optically excited from the substrate side and the same orientation for sample excitation is followed in all the results later on in the paper. Since, the sign and magnitude of the emitted THz pulse is directly related with the SHC, hence, by looking at the THz signal's amplitude and phase, a qualitative information about the SHC can be obtained.³² For exactly determining the dominating role of either the SHC or OHC, detailed analysis is presented below.

By employing the four-point van der Pauw method, electrical longitudinal resistivity (ρ) was also determined for the Nb and NiFe/Nb films in the entire experimental temperature range (see inset of Fig. 2(a)). The resistivity information, together with the THz amplitude data in Fig. 2(a), are used to obtain the results presented in Fig. 2(c), where we have plotted the behavior of extracted effective spin(orbital) Hall resistivity (ρ_{SOH}^{eff}) with respect to the squared longitudinal resistivity (ρ_{NM}^2) of NM = Nb. See supplementary information S10 for details to obtain ρ_{SOH}^{eff} . This data is analyzed in the light of a temperature scaling relation⁵⁵⁻⁵⁸ for the spin Hall resistivity (ρ_{SH}) as,

$$\rho_{SH}(T) = \sigma_{SH}^{int} \cdot \rho_{NM}^2(T) + \sigma_{SJ} \cdot \rho_{0,NM}^2 + \alpha_{SS} \cdot \rho_{0,NM} \quad (2)$$

Here, σ_{SH}^{int} , σ_{SJ} , $\rho_{0,NM}$, α_{SS} are intrinsic spin Hall conductivity, side-jump spin Hall conductivity, residual resistivity, and skew scattering angle of the NM layer, respectively. The second and the third terms on the right-hand side of Eq. (2) represent the extrinsic contributions to scattering. Because of the correspondence between the SHE and OHE, a similar equation for the orbital Hall resistivity (ρ_{OH}) can be constructed by accounting for the intrinsic and extrinsic orbital scattering processes.^{8,15,20} Hence, temperature-dependence of ρ_{OH} can be expressed as,

$$\rho_{OH}(T) = \eta_{L-S} \cdot \sigma_{OH}^{int} \cdot \rho_{NM}^2(T) + Ex \quad (3)$$

On the right side of the above equation, the first term, consisting of the intrinsic orbital Hall conductivity, σ_{OH}^{int} , represents the intrinsic scattering, while the extrinsic contributions are represented by the second term, Ex . The extrinsic contribution to the spin or the orbital Hall resistivity in the above equations, is usually neglected for pure materials^{57,59} while, it can be significantly high for materials with high impurity concentration. For capturing the temperature-dependence of the effective Hall resistivity, Eqs. (2) and (3) can be combined and ρ_{SOH}^{eff} is expressed as following,

$$\rho_{SOH}^{eff} = (\sigma_{SH}^{int} + \eta_{L-S} \cdot \sigma_{OH}^{int}) \cdot \rho_{NM}^2 = (\sigma_{SOH}^{int})_{NM}^{eff} \cdot \rho_{NM}^2 \quad (4)$$

For simplicity, we have ignored the temperature variation of the extrinsic terms in both the spin and the orbital Hall resistivities. Also, we have used $(\sigma_{SOH}^{int})_{NM}^{eff} = (\sigma_{SH}^{int} + \eta_{L-S} \cdot \sigma_{OH}^{int})$ to represent the effective intrinsic spin(orbital) Hall conductivity. As shown in S10 of the supporting information, the effective Hall resistivity is related to the amplitude of the THz signal,^{55,58} and it is given by the relation,

$$\rho_{SOH}^{eff} = E_{THz} \cdot \left(\frac{\rho_{xx}^{NM}}{\rho_{FM/NM}} \right) \left(\frac{d}{\lambda_{rel}} \right) \frac{1}{e \cdot J_s} \quad (5)$$

The extracted data for ρ_{SOH}^{eff} vs ρ_{NM}^2 at each temperature, has been presented in Fig. 2(b), where, the continuous curve represents a linear fit with slope, $(\sigma_{SOH}^{int})_{NM}^{eff}$ as per Eq. (4). With the information at hand that, for Nb, σ_{SH}^{int} is negative and η_{L-S} is positive, a positive value of $(\sigma_{SOH}^{int})_{NM}^{eff}$ from Fig. 2(b) clearly reveals that $\eta_{L-S} \cdot \sigma_{OH}^{int}$ for Nb is a large positive value. The experimental value of $(\sigma_{SOH}^{int})_{NM}^{eff}$ as obtained from Fig. 2(b) is $\sim +281 \Omega^{-1} cm^{-1}$. From the theoretically provided values of different parameters of Nb film, i.e., $\sigma_{SH}^{int} (\Omega^{-1} cm^{-1}) \cong -100^{14,18}$, $\eta_{L-S} \cong 0.045^8$, and $\sigma_{OH}^{int} (\Omega^{-1} cm^{-1}) \cong 6000^{18}$, we obtain, $(\sigma_{SOH}^{int})_{NM}^{eff} = (\sigma_{SH}^{int} + \eta_{L-S} \cdot \sigma_{OH}^{int}) \sim +140 \Omega^{-1} cm^{-1}$. Clearly, the theoretically calculated and experimentally obtained values of $(\sigma_{SOH}^{int})_{NM}^{eff}$ compare quite well signifying the origin of the orbital current in the NiFe layer that governs the THz emission from NiFe/Nb via IOHE, provided its efficient conversion to the transient charge current in Nb takes place. In fact, theoretical predictions of a gigantic +ve value of OHC over a small -ve value of SHC in Nb⁶⁰, are well supported by our experimental observations.

Fig. 2. Temperature-dependent THz emission from NiFe/Nb (Nb = light metal). (a) Peak-to-peak value of THz amplitude as a function of varying sample temperature. The error bars at each temperature correspond to the largest absolute deviation of the peak-to-peak THz amplitude from the mean of three measurements. Inset: The optical excitation from the substrate side; and temperature-dependent resistivity (ρ) variations of NiFe/Nb and Nb samples measured by the four-point van der Pauw method. (b) Effective spin(orbital) Hall resistivity as a function of the squared longitudinal resistivity for the Nb film. Solid line is fit to the data using Eq. (4).

2.2 FM(CoFeB,Fe)/NM(Pt,Ta): ISHE mediated THz emission from heavy metal layers based FM/NM heterostructures

Experimental results are now presented and discussed on Fe/Ta and CoFeB/Pt, in conjunction with those on CoFeB/Ta, presented elsewhere,⁵⁵ and re-examined again as presented in the supplementary information S8. These bilayer heterostructures contain a heavy metal layer, either Pt or Ta, and are chosen selectively for their opposite sign of the spin Hall angle.^{32,34} Moreover, CoFeB and Fe are selected because of their negligible⁸ spin-orbit correlation factor, i.e., η_{L-S}^{CoFeB} and $\eta_{L-S}^{Fe} \sim 0$ as compared to Ni or NiFe. By these choices, we initially ensure that there is no fractional conversion of the spin current into orbital current within the FM layer. Consequently, from the femtosecond pulse excited FM/NM bilayers, THz generation takes place via ISHE only. For the CoFeB/Pt with the Pt layer having positive spin Hall angle ($\theta_{SH}^{Pt} > 0$), the experimental configuration is depicted in Fig. 3(a), where the directions of the external magnetic field, spin current, charge currents and the THz signal polarities are indicated. Owing to the opposite sign of the spin Hall angle in Ta ($\theta_{SH}^{Ta} < 0$), emission of opposite polarity THz signal from Fe/Ta, under the same experimental configuration, is depicted in Fig. 3(b). In both cases, the THz signal polarity dependence on the experimental configuration, including the direction of the external magnetic field and the optical excitation, are found to be as expected from the ISHE mediated THz emission.^{29,32} Variations of the peak-to-peak THz signal amplitude as a function of the sample temperature for the CoFeB/Pt and Fe/Ta bilayers, are presented in Figs. 3(c) and 3(d), respectively. The temperature-dependent longitudinal resistivities for both the samples is provided in the supplementary information S7. From the experimentally measured THz amplitude and resistivities, we have derived the effective Hall resistivity, ρ_{SOH}^{eff} at each temperature using the procedure discussed in S10 of Supplementary information, and the results for the same as a function of squared longitudinal resistivity, ρ_{NM}^2 ($NM = Pt, Ta$) are presented in Figs. 3(e) and 3(f) for CoFeB/Pt and Fe/Ta, respectively. In the

207 next paragraph, we establish, sole ISHE origin of the THz signal generation through the dominance of the extracted respective
 208 spin Hall conductivities in CoFeB/Pt and Fe/Ta bilayers.

209 We fit the results in Figs. 3(e) and 3(f) using Eq. (4) to obtain $(\sigma_{SOH}^{int})^{eff}$ from the slope of the linear fit. Firstly, in the case
 210 of Fe/Ta, a negative slope value, i.e., $(\sigma_{SOH}^{int})^{eff} = (\sigma_{SH}^{int} + \eta_{L-S} \cdot \sigma_{OH}^{int}) < 0$, is obtained. As it is well known in the literature that
 211 for Ta, σ_{SH}^{int} is negative, while, σ_{OH}^{int} is positive.^{14,18} Also, the spin-orbit correlation factor for Fe, (η_{L-S}) , is known to be
 212 insignificantly positive.⁸ Therefore, a negative value of $(\sigma_{SOH}^{int})^{eff}$ directly implies that $\sigma_{SH}^{int} > \eta_{L-S} \cdot \sigma_{OH}^{int}$. Moreover, positive
 213 OHC value of Ta demands the slope to be positive in Fe/Ta sample, which is not the case here. Hence, it can be concluded that
 214 the THz emission from Fe/Ta is entirely due to the spin-to-charge conversion via ISHE in Ta. On the other hand, a positive slope
 215 of the fit, i.e., $(\sigma_{SOH}^{int})^{eff} > 0$ is obtained in the case of CoFeB/Pt. Since, σ_{SH}^{int} , σ_{OH}^{int} and η_{L-S} , all are positive for Pt,^{14,18} the
 216 positive valued effective conductivity, $(\sigma_{SOH}^{int})^{eff} = (\sigma_{SH}^{int} + \eta_{L-S} \cdot \sigma_{OH}^{int})$ is as per the expectation. Thanks to the negligible value
 217 of η_{L-S}^{CoFeB} , the contribution, $\eta_{L-S} \cdot \sigma_{OH}^{int}$ becomes far smaller than σ_{SH}^{int} , and this would results into the condition $\sigma_{SH}^{int} > \eta_{L-S} \cdot \sigma_{OH}^{int}$,
 218 same as in the case of Ta. The dominating σ_{SH}^{int} dictates flow of majorly a spin current from CoFeB and its conversion to charge
 219 current takes place in Pt via ISHE. Therefore, it can be concluded here that an ISHE mediated THz pulse emission takes place in
 220 the CoFeB/Pt bilayer.

221 **Fig. 3. Temperature-dependence of THz signal and effective Hall resistivity for CoFeB/Pt and Fe/Ta bilayers (Pt, Ta = heavy**
 222 **metal).** Schematic illustration of the ultrafast optical excitation, spin magnetic moment transport and its conversion to transient charge
 223 current through ISHE to emit a THz pulse from (a) CoFeB/Pt and (b) Fe/Ta. The spin Hall angle (θ_{SH}) and the direction of the external
 224 magnetic field are indicated. Temperature-dependent variation of the THz signal peak-to-peak amplitude for (c) CoFeB/Pt, and (d) Fe/Ta.
 225 Effective Hall resistivity, ρ_{SOH}^{eff} as a function of the squared longitudinal resistivity, ρ_{NM}^2 for (e) Pt and (f) Ta. The continuous curves in (e)
 226 and (f) are linear fits to the data using Eq. (4).
 227

228 2.3 CoFeB/W/Ta: A heavy metal insertion layer enhances orbital transport and hence the THz generation efficiency

229 In the above two sections, we have established roles of solely IOHE in NiFe/Nb and ISHE in CoFeB/Pt and Fe/Ta, as the
 230 reason for the generation of THz pulses from them. Due to the large OHC, the existence of high IOHE is expected in some heavy
 231 metals. But to harness the effect for its direct observation, one needs to choose their appropriate combinations with the FM layers.
 232 To launch an orbital current into the heavy metal layer, use⁵⁴ of Ni or NiFe, or similar other FM layers, would be the proper
 233 choice; otherwise, despite of large OHC in heavy metals, especially, nearly an order high OHC than SHC in Ta, it is very difficult
 234 to observe orbital transport in either Fe/Ta or CoFeB/Pt. Here, we show that by adding/interfacing a heavy metal W-insertion
 235 layer in technologically relevant CoFeB/Ta heterostructure, the orbital transport gets pronounced. For this study,
 236 CoFeB(2)/W(2)/Ta(2) and CoFeB(2)/W(1)/Ta(2), where, the integers inside small parentheses represent layer thickness in nm,
 237 were fabricated. The thickness and good interface quality of the W-insertion layer is confirmed by analyzing the elemental stack
 238

239
240
241
242
243

using secondary ion mass spectroscopy (SIMS) technique, as shown in supplementary information S2. The thickness constraint on the W-insertion layer is motivated by the previous studies.^{61,62} We may emphasize that by adding the W-insertion layer, the magnetic properties of the trilayers are nearly unchanged from the bilayer counterparts, as confirmed from both the MH and MT-measurements (see S3 and S4 of the supplementary information).

244
245
246
247
248
249
250
251
252
253

Fig. 4. Enhanced spin-to-orbital current conversion by W-insertion layer in CoFeB/W/Ta. (a) Schematic to illustrate spin current conversion into orbital current by large negative spin-orbit correlation factor, $\eta_{L-S}^W (< 0)$ of the W-insertion layer in the CoFeB/W/Ta heterostructure, ultrafast optically excited from the substrate side. The transient charge currents due to spin-charge conversion in W and Ta layers are labelled as J_{C-W} and J_{C-Ta} , respectively. The charge current in Ta layer due to orbital-to-charge conversion is represented by J_{CL} . (b) Schematic illustration of spin and orbital magnetic moments in a material of negative spin-orbit correlation ($\eta_{L-S} < 0$), i.e., W and Ta. (c) THz time-domain traces with varying thickness, $t = 0, 1, 2$ nm of the W-insertion layer in CoFeB/W(t)/Ta heterostructure. (d) Peak-to-peak THz signal amplitude variation with respect to the temperature of CoFeB/W(2)/Ta. (e) Effective Hall resistivity, ρ_{SOH}^{eff} as a function of the squared longitudinal resistivity, $\rho_{W/Ta}^2$. (f) Extracted values of effective intrinsic Hall conductivity, $(\sigma_{SOH}^{int})^{eff}$ for different bi- and trilayer heterostructures used in the current study.

254

255

Figure 4(a) schematically shows the ultrafast optically excited spin current injection into the W layer, where it induces the orbital current generation through the efficient L-S conversion in the W layer, and also to charge current via ISHE in W. The α -phase W here, has a distinctive quality of generating spin current induced orbital current⁵⁴. Due to the negative η_{L-S}^W of W, as explained in Fig. 4(b), the orbital and spin magnetic moments are oppositely polarized in the propagation of spin and orbital current. Both spin and orbital current are further encountered to the adjacent Ta heavy metal layer. Ta has been shown^{8,14} to possess more than an order higher and positive value of OHC than its SHC (see Table I); therefore, maximal conversion of orbital current to charge current is expected. On the other hand, the conversion from spin to charge current in α -phase Ta is weaker due to its small value of SHC and hence the spin Hall angle. Thus, the total current generated from the heterostructure is the sum of the charge current converted either from ISHE or IOHE, and can be expressed, similar to Eq. 1, as

264

$$J_C = J_{C-ISHE} + J_{C-IOHE} = (\theta_{SH}^{Ta} \cdot J_S + \theta_{SH}^W \cdot J_S) + \theta_{OH}^{Ta} \cdot \eta_{L-S}^W \cdot J_S \quad (6)$$

265

Here, the sign of θ_{SH}^{Ta} , θ_{SH}^W , η_{L-S}^W is negative, while it is positive for θ_{OH}^{Ta} , therefore, the total sum of transient charge current adds up constructively to give the efficient generation of THz radiation.

266

Figure 4(c) presents THz time-domain traces emitted from CoFeB/W(t)/Ta trilayer samples of varying thickness of the W-insertion layer from $t = 0$ to 2 nm. An increasing trend in the THz amplitude with t can be clearly seen. This observation is in contradiction with a previous study³⁹ where, irrespective of the type of insertion layer material, a monotonic decrease in the THz

267
268
269

270 emission efficiency with the increasing insertion layer thickness is reported. There, the results were interpreted in the context of
 271 increased spin memory loss in the insertion layer. At the same time, in few other reports in the recent literature,⁶¹⁻⁶⁴ significant
 272 enhancement in the spin current flow due to an insertion layer was attributed to the atomically thin nature of the insertion layer.
 273 We argue that such inconsistency in the literature is arising because of the fact that conventionally, the experimental outcomes
 274 have been interpreted solely in terms of the spin current, neglecting the effect of orbital contribution altogether, though the latter
 275 can be gigantic even in the light metals with weak SOC.^{13,14} The orbital current diffusion length can be several times larger than
 276 the spin counterpart.^{7,25,48,53,65} The induced orbital current within the W-insertion layer is given by⁴⁸ $J_L \propto \int_0^d J_S \eta_{L-S}^W dt$, according
 277 to which, higher thickness and strength of η , are favourable. These reasons clearly justify our observation of W-insertion layer
 278 thickness dependent enhancement in the THz emission via IOHE in CoFeB/W/Ta. Much importantly, the nearly one order
 279 increase in the THz generation efficiency of CoFeB/W/Ta relative to CoFeB/Ta, as shown in Fig. 4(c), is attributed to IOHE due
 280 to the emergence of long-range orbital current within the W-insertion layer.

281 **TABLE. 1.** Comparison of the spin (σ_{SH}), orbital (σ_{OH}), and effective Hall conductivity,
 282 $(\sigma_{SOH}^{int})^{eff}$ in different materials and heterostructures. Values, where available, of the spin-orbit correlation factor, η_{L-S}
 283 are also listed.

Materials and heterostructures	σ_{SH} ($\Omega^{-1} \cdot cm^{-1}$)	σ_{OH} ($\Omega^{-1} \cdot cm^{-1}$)	$(\sigma_{SOH}^{int})^{eff}$ ($\Omega^{-1} \cdot cm^{-1}$)	η_{L-S}
Pt	+2152 ⁸ +815 ¹⁴ +1739 ¹⁸	+2919 ⁸ +1918 ¹⁴ +4330 ¹⁸		+ve
CoFeB/Pt	--	--	+2208 ⁸ +2140 ⁸ +2704 (this work)	--
Ta	-274 ⁸ -286 ¹⁴ -7 ¹⁸	+6803 ⁸ +3820 ¹⁴ +3950 ¹⁸		-ve
Co/Ta	--	--	-98 ⁸	--
FeB/Ta	--	--	-359 ⁸	--
Fe/Ta	--	--	-211 ⁸ -83 (this work)	--
W	-324 ¹⁴ -83 ¹⁸	+3293 ¹⁴ +4490 ¹⁸		-ve
CoFeB/W(1)/Ta	--	--	-1250 (this work)	--
CoFeB/W(2)/Ta	--	--	-3256 (this work)	--
Nb	-117 ¹⁴ -48 ¹⁸	+3641 ¹⁴ +5930 ¹⁸		-ve
Ni/Nb	--	--	+293 ⁵³	--
Co/Nb	--	--	+5 ⁶⁶	--
NiFe/Nb	--	--	+281 (this work)	--

284 To strengthen our point that, indeed the orbital current within the W-insertion layer contributes to enhance the THz pulse
 285 emission from CoFeB/W/Ta as compared to CoFeB/Ta bilayer counterpart, we now present temperature-dependent results on
 286 them. The governing role of IOHE for THz emission from CoFeB/W/Ta, due to the orbital current within the W-insertion layer
 287 that increases in proportion with its thickness, is brought out clearly. In Fig. 4(d), we have plotted the variation in the THz signal
 288 amplitude as a function of CoFeB/W(2)/Ta sample temperature. The corresponding longitudinal resistivities are also recorded at
 289 each sample temperature and presented in supplementary section S7. The resistivity information, together with the THz amplitude
 290 data in Fig. 4(d), are used to obtain the results presented in Fig. 4(e), where we have plotted the behavior of extracted effective
 291 Hall resistivity (ρ_{SOH}^{eff}) with respect to the squared longitudinal electrical resistivity ($\rho_{W/Ta}^2$) following the same procedure
 292 discussed earlier in the paper. Additional results obtained on CoFeB/W(1)/Ta sample with different thicknesses of the W-
 293 insertion layer, are included in the supplementary section S9. We fit the experimental data in Fig. 4(e) using Eq. (4) to obtain
 294 $(\sigma_{SOH}^{int})^{eff}$ from the slope of the linear fit. Clearly, the data fits for a negative slope, i.e., $(\sigma_{SOH}^{int})^{eff} = (\sigma_{SH}^{int} + \eta_{L-S} \cdot \sigma_{OH}^{int}) < 0$
 295 and provides a value of $(\sigma_{SOH}^{int})^{eff} \sim -3256 (\hbar/e) \Omega^{-1} cm^{-1}$, which is much larger than the theoretical values of the intrinsic
 296

297 spin Hall conductivities¹⁵ of both α -phase W⁶⁷ and Ta⁶⁸, i.e., $\sim -700 (\hbar/e) \Omega^{-1} \text{cm}^{-1}$, and $\sim -103 (\hbar/e) \Omega^{-1} \text{cm}^{-1}$,
 298 respectively. Therefore, it simply suggests the dominance of $\eta_{L-S} \cdot \sigma_{OH}^{int}$ term in the overall effective intrinsic Hall conductivity
 299 which is negatively large due to much larger orbital Hall conductivity than the spin Hall conductivity, $\sigma_{OH}^{int} \gg \sigma_{SH}^{int}$, in Ta making
 300 the orbital to charge conversion more pronounced. A value of $(\sigma_{SOH}^{int})^{eff} \sim -1250 (\hbar/e) \Omega^{-1} \text{cm}^{-1}$ is obtained by us for the
 301 CoFeB/W(1)/Ta, where the magnitude of the THz signal reduces in proportion with the thickness of the W-insertion layer.
 302 Therefore, the large negative value of $(\sigma_{SOH}^{int})^{eff}$ for CoFeB/W/Ta strengthens our argument that the enhanced THz signal from
 303 CoFeB/W/Ta is due to the orbital current generation within the W-insertion layer, thus providing an additional charge current by
 304 orbital to charge current conversion via IOHE in the Ta-layer of the heterostructure.

305 **Figure 4(g)** summarizes our results on the effective intrinsic spin(orbital) Hall conductivity of different heterolayer systems
 306 that are experimentally determined in a non-contact manner and non-invasively by THz emission time-domain spectroscopy. To
 307 have a ready comparison of the values for different materials and heterostructures, available from different resources, either
 308 experimental or theoretical, Table 1 lists them together with the values of the spin-orbit correlation factor, wherever applicable.

309 3. CONCLUSION

310 In summary, we have experimentally demonstrated the ultrafast optically induced orbital current and its conversion to
 311 transient charge current via IOHE by temperature-dependent THz emission measurements. To show the role of the spin and
 312 orbital currents exclusively, we have chosen the material combinations in the layered heterostructures such that they comprise of
 313 either a heavy or a light metal layer. THz pulses emitted from them have been measured, where temperature-dependence of the
 314 THz amplitude helps to disentangle the contributions from the spin and the orbital transports. Purely orbital-charge conversion
 315 via IOHE in NiFe/Nb and spin-charge conversion via ISHE in CoFeB/Pt and Fe/Ta, are manifested from the extracted values of
 316 effective intrinsic Hall conductivities. We also find that an insertion layer of W heavy metal in CoFeB/W/Ta, provides a pathway
 317 to constitute an ultrafast orbital current within it and subsequently its conversion to transient charge current in the Ta layer via
 318 IOHE that significantly enhances the THz generation efficiency as compared to the CoFeB/Ta counterpart. These findings will
 319 be proven to be highly useful in efforts towards realizing ultrafast orbitronic devices as well as adding new knowledge of the
 320 underlying physics.

321 METHODS

322 Heterostructures comprising of ferromagnetic Co₂₀Fe₆₀B₂₀ (CoFeB) and Ni₉₀Fe₁₀ (NiFe) layers, and nonmagnetic Pt
 323 (platinum), Ta (tantalum), W (tungsten), Nb (niobium) material layers were created by using ultra high vacuum radio frequency
 324 magnetron sputtering technique. Bilayer systems, Sub./CoFeB(2)/Pt(3), Sub./CoFeB(2)/Ta(2), Sub./Fe(2)/Ta(3) and
 325 Sub./NiFe(5)/Nb(10), and trilayer systems, Sub./CoFeB(2)/W(2)/Ta(2) and Sub./CoFeB(2)/W(1)/Ta(2), were deposited layer by
 326 layer on 1 mm thick quartz substrates (Sub.). The information related to the film thickness, roughness and phase is obtained by
 327 using various structural and topographical measurements techniques, such as X-ray diffraction (XRD), X-ray reflectivity (XRR),
 328 and atomic force microscopy (AFM), and can be found in the Supplementary Section S1. SIMS depth profile experiments
 329 reconfirmed the elemental stack and the quality of interfaces in the heterostructures (see supplementary information section S2).
 330 The magnetic measurements shown in Supplementary Sections S3 and S4, were performed in a magnetic properties
 331 measurements system (MPMS3, Quantum Design). We have employed a closed-cycle helium optical cryostat system (SHI-4-2-
 332 XG, Janis) operating in the temperature range of 10–450 K for all the temperature-dependent electrical transport and time-domain
 333 THz experiments. Complete details of the temperature-dependent THz setup⁵⁵ are provided in the supplementary information
 334 section S5. A regenerative femtosecond amplifier (Astrella, Coherent Inc.) providing laser pulses of ~ 50 fs pulse duration at 1
 335 kHz repetition rate and centred at 800 nm wavelength, were used for the THz generation and detection. The collimated optical
 336 excitation (pump) beam diameter on the sample was kept at ~ 3 mm. All the samples were excited from the substrate side, unless
 337 specified. The emitted THz pulses were collected from behind the sample by a set of two 90° off-axis gold-coated parabolic
 338 mirrors of focal length of 15 cm. THz pulses were detected by electro-optic sampling scheme in a (110)-oriented ZnTe crystal
 339 of thickness 500 micron by using a combination of a quarter wave plate, a Wollaston prism, a balanced photodiode and a lock-
 340 in amplifier.⁵⁵ The THz setup was under the normal conditions of the room temperature and humidity.

- 345 1 Hirohata, A. *et al.* Review on spintronics: Principles and device applications. *Journal of Magnetism and Magnetic Materials*
346 **509**, 166711, (2020).
- 347 2 Dieny, B. *et al.* Opportunities and challenges for spintronics in the microelectronics industry. *Nature Electronics* **3**, 446-459,
348 (2020).
- 349 3 Sinova, J., Valenzuela, S. O., Wunderlich, J., Back, C. H. & Jungwirth, T. Spin Hall effects. *Reviews of Modern Physics* **87**,
350 1213-1260, (2015).
- 351 4 Hirsch, J. E. Spin Hall Effect. *Physical Review Letters* **83**, 1834-1837, (1999).
- 352 5 Manchon, A., Koo, H. C., Nitta, J., Frolov, S. M. & Duine, R. A. New perspectives for Rashba spin-orbit coupling. *Nature*
353 *Materials* **14**, 871-882, (2015).
- 354 6 Edelstein, V. M. Spin polarization of conduction electrons induced by electric current in two-dimensional asymmetric
355 electron systems. *Solid State Communications* **73**, 233-235, (1990).
- 356 7 Lee, S. *et al.* Efficient conversion of orbital Hall current to spin current for spin-orbit torque switching. *Communications*
357 *Physics* **4**, 234, (2021).
- 358 8 Lee, D. *et al.* Orbital torque in magnetic bilayers. *Nature Communications* **12**, 6710, (2021).
- 359 9 Go, D., Jo, D., Lee, H.-W., Kläui, M. & Mokrousov, Y. Orbitronics: Orbital currents in solids. *Europhysics Letters* **135**,
360 37001, (2021).
- 361 10 Zhang, S. & Yang, Z. Intrinsic Spin and Orbital Angular Momentum Hall Effect. *Physical Review Letters* **94**, 066602, (2005).
- 362 11 Bernevig, B. A., Hughes, T. L. & Zhang, S.-C. Orbitronics: The Intrinsic Orbital Current in p-Doped Silicon. *Physical Review*
363 *Letters* **95**, 066601, (2005).
- 364 12 Kittel, C. *Introduction to solid state physics*. 8 edn, (John Wiley & Sons, 2004).
- 365 13 Jo, D., Go, D. & Lee, H.-W. Gigantic intrinsic orbital Hall effects in weakly spin-orbit coupled metals. *Physical Review B*
366 **98**, 214405, (2018).
- 367 14 Kontani, H., Tanaka, T., Hirashima, D. S., Yamada, K. & Inoue, J. Giant Orbital Hall Effect in Transition Metals: Origin of
368 Large Spin and Anomalous Hall Effects. *Physical Review Letters* **102**, 016601, (2009).
- 369 15 Tanaka, T. *et al.* Intrinsic spin Hall effect and orbital Hall effect in 4d and 5d transition metals. *Physical Review B* **77**, 165117,
370 (2008).
- 371 16 Choi, Y.-G. *et al.* Observation of the orbital Hall effect in a light metal Ti. arXiv:2109.14847 (2021).
- 372 17 Sala, G. & Gambardella, P. Giant orbital Hall effect and orbital-to-spin conversion in 3d, 5d, and 4f metallic heterostructures.
373 *Physical Review Research* **4**, 033037, (2022).
- 374 18 Salemi, L. & Oppeneer, P. M. First-principles theory of intrinsic spin and orbital Hall and Nernst effects in metallic
375 monoatomic crystals. *Physical Review Materials* **6**, 095001, (2022).
- 376 19 Go, D. & Lee, H.-W. Orbital torque: Torque generation by orbital current injection. *Physical Review Research* **2**, 013177,
377 (2020).
- 378 20 Go, D., Jo, D., Kim, C. & Lee, H.-W. Intrinsic Spin and Orbital Hall Effects from Orbital Texture. *Physical Review Letters*
379 **121**, 086602, (2018).
- 380 21 Sahu, P., Bhowal, S. & Satpathy, S. Effect of the inversion symmetry breaking on the orbital Hall effect: A model study.
381 *Physical Review B* **103**, 085113, (2021).
- 382 22 Bhowal, S. & Satpathy, S. Intrinsic orbital moment and prediction of a large orbital Hall effect in two-dimensional transition
383 metal dichalcogenides. *Physical Review B* **101**, 121112, (2020).
- 384 23 Cysne, T. P. *et al.* Disentangling Orbital and Valley Hall Effects in Bilayers of Transition Metal Dichalcogenides. *Physical*
385 *Review Letters* **126**, 056601, (2021).
- 386 24 Back, I. & Lee, H.-W. Negative intrinsic orbital Hall effect in group XIV materials. *Physical Review B* **104**, 245204, (2021).
- 387 25 Santos, E. *et al.* Inverse Orbital Torque via Spin-Orbital Entangled States. arXiv:2204.01825 (2022).
- 388 26 Go, D. *et al.* Theory of current-induced angular momentum transfer dynamics in spin-orbit coupled systems. *Physical Review*
389 *Research* **2**, 033401, (2020).
- 390 27 Kimura, T., Otani, Y., Sato, T., Takahashi, S. & Maekawa, S. Room-Temperature Reversible Spin Hall Effect. *Physical*
391 *Review Letters* **98**, 156601, (2007).
- 392 28 Papaioannou, E. T. & Beigang, R. THz spintronic emitters: a review on achievements and future challenges. *Nanophotonics*
393 **10**, 1243-1257, (2021).
- 394 29 Kumar, S. *et al.* Optical damage limit of efficient spintronic THz emitters. *iScience* **24**, 103152, (2021).
- 395 30 Torosyan, G., Keller, S., Scheuer, L., Beigang, R. & Papaioannou, E. T. Optimized Spintronic Terahertz Emitters Based on
396 Epitaxial Grown Fe/Pt Layer Structures. *Scientific Reports* **8**, 1311, (2018).
- 397 31 Yang, D. *et al.* Powerful and Tunable THz Emitters Based on the Fe/Pt Magnetic Heterostructure. *Advanced Optical*
398 *Materials* **4**, 1944-1949, (2016).

- 399 32 Seifert, T. *et al.* Efficient metallic spintronic emitters of ultrabroadband terahertz radiation. *Nature Photonics* **10**, 483, (2016).
- 400 33 Kampfrath, T. *et al.* Terahertz spin current pulses controlled by magnetic heterostructures. *Nature Nanotechnology* **8**, 256,
401 (2013).
- 402 34 Kumar, S., Nivedan, A., Singh, A. & Kumar, S. THz pulses from optically excited Fe-, Pt- and Ta-based spintronic
403 heterostructures. *Pramana* **95**, 75, (2021).
- 404 35 Zhou, C. *et al.* Broadband Terahertz Generation via the Interface Inverse Rashba-Edelstein Effect. *Physical Review Letters*
405 **121**, 086801, (2018).
- 406 36 Jungfleisch, M. B. *et al.* Control of Terahertz Emission by Ultrafast Spin-Charge Current Conversion at Rashba Interfaces.
407 *Physical Review Letters* **120**, 207207, (2018).
- 408 37 Qiu, H. *et al.* Ultrafast spin current generated from an antiferromagnet. *Nature Physics* **17**, 388-394, (2021).
- 409 38 Seifert, T. *et al.* Ultrabroadband single-cycle terahertz pulses with peak fields of 300 kV cm⁻¹ from a metallic spintronic
410 emitter. *Applied Physics Letters* **110**, 252402, (2017).
- 411 39 Hawecker, J. *et al.* Spin Injection Efficiency at Metallic Interfaces Probed by THz Emission Spectroscopy. *Advanced Optical*
412 *Materials* **9**, 2100412, (2021).
- 413 40 Gueckstock, O. *et al.* Terahertz Spin-to-Charge Conversion by Interfacial Skew Scattering in Metallic Bilayers. *Advanced*
414 *Materials* **33**, 2006281, (2021).
- 415 41 Cheng, L., Li, Z., Zhao, D. & Chia, E. E. M. Studying spin-charge conversion using terahertz pulses. *APL Materials* **9**,
416 070902, (2021).
- 417 42 Agarwal, P. *et al.* Ultrafast Photo-Thermal Switching of Terahertz Spin Currents. *Advanced Functional Materials* **31**,
418 2010453, (2021).
- 419 43 Dang, T. H. *et al.* Ultrafast spin-currents and charge conversion at 3d-5d interfaces probed by time-domain terahertz
420 spectroscopy. *Applied Physics Reviews* **7**, 041409, d (2020).
- 421 44 Seifert, T. S. *et al.* Terahertz spectroscopy for all-optical spintronic characterization of the spin-Hall-effect metals Pt, W and
422 Cu₈₀Ir₂₀. *Journal of Physics D: Applied Physics* **51**, 364003, (2018).
- 423 45 Gorchon, J., Mangin, S., Hehn, M. & Malinowski, G. Is terahertz emission a good probe of the spin current attenuation
424 length? *Applied Physics Letters* **121**, 012402, (2022).
- 425 46 Kumar, S. & Kumar, S. Ultrafast light-induced THz switching in exchange-biased Fe/Pt spintronic heterostructure. *Applied*
426 *Physics Letters* **120**, 202403, (2022).
- 427 47 Bull, C. *et al.* Spintronic terahertz emitters: Status and prospects from a materials perspective. *APL Materials* **9**, 090701,
428 (2021).
- 429 48 Xu, Y. *et al.* Inverse Orbital Hall Effect Discovered from Light-Induced Terahertz Emission. arXiv:2208.01866 (2022).
- 430 49 Shi, Z. *et al.* Effect of band filling on anomalous Hall conductivity and magneto-crystalline anisotropy in NiFe epitaxial thin
431 films. *AIP Advances* **6**, 015101, (2016).
- 432 50 Stamm, C., Pontius, N., Kachel, T., Wietstruk, M. & Dürr, H. A. Femtosecond x-ray absorption spectroscopy of spin and
433 orbital angular momentum in photoexcited Ni films during ultrafast demagnetization. *Physical Review B* **81**, 104425, (2010).
- 434 51 Rouzegar, R. *et al.* Laser-induced terahertz spin transport in magnetic nanostructures arises from the same force as ultrafast
435 demagnetization. *Physical Review B* **106**, 144427, (2022).
- 436 52 Stamm, C. *et al.* Femtosecond modification of electron localization and transfer of angular momentum in nickel. *Nature*
437 *Materials* **6**, 740-743, (2007).
- 438 53 Bose, A. *et al.* Detection of long-range orbital-Hall torques. arXiv:2210.02283 (2022).
- 439 54 Hayashi, H. *et al.* Observation of long-range orbital transport and giant orbital torque. arXiv:2202.13896 (2022).
- 440 55 Kumar, S. & Kumar, S. Large interfacial contribution to ultrafast THz emission by inverse spin Hall effect in CoFeB/Ta
441 heterostructure. *iScience* **25**, 104718, (2022).
- 442 56 Sagasta, E. *et al.* Unveiling the mechanisms of the spin Hall effect in Ta. *Physical Review B* **98**, 060410, (2018).
- 443 57 Sagasta, E. *et al.* Tuning the spin Hall effect of Pt from the moderately dirty to the superclean regime. *Physical Review B* **94**,
444 060412, (2016).
- 445 58 Matthiesen, M. *et al.* Temperature dependent inverse spin Hall effect in Co/Pt spintronic emitters. *Applied Physics Letters*
446 **116**, 212405, (2020).
- 447 59 Fert, A. & Levy, P. M. Spin Hall Effect Induced by Resonant Scattering on Impurities in Metals. *Physical Review Letters*
448 **106**, 157208, (2011).
- 449 60 Suess, D., Fidler, J., Zimanyi, G., Schrefl, T. & Visscher, P. Thermal stability of graded exchange spring media under the
450 influence of external fields. *Applied Physics Letters* **92**, 173111, (2008).
- 451 61 Li, Y. *et al.* Enhancing the Spin-Orbit Torque Efficiency by the Insertion of a Sub-nanometer β -W Layer. *ACS Nano* **16**,
452 11852-11861, (2022).

453 62 Lu, Q. *et al.* Enhancement of the Spin-Mixing Conductance in CoFeB/W Bilayers by Interface Engineering. *Physical Review Applied* **12**, 064035, (2019).
454
455 63 Wahada, M. A. *et al.* Atomic Scale Control of Spin Current Transmission at Interfaces. *Nano Letters* **22**, 3539-3544, (2022).
456 64 Hait, S. *et al.* Spin Pumping through Different Spin–Orbit Coupling Interfaces in β -W/Interlayer/Co₂FeAl Heterostructures. *ACS Applied Materials & Interfaces* **14**, 37182-37191, (2022).
457 65 Seifert, T. S. *et al.* Time-domain observation of ballistic orbital-angular-momentum currents with giant relaxation length in tungsten. arXiv:2301.00747 (2023).
458 66 Liu, F., Liang, B., Xu, J., Jia, C. & Jiang, C. Giant efficiency of long-range orbital torque in Co/Nb bilayers. arXiv:2211.04809 (2022).
460 67 McHugh, O. L. W., Goh, W. F., Gradhand, M. & Stewart, D. A. Impact of impurities on the spin Hall conductivity in beta-W. *Physical Review Materials* **4**, 094404, (2020).
462 68 Kumar, A., Bansal, R., Chaudhary, S. & Muduli, P. K. Large spin current generation by the spin Hall effect in mixed crystalline phase Ta thin films. *Physical Review B* **98**, 104403, (2018).
464
465

466 **ACKNOWLEDGMENTS**

467 SK acknowledges the Science and Engineering Research Board (SERB), Department of Science and Technology, Government of India, for financial support through project no. CRG/2020/000892, Joint Advanced Technology Center, IIT Delhi, is also
468 acknowledged for support through EMDTERA#5 project. The authors acknowledge CRF, IIT Delhi for SIMS facilities. One of
469 the authors (Sandeep Kumar) acknowledges the University Grants Commission, Government of India, for Senior Research
470 Fellowship.
471

472 **AUTHOR CONTRIBUTIONS**

473 Sunil Kumar supervised the work. Sandeep Kumar performed the experiments. Sunil Kumar and Sandeep Kumar analyzed the
474 data and wrote the manuscript.

475 **COMPETING INTERESTS**

476 The authors declare no conflict of interests.

477 **ADDITIONAL INFORMATION**

478 Supporting Information is available.
479

Response Letter

Manuscript ID: NCOMMS-23-03062

We thank all the referees for critically assessing our manuscript and providing us with many important comments, which have helped us to improve the discussion in our paper. We appreciate the time and consideration taken by all the referees. Below, we have provided our detailed point-by-point replies to all the queries and comments raised by the referees. In some cases, more experiments have been performed by us to validate various points in our manuscript, which were also critically evaluated by the referees.

The modifications in the revised manuscript (and the Supplementary information) are indicated in the red color.

Reviewer #1

Remarks by the reviewer: Orbitronics exploits the potential of the orbital degree of freedom for information technology. Recently this topic has received increasing attention from the community. In this manuscript, the authors studied the THz generation from orbital currents in ferromagnet/nonmagnet heterostructures, including NiFe/Nb, CoFeB/Pt, CoFeB/Ta, and CoFeB/W/Ta. They concluded that the THz emission is driven by orbital effects for NiFe/Nb and CoFeB/W/Ta. I think the arguments and reasoning presented for NiFe-based samples are not solid, undermining their conclusion's validity. Moreover, it is very likely that the THz emission from CoFeB/W/Ta arises from the inverse spin Hall effect instead of any orbital effects, as claimed by the authors.

Authors' response: We thank the reviewer for finding out time and providing critical assessment to our manuscript. We have now taken care of all the issues pointed out by the referee in his/her specific comments as below. This has indeed helped us to improve the presentation of our manuscript. To validate our conclusions more emphatically, we have conducted additional experiments, which are now incorporated in our revised manuscript. We may mention the following important points regarding the revision done by us.

1. We have provided new experimental results on CoFeB/W having different thicknesses of the W layer to further strengthen our claim on the orbital current as the main source of enhancement in the THz signal from the CoFeB/W/Ta heterostructure.
2. Additional experiments are performed on single FM layer samples of NiFe and CoFeB to rule out AHE. More experiments are done on the bilayer NiFe/Nb with varying thickness of Nb layer to reiterate our original findings and conclude on the orbital diffusion length in this set of

experiments. In fact, this particular observation will be a new value addition to the existing literature.

3. The normal optical transmission through the substrate inside the optical cryostat has been checked with the temperature. We find negligible change and hence no effect on any of the experimental data and conclusions in our manuscript.

4. Other than the new experimental results, we have repeated some of the experiments on NiFe/Nb, CoFeB/W/Ta, CoFeB/Ta, etc. to provide a fresh comparison to new results by keeping the similar experimental conditions.

Thus, we strongly believe that our manuscript is highly suitable for publication in Nature Communications.

Our detailed replies to the specific comments by the referee are given below.

Comment#1: The terahertz emission from anomalous Hall effect has been reported in FeMnPt by Zhang et al. (Phys. Rev. Applied 12, 054027 (2019).) and more recently in CoFeB by Liu et al. (Physical Review B 104, 064419 (2021).) and Mottamchetty et al. (arXiv:2302.07398). A net charge current perpendicular to the film can also be excited by femtosecond laser excitation. Due to the anomalous Hall effect, the net charge current is converted into a transverse transient charge current, generating THz radiation. Judging from the low signal-to-noise ratio, the signals from NiFe/Nb in this manuscript appear quite weak. The weak emission probably just comes from the anomalous Hall effect (AHE) of NiFe layers. Such a mechanism needs to be considered or excluded carefully. Otherwise, all the remaining discussions will be unreliable for readers.

Authors' reply: The referee's concern is valid since we have not provided any experimental confirmation and related discussion demonstrating that the AHE is not an active participant in our THz analysis. We agree to the referee's point that AHE can help convert any longitudinal transient current induced by the femtosecond excitation to a transverse charge current for further generation of THz radiation. However, this was not the case in our experiments and hence we ignored it initially. We thank the referee for raising this important point that helped us improve our discussion in the paper with better clarity and avoid any confusion in the reader's mind about the role of AHE. Below, we demonstrate that AHE is not a player in the THz emission process in our case.

Figure R1

We performed THz emission measurements for front and back side optical illumination, the results from the same are shown in Figure R1. The terms, back side and front side are designated for the optical pumping from the substrate side and film side, respectively. These were done on bare $\text{Ni}_{0.9}\text{Fe}_{0.1}$ and CoFeB FM layers as shown in Figure R1(a) and R1(c), respectively. In Fig. R1(b), the THz signal from bare $\text{Ni}_{0.9}\text{Fe}_{0.1}$ is compared with bilayer NiFe(5nm)/Nb(10nm) bilayer sample. The maximum of all the THz waveforms is set to zero for comparisons. It is evident from the results in Figures R1(a) and R1(c) that the THz signal polarity remains the same irrespective of the sample excitation geometry.

The THz emission from a single FM layer has been demonstrated in various reports, {*Q. Zhang et al., Physical Review Applied, 12, 054027, (2019)*; *L. Huang et al., Scientific Reports, 10, 15843, (2020)*; *Y. Liu et al., Physical Review B, 104, 064419, (2021)*; *V. Mottamchetty et al., arXiv:2302.07398, (2023)*} and the origin of the THz emission is attributed mainly to the anomalous Hall effect (AHE) {*Q. Zhang et al., Physical Review Applied, 12, 054027, (2019)*; *Y. Liu et al., Physical Review B, 104, 064419, (2021)*; *V. Mottamchetty et al., arXiv:2302.07398, (2023)*} and ultrafast demagnetization {*E. Beaurepaire et al., Physical Review Letters, 76, 4250-4253, (1996)*; *E. Beaurepaire et al., Applied Physics Letters, 84, 3465-3467, (2004)*; *W. Zhang et al., Nature Communications, 11, 4247, (2020)*} (UDM) processes. While AHE mediated THz emission is a result of net backflow current and a combined effect of both interface as well as bulk, the UDM induced THz emission process relies only on the bulk properties of the material layer. Both the effects can be distinguished qualitatively by analyzing THz polarity behavior with respect to the sample excitation geometry. In case of the THz emission from AHE, the polarity of the THz waveform gets reversed when the sample is flipped. This is due to the change in the direction of the net backflow current. However, in the case of UDM, the waveform polarities remain the same

upon sample flipping due to the fact that the magnetization dynamics is insensitive to the excitation direction of the sample.

The UDM scenario is well aligned with our THz emission measurements performed on bare NiFe and CoFeB sample as shown in Figures R1(a) and R1(c), respectively. The consistency of our observation with the UDM, ruled out the possibilities of AHE, {*Q. Zhang et al., Physical Review Applied, 12, 054027, (2019); Y. Liu et al., Physical Review B, 104, 064419, (2021); V. Mottamchetty et al., arXiv:2302.07398, (2023)*} mechanisms in the THz radiation from our NiFe(5nm)/Nb(10nm) sample. We have also compared the THz signal from bare NiFe with the one from NiFe/Nb bilayer in Figure R1(b). A huge difference between the two signal amplitudes further suggests the weak THz signal emitted from bare NiFe is magnetic dipole radiation due to UDM. The outcomes similar to bare NiFe were also observed in the case of bare CoFeB sample (see Fig. R1(c)), which further strengthens the fact that the THz emission is not due to AHE. Therefore, in consistency with the discussion in the above, we have demonstrated through the results in Figure R1 that AHE can be safely ruled out for any role to the THz generation in our case.

To satisfy the concern by the reviewer and to bring more clarity for the readers, we have added the above figure and corresponding discussion in Section S12 of the Supplementary information of our revised manuscript.

Moreover, the following text has been added in the revision of the main manuscript.

On page 2 of the manuscript: "...whereas, the same for bare NiFe FM layer is presented in the Supplementary Section S12."

On page 3 of the manuscript: "If the THz signal from the NiFe/Nb bilayer structure is a result of the inverse spin Hall effect (ISHE), its polarity would be expected to be the same as that of the Fe/Ta bilayer but opposite to that of the CoFeB/Pt bilayer structures. This distinction arises because Nb and Ta both have the negative sign of θ_{SH}^{Ta} , while Pt has a positive sign of θ_{SH}^{Pt} . However, the observed polarity of the THz signals in the NiFe/Nb bilayer structure opposite to that of Fe/Ta bilayer structure but coincides with the CoFeB/Pt bilayer structure, as schematically depicted in Figs. 1(a), 3(a), and 3(b), respectively. In fact, the polarity of the THz signals emitted from NiFe/Nb bilayer structure aligns with the sign of $\theta_{OH}^{Nb} \cdot \eta_{L-S}^{NiFe}$ implying that the THz emission from NiFe/Nb takes place via IOHE, mainly."

On page 6 of the manuscript: “Moreover, ultrafast demagnetization mechanism is mainly responsible for the THz emission from the bare FM layer (see Supplementary Section S12).”

Comment#2: The authors studied the temperature dependence of THz emission and temperature dependence of Hall resistivity. The authors argued that the spin Hall resistivity is proportional to the terahertz electric field according to Eqn 5. A similar equation has been used to interpret the terahertz emission from the inverse spin Hall effect, but a simple extension to the case of orbital current like Eqn 5 is much too risky. In the derivation of Eqn 5, it was implicitly assumed the orbital diffusion length is the same as the spin diffusion length, which is not supported by recent observations (Commun Phys 6, 1 (2023)). It has been well established that the diffusion length orbital current is more than one order of magnitude higher than that of the spin current. In fact, it was argued that the orbital angular momentum and spin angular momentum are fundamentally different (arXiv:2106.07928). The derivation of Eqn 5 needs to be revised. In the manuscript, the fittings using Eqn 5 were not convincing enough.

Authors' reply: Thank you for bringing up this concern. We agree with the reviewer's assessment that while the orbital and spin currents exhibit distinct transport properties, their diffusion length is the key parameter that highlights these differences. As pointed out by the referee also, use of Eq. 5, even for the orbital picture, is valid only with the assumption that the spin and orbital diffusion lengths are of the same order. In the literature, most of the studies point nearly an order higher orbital diffusion length (OD) than the spin diffusion length (SD). We may argue that such an observation of the inequality between the two cannot be generalized for all classes of materials. Almost in all of the previous reports on this fact of nearly an order higher orbital diffusion length, one has dealt with heavy metals (Pt{*E. Santos et al., Physical Review Applied, 19, 014069, (2023)*}, W{*H. Hayashi et al., Communications Physics, 6, 32, (2023)*; *T. S. Seifert et al., arXiv:2301.00747, (2023)*}) and magnetic metals (Cr{*S. Lee et al., Communications Physics, 4, 234, (2021)*}, Ni{*A. Bose et al., arXiv:2210.02283, (2022)*}). For heavy metals, their strong spin-orbit coupling can suppress the spin diffusion length, and thus this inequality may hold. For magnetic metals, their magnetism can suppress the spin diffusion length. But in nonmagnetic light metals, like Nb, neither spin-orbit coupling, nor magnetism can strongly suppress the spin diffusion length.

In a previous experimental study, the spin diffusion length in Nb was reported to be of ~30 nm at the room temperature. {*K.-R. Jeon et al., Physical Review Applied, 10, 014029, (2018)*} However, in a recent study, a relatively smaller value of orbital diffusion length for Nb in the Co/Nb sample, has been reported {*F. Liu et al., Physical Review B, 107, 054404, (2023)*} to be just ~3 nm. This contradiction is a clear case that orbital diffusion length cannot always be assumed

larger than the spin diffusion length, as inferred from the comment by the referee. Hence, the difference between the spin and orbital diffusion lengths is still to be debated and more studies to be carried out.

To address the above issue, here we present new results on the THz emission from NiFe/Nb samples to experimentally determine the orbital diffusion length for our case. Figure R2(a) presents the raw data for different NiFe/Nb bilayer heterostructures and explicit Nb-thickness dependence of the THz signal magnitude normalized by optical absorptance, is shown in Figure R2(b). For these measurements, the NiFe layer thickness was kept same at 5 nm for all. We can see that while the signal polarity is same in all the cases, the magnitude of the emitted THz signal reduces decreases strongly for higher thicknesses of the Nb-layer in NiFe/Nb. The observation on similar THz polarity is consistent with IOHE scenario presented in our main manuscript for NiFe(5nm)/Nb(10nm) sample, we believe that our new result on the Nb-thickness dependence (Fig. R2(b)) will be a new value addition to the field. From Fig. R2(b), we conclude that the orbital diffusion length in Nb is of ~ 25 nm at room temperature, that is in the same order as the reported value of spin diffusion length. It may be noted that a similar conclusion is also drawn at low temperature. With the above arguments, reasonably supported by our new results, we have justified the feasibility of Eq. (5) in our case. Moreover, a small difference between the spin diffusion length and the orbital diffusion length would not affect the qualitative analysis and our main conclusions in the manuscript.

Figure R2

Based on the above explanations, we have added the following text in our revised manuscript and Supporting Information.

- Added a line “Typically, the orbital diffusion length in Nb is comparable to that of spin diffusion length (see Supplementary Section S13). Due to the negligible SOC and subsequently smaller spin Hall angle in Nb,....” by rearranging the text in paragraph 1 of page 3 of our revised manuscript.
- Added a new section S13 to our revised supplementary information.
- On page 7 of the Supporting Information: “Equation S7 is valid for the orbital picture under the assumption that the spin and orbital diffusion length are of the same order. However, this is indeed the case for Nb, as discussed in Section S13.”

Comment#3: There is a systematic temperature difference for the two experiments, i.e., laser-induced THz emission and Hall measurement. For the experiment of the temperature dependence of THz emission, it is well known that the laser results in an average temperature increase due to the absorption of laser power. The average temperature is higher than the environmental temperature of the cryostat. But for the experiment on Hall resistivity, the temperature is expected to be close to the environmental temperature. I wonder how the temperatures are calibrated and treated in the two experiments.

Authors' reply: Thank you for bringing up the issue and we agree with the reviewer’s concern on the laser heating induced average temperature rise in the system. In the below we have elaborated on how it is not a concern in our experiments.

From the well-known thermodynamic relation {*J. Khachan, Thermal Properties of Matter, 4-1-4-8, (2018)*} between the specific heat (C) and temperature difference (ΔT), the amount of heat (Q) per unit mass (m) that needs to be added in order to raise its average temperature by 1 K is given by $Q = Cm\Delta T$. This can be rewritten in terms of the laser-fluence (Φ) and optical absorbance as $\Delta T = Q/Cm = A\Phi S/Cm$. Here, A and S represent the optical absorbance and the area of excitation laser beam spot, respectively. Now, considering the parameters used in our measurements, i.e., A for the thin film to be of ~ 0.5 , laser spot diameter = 3 mm, $\Phi = 0.5 \text{ mJ/cm}^2$, and values of the other parameters, calculated separately, i.e., specific heat capacity and mass (both are intensive thermodynamical variables) of the FM layer to be $C_{\text{NiFe}} = 0.52 \text{ J/g.K}$, $m_{\text{NiFe}} = 4.4 \mu\text{g}$, and those of the NM layer to be $C_{\text{Nb}} = 0.26 \text{ J/g.K}$, $m_{\text{Nb}} = 8.57 \mu\text{g}$, and the quartz substrate to be $C_{\text{Qz}} = 741 \text{ J/g.K}$, $m_{\text{Qz}} = 0.26 \mu\text{g}$, we obtain an average temperature rise ΔT of $\sim 2 \text{ K}$ for our NiFe(5)/Nb(10) sample and negligibly small value for Qz substrate. Therefore, we have omitted the effect of laser induced heating in our temperature dependent measurements and it is safe to use common temperature values for both the THz emission and the Hall resistivity.

Moreover, the laser induced heating of the sample can make a difference if the absorption due to which temperature rise occurs is changing with the cryostat temperature. Therefore, we measured the optical transmission through the cryostat having the sample substrate mounted in the cold finger, at all of the cryostat temperatures and the results is presented below in Fig. R3. The transmitted intensity is measured using a low power meter. It can be seen that in the entire temperature range, there is hardly any variation in the sample absorption.

Figure R3

Comment#4: As mentioned above, the THz emission might be from AHE, which also depends on the temperature. Thus, the temperature dependence of AHE will modify the temperature dependence of THz emission. The authors should consider the temperature dependence of AHE in the related discussion about Figures 2, 3, and 4.

Authors' reply: We agree with the reviewer's point that the presence of AHE and its dependance on sample temperature may have an impact on the THz emission behavior in those cases where AHE is the main mechanism for THz emission. As described in our response to comment #1 by the referee, AHE is not present in our case and hence the situation pointed out by the referee does not arise. The THz signal solely due to ultrafast demagnetization in the FM layer is comparatively much weaker and hence temperature dependence in its part will not affect our conclusions, where the ISHE and IOHE induced THz emission is one or more orders stronger. Moreover, we have shown in Section S4 of the Supplementary Information of our manuscript that the sample magnetization is nearly constant in the entire experimental temperature window. Therefore, the temperature dependent change in the UDM mediated THz emission is expected to be negligible.

Comment#5: To derive Eqn 5, it was assumed that the spin current stayed the same for all the temperatures. Matthiesen et al. (Appl. Phys. Lett. 116, 212405 (2020).) measured the temperature dependence of THz emission. When the Curie temperature of Co was much higher than the

temperature under consideration, the demagnetization and spin current can be taken as constant below room temperature. However, the situation for NiFe is different because the Curie temperature of NiFe is much lower than Co. Furthermore, the Curie temperature decreases when the thickness of NiFe is reduced. In this case, one must pay attention to the temperature dependence of the spin current below room temperature.

Authors' reply: Thank you for the comment. We agree with the referee's point that care must be taken when the Curie temperature is low. The Curie temperature of NiFe thin films depends on the specific composition of the alloy, structural phase, and thickness of the respective film, as well as any other factors that might affect its magnetic properties. {*L. F. Yin et al., Physical Review Letters, 97, 067203, (2006)*; *P. Yu et al., Physical Review B, 77, 054431, (2008)*; *X. Zhang et al., APL Materials, 7, 111112, (2019)*} It has been seen in experiments that the Curie temperature of NiFe alloys can vary in the range from ~400 to ~900 K. NiFe alloys with higher nickel content tend to have higher Curie temperatures, while those with higher iron content have lower Curie temperatures.

However, the Curie temperature of Ni_{0.9}Fe_{0.1} thin film sample is higher than the room temperature in our case. We have performed magnetization versus temperature measurements on Ni₉₀Fe₁₀/Nb heterostructures which is shown in Figure S4(c) of the Supplementary Information of our manuscript. The variation in the sample magnetization is almost insensitive in the given range of temperature, which suggests that the magnetization transition point (or Curie point) is far from our experimental temperature window.

From the above discussion, we have added a line, “.....as confirmed by M-T measurements shown in Section S4.” to the supporting information Section S10 in our revised manuscript.

Comment#6: After inserting a W layer, the terahertz emission increased significantly. The authors attribute the enhancement of THz emission to the generation of orbital current in the W layer. A big assumption here is that the W layer would convert spin current to orbit current. In literature, Gd or Pt was used to convert orbit current to spin current, but not W. As it has not been reported before, the role of W needs to be proved more rigorously. The spin Hall angle of W is much greater than that of Ta, and thus the THz emission from W/FM is much stronger than Ta/FM. For example, Wu et al. have shown that the THz emission from W/Co is about ten times higher than that from Ta/Co (*Advanced Materials* 29, 1603031 (2017).). The authors assumed that the spin currents are the same with and without the insertion W layer, but the strength of the spin current that flows into the nonmagnetic layers depends on the thickness of the nonmagnetic layer. The calculation by Torosyan et al. (*Scientific Reports* 8, 1311 (2018).) shows that if the total thickness of the nonmagnetic layer is less than the spin diffusion length, the spin current will increase as the thickness of the nonmagnetic layer increases. After inserting a W layer, both the spin Hall angle and the spin current will increase, leading to a significant enhancement of terahertz emission. The

orbital effects are likely to be irrelevant for CoFeB/W/Ta, and the enhancement can be explained using the inverse spin Hall effect only. I think the conclusion drawn by the authors does not hold.

Authors' reply: Thank you for your valuable feedback. We appreciate your concern regarding the role of the W interlayer in the enhancement of THz emission from CoFeB/Ta heterostructures. We agree that a comparison between CoFeB/W, CoFeB/Ta bilayers, and CoFeB/W/Ta trilayer would be helpful in clarifying the role of the insertion layer. We have performed additional experiments on CoFeB/W bilayer system to address this concern.

Before getting to our detailed reply, we would like to bring to the attention of the referee that another study on arXiv {*P. Wang et al., arXiv:2305.05830, (2023)*}, that has appeared a couple of days ago, also presents role of W-insertion layer in a Co/W/Ti and Co/W/Mn trilayer heterostructures. The introduction of a W layer has been found to significantly enhance (more than an order) the orbitronic terahertz emission. This enhancement is attributed to the generation of additional large orbital current within the W-insertion layer.

Figure R4 presents our experimental results on CoFeB/W bilayer. The results on the CoFeB/Ta and CoFeB/W/Ta are already included in our original manuscript. From Fig. R4, following observations are made: (i) THz signal emitted from CoFeB/W(2nm) bilayer sample is ~ 2.5 times higher than that from CoFeB/Ta(2nm) whereas their polarities are same. (ii) THz signal from CoFeB/W bilayer decreases if the W layer thickness is changed from 2 to 3 nm. (iii) CoFeB/W(2)/Ta(2) sample emits ~ 10 and ~ 4 times higher THz signal as compared to that from CoFeB/Ta(2) and CoFeB/W(2) samples, respectively. These observations clearly suggest a different origin than ISHE for the THz emission here.

Figure R4

Based on the above observations, the following points are made.

- (1) Analogous to CoFeB/Ta(2nm) bilayer, ISHE is the origin of THz emission from CoFeB/W(2nm). The similar polarity and different THz amplitudes are quite consistent with the sign and magnitude of the spin Hall angles in W and Ta.
- (2) If the THz emission in the CoFeB/W(2)/Ta(2) trilayer was solely due to ISHE, one would expect THz signal amplitude to be nearly equal to the sum of the THz signals emitted from the individual bilayer counterparts, namely CoFeB/Ta(2) and CoFeB/W(2). On the other hand, we measure THz signal from the CoFeB/W(2)/Ta(2) to be nearly three times stronger than the total sum of the signals from the two layers individually. Therefore, we conclude that THz generation from the CoFeB/W(2)/Ta(2) is not solely due to ISHE.
- (3) In the case of CoFeB/W bilayer, THz signal strength decreases while increasing the W-layer thickness from 2 to 3 nm. This clearly indicates that the spin diffusion length in the W-layer, and hence its optimum thickness for the THz emission via ISHE, is about 2-3 nm. Such a value of the spin diffusion length in W-layer matches well with the literature. {*T.-C. Wang et al., Physical Review Materials, 2, 014403, (2018)*}
- (4) Continuing with point 3 above, the presence of such a thick W-insertion layer ($W = 2\text{nm}$) would have significantly reduced the spin current being transmitted to the adjacent Ta layer in the CoFeB/W/Ta trilayer structure. Therefore, complete spin to charge conversion within the W-layer or in parts within the W-layer and the Ta-layer of the heterostructure, one cannot explain the nearly 10 times [4 times] THz signal enhancement in CoFeB/W(2)/Ta(2) as compared with the CoFeB/Ta(2) [CoFeB/W(2)] from the ISHE as the source. Additionally, enhancement in the spin current transparency due to an insertion layer is only possible if a very small thickness of it, just a few Angstroms, {*Q. Lu et al., Physical Review Applied, 12, 064035, (2019)*; *Y. Li et al., ACS Nano, 16, 11852-11861, (2022)*} is used. For thick (a few nm) insertion layers, like the case in our experiments, the spin transparency reduces significantly. {*J. Hawecker et al., Advanced Optical Materials, 9, 2100412, (2021)*}

From the above points, we therefore conclude that THz enhancement from CoFeB/W(2)/Ta(2) as compared to the CoFeB/W(2) and CoFeB/Ta(2) is not due to either an increased spin Hall angle or the spin transparency as required for the ISHE to play the dominant role in the THz emission.

Hence, the role of the W-insertion layer as an orbital transport channel rather than simply as an ISHE interface was considered, which explains our observations in the manuscript.

As pointed out by the referee that Gd and Pt were used before as the insertion layer, we agree that W is relatively new and less explored material in that regard. Although W is a relatively new candidate as an interlayer material, its large orbital Hall angle {*H. Hayashi et al., Communications Physics, 6, 32, (2023)*} and comparable spin Hall angle {*T.-C. Wang et al., Physical Review Materials, 2, 014403, (2018)*; *H. Hayashi et al., Communications Physics, 6, 32, (2023)*} to that of Pt, {*T. Fache et al., Physical Review B, 102, 064425, (2020)*} make it a highly promising material for further study in this field other than the already reported materials {*S. Lee et al., Communications Physics, 4, 234, (2021)*; *G. Sala et al., Physical Review Research, 4, 033037, (2022)*} such as, Pt, Gd, Tb. In fact, in a recent report, {*P. Wang et al., arXiv:2305.05830, (2023)*} the presence of W interlayer has been shown to provide an additional conversion of the orbital-charge current in the Ti and Mn layers, significantly enhancing the orbitronic terahertz emission.

From the above clarifications and discussion, we have added the following in our revised manuscript.

- We have added a line in para 1 of page 8: “ While the THz amplitude for $t = 2$ nm is ~ 10 times stronger than that from $t = 0$ nm, the same is nearly 4 times higher than from the CoFeB(2)/W(2) bilayer (see Supplementary Section S14)..”
- Figure R4 and related discussion is added as a new section S14 in the supplementary information of our revised manuscript.

Reviewer #2

Remarks by the Reviewer: This work by Kumar and Kumar examines the nonlocal orbital current in bilayer and trilayer heterostructures by inducing femtosecond photoexcitations and measuring the resulting THz emission pulses. This work investigates three classes of systems; NiFe/Nb, FM(CoFeB,Fe)/NM(Pt,Ta), and CoFeB/W/Ta. For the first system, NiFe/Nb, the authors concluded that the emitted THz pulses are governed by orbital-to-charge conversion (or IOHE). For the second system, FM(CoFeB,Fe)/NM(Pt,Ta), on the other hand, they concluded that the emitted THz pulses are governed by spin-to-charge conversion (or ISHE). For the third system, CoFeB/W/Ta, they found an order of magnitude enhancement of the THz emission and attributed it to the IOHE due to the emergence of long-range orbital current within the W-insertion layer. I have comments and questions on the following technical aspects.

Authors' response: We thank the referee for his/her valuable time going through the manuscript and providing us the feedback for its improvement. Below, we have provided point-by-point replies to all the comments by the referee. To support the conclusions in the manuscript, we have carried out more experiments and the same are included in our responses below.

Comment#1: The basic idea of this work resembles that of Xu et al. [48], although the precise material systems are different. I suggest the authors clarify the difference from this earlier work.

Authors' reply: We thank the referee for his/her comment and for providing us with the opportunity to clarify the unique aspects of our work. However, before delving into the specific details, we would like to bring to attention of the referee, an another study that has appeared on arXiv {*P. Wang et al., arXiv:2305.05830, (2023)*}, just a couple of days ago. The main point in this manuscript is also a many-fold enhancement of THz emission due to W-insertion layer in FM/W/NM trilayers as compared to the FM/NM counterparts, though the FM and NM layers in that study consist of different materials. Undoubtedly, our manuscript should be considered first to report such an observation. Furthermore, our is the first study which brings out systematic temperature dependence from the highly challenging low-temperature experiments at THz frequencies. Also, the NiFe/Nb system that is studied in our manuscript is a suitable system for exploring orbital transport phenomena.

We may like to add a point here that the study by Xu et al., first appeared on arXiv, when we were about to submit our manuscript to the journal. Since it was the beginning of this emerging concept in the field of THz spectroscopy, everyone who started work in this area coincidentally had a similar type of idea. The work by Xu et al., [48] is possibly the first study, where the authors have demonstrated the role of inverse orbital Hall effect in the THz emission from selectively

chosen FM/NM heterostructures at room temperature. However, along with the difference in the material systems, our approach to confirm orbital transport is quite different. Specifically, our work focuses on temperature-dependent THz emission studies, which create unique challenges and opportunities that were not addressed in earlier works. In summary, we believe that the differences in the material systems and approach used in our work are significant and contribute to the advancement of such an emerging field.

Comment#2: The analysis of the signal in the bilayer, NiFe/Nb, assumes that the orbital diffusion length in Nb is much larger than the spin diffusion length. But I am not sure whether this inequality holds for the nonmagnetic light metal Nb. The previous reports on the inequality dealt with heavy metals (Pt [25], W [65]) and magnetic metals (Cr [7], Ni [53]). For heavy metals, their strong spin-orbit coupling can suppress the spin diffusion length, and thus this inequality may hold. For magnetic metals, their magnetism can suppress the spin diffusion length. But for nonmagnetic light metal Nb, neither spin-orbit coupling nor magnetism can strongly suppress the spin diffusion length. Thus, whether this inequality holds is not obvious. Moreover, considering that a few previous works reported long spin diffusion length for Nb, I recommend the authors to verify the inequality explicitly. The previous experiment reported the following numbers for the spin diffusion length of Nb; 780 nm at 9 K (slightly above the superconducting transition temperature of Nb) [Appl. Phys. Lett. 65, 1460 (1994)] and 30 nm at room temperature [Phys. Rev. Applied 10, 014029 (2018)]. The previous calculation [Physics of the Solid State 51, 2211 (2009)] for Nb reported the spin relaxation length of 160 nm at the electron excitation energy 0.3 eV and 30 nm at the excitation energy 0.9 eV.

Authors' reply: Thank you for the comment. We agree with the referee's point that the inequality in the spin and orbital diffusion lengths may not be necessarily the same in all light metals, unlike the heavy metals in which case, they always possess a larger orbital diffusion length than the spin diffusion length. In case of light metals, this inequity seems to hold for of light metal titanium (Ti), where a relatively larger orbital diffusion length has been reported compared to its spin diffusion length. It is a widely used light metal for realizing orbital effects against the suppressed spin effects as it possesses a weak spin orbit coupling and about two order higher positive orbital Hall conductivity than its negative spin Hall conductivity {*Y.-G. Choi et al., arXiv:2109.14847, (2021); L. Salemi et al., Physical Review Materials, 6, 095001, (2022)*}. The experimentally reported values of the orbital diffusion length {*Y.-G. Choi et al., arXiv:2109.14847, (2021); H. Hayashi et al., Communications Physics, 6, 32, (2023)*} (~40-70 nm at the room temperature) is nearly 4 times larger than the spin diffusion length {*C. Du et al., Physical Review B, 90, 140407, (2014); H. Hayashi et al., Communications Physics, 6, 32, (2023)*} (~13 nm at the room temperature), thus holds the inequality. Like Ti, Nb also is a weak spin-orbit coupling light metal that possesses nearly 2 orders higher positive orbital Hall conductivity than the negative spin Hall conductivity {*T. Tanaka et*

al., Physical Review B, 77, 165117, (2008); L. Salemi et al., Physical Review Materials, 6, 095001, (2022). However, the inequality in the orbital and spin diffusion lengths is not the same. The spin diffusion length in Nb has been reported *{Phys. Rev. Applied 10, 014029 (2018)}* to be of ~ 30 nm at higher than the superconducting temperatures in it. On the other hand, the orbital diffusion length in it is just a few nm only *{F. Liu et al., Physical Review B, 107, 054404, (2023)}*. Therefore, the difference between the orbital and the spin diffusion lengths in all light metals do not necessarily follow the same inequality, but rather, it depends on the materials, combinations, interfaces, etc., and hence demands an explicit verification.

We would like to emphasize that our time domain THz emission spectroscopy helps to provide an estimation of the orbital diffusion length in Nb layer of the NiFe/Nb heterostructure. As we have established in our original manuscript from the analysis of THz amplitude, polarity, and their temperature dependency that IOHE is the source of THz emission from NiFe/Nb, below we provide results from additional experiments that we have carried out for the Nb-layer thickness dependence. Such a measurement helps to provide direct estimation of the orbital diffusion length in Nb. Figure R5(a) presents the raw data for different NiFe/Nb bilayer heterostructures and explicit Nb-thickness dependence of the THz signal magnitude, normalized with respect to optical absorptance, is shown in Figure R5(b). For these measurements, the NiFe layer thickness was kept same as it was in original manuscript, i.e., 5 nm. We can see that while the signal polarity is same for all the samples, the magnitude of the emitted THz signal decreases strongly for higher thicknesses. The observation of the THz polarity being invariant with the increasing Nb-thickness, is consistent with our proposal of IOHE induced THz emission from NiFe(5)/Nb(10) in the original manuscript. We believe that these new results on the Nb-thickness dependence (Fig. R5(b)) will be a great value addition to the field. From Fig. R5(b), one can conclude that the orbital diffusion length in Nb is of ~ 25 nm. It may be noted that a similar conclusion is also drawn at low temperature.

Figure R5

Based on the above discussion, we have made the following changes in our revised manuscript and supporting information.

- Added a line “Typically, the orbital diffusion length in Nb is comparable to that of spin diffusion length (see Supplementary Section S13). Due to the negligible SOC and subsequently smaller spin Hall angle in Nb,...” by rearranging the text in paragraph 1 of page 3 of our revised manuscript.
- Added a new section S13 to our revised supplementary information.
- On page 7 of the Supporting Information: “Equation S7 is valid for the orbital picture under the assumption that the spin and orbital diffusion length are of the same order. However, this is indeed the case for Nb, as discussed in Section S13.”

Comment#3: In Sec. 2.1, it was mentioned that the ultrafast excitation transiently enhances the spin-orbit coupling based on Refs. [50,52]. However, these two experiments [50,52] used the soft x-ray, whereas the present experiment uses near-infrared pulses for excitation. Considering that near-infrared pulses have much smaller excitation energy than the soft x-ray, it is not clear whether the ultrafast excitation transiently enhances the spin-orbit coupling.

Authors' reply: Thank you for your insightful comment. Soft x-ray femtosecond pulse excitation induced transient changes (about a few percent) in the SOC is subject to not only the electron excitation energy but also the interplay among various other factors, {*C. Stamm et al., Nature Materials, 6, 740-743, (2007)*; *C. Boeglin et al., Nature, 465, 458-461, (2010)*; *C. Stamm et al., Physical Review B, 81, 104425, (2010)*} including, the excitation intensity, pulse duration, nature of material, etc. Moreover, it is important to note that the observed phenomenon of transient SOC enhancement for soft X-rays does not necessarily guarantee its occurrence for low-energy NIR pulses as well, unless supported by experimental evidence. However, direct evidence to this fact

is difficult for us to provide in our experiments. Therefore, we have omitted the related statement from our discussion in the revised manuscript. The corresponding sentence in paragraph 3 of page 2 in the revised manuscript has been changed as following.

“Therefore, an ultrafast optically induced orbital current sets in,^{48,52} which possess similar symmetry properties to the spin current but can exhibit relatively different transport dynamics.^{9,53} Furthermore, as the ultrafast excitation of spin and orbital magnetization has been reported^{50,54-56} to exhibit a similar evolution, the emergence of orbital current can be comprehended through the analogy with the already established spin current formation.⁵² ”

Comment#4: In Sec. 2.3, “the nearly one order of increase in the THz generation efficiency of CoFeB/W/Ta relative to CoFeB/Ta is attributed to IOHE due to the emergence of long-range orbital current within the W-insertion layer.” According to this attribution, the THz generation efficiency is expected to be similarly large also for a bilayer CoFeB/W. Unfortunately, this bilayer system is not examined by the author. If this is verified, it can be strong supporting evidence for the attribution.

Authors' reply: We appreciate the concern of the referee. Indeed, it will be the strong supporting evidence for attributing the THz enhancement from CoFeB/W/Ta to IOHE. We have, therefore, carried out more experiments on CoFeB/W. To ensure consistency in our comparisons, we repeated the same experiments on the CoFeB/Ta and CoFeB/W/Ta samples, which were already included in the original manuscript.

Figure R6 presents the fresh results obtained on CoFeB/W bilayers, where, direct comparison between CoFeB/Ta(2) and CoFeB/W(2) as well as with the CoFeB/(W(2)/Ta(2)) can be seen.

Figure R6

From Fig. R6(a), we can note the following points: (i) THz signal emitted from CoFeB/W(2) is almost 3-times higher than that from CoFeB/Ta(2). Since the THz emission from these bilayers is exclusively due to ISHE, the similar polarity and different THz amplitudes are quite consistent with the sign and magnitude of the spin Hall angles in W and Ta. (ii) About 10 times enhanced THz signal from CoFeB/W(2)/Ta(2) in comparison with that from CoFeB/Ta(2) and nearly 4 times in comparison with that from CoFeB/W(2) are observed in Fig. R6(b). Such an enhancement cannot be possible with the ISHE picture drawn for the CoFeB/W/Ta also. If the THz emission from the CoFeB/W(2)/Ta(2) trilayer was solely due to the ISHE, we would expect its amplitude to be equal to the sum of the THz signal amplitudes emitted from the individual bilayer counterparts, i.e., CoFeB/Ta(2) and CoFeB/W(2). However, the total sum of the THz signals from the two bilayers is still approximately three times lesser than the THz signal amplitude from the CoFeB/W(2)/Ta(2) sample. We may note that the optimum thicknesses of the W and Ta layers we have used are close to the spin diffusion lengths in them. *{T.-C. Wang et al., Physical Review Materials, 2, 014403, (2018)}*

We reiterate that the nearly one order higher THz generation efficiency of the CoFeB/W/Ta system is due to the emergence of long-range orbital current within the W-insertion layer. In the W-layer of the heterostructure, the spin current gets converted to charge current due to the ISHE as well as into additionally large orbital current as per the spin-orbital correlation factor. Now, this orbital current is efficiently converted to charge current through IOHE in the Ta layer. Coherent addition of all the in-phase charge currents from ISHE in W layer and IOHE in the Ta layer results in much enhanced THz radiation from the trilayer.

In addition to above, finally, we would like to bring to the attention of the referee that another study on arXiv *{P. Wang et al., arXiv:2305.05830, (2023)}*, that has appeared a couple of days ago, also presents role of W-insertion layer in a Co/W/Ti and Co/W/Mn trilayer heterostructures. The introduction of a W layer has been found to significantly enhance (more than an order) the orbitronic terahertz emission. This enhancement is attributed to the generation of additional large orbital current within the W-insertion layer.

From the above discussion, we have added the following in our revised manuscript.

- We have added a line in para 1 of page 8: “While the THz amplitude for $t = 2$ nm is ~ 10 times stronger than that from $t = 0$ nm, the same is nearly 4 times higher than from the CoFeB(2)/W(2) bilayer (see Supplementary Section S14)..”
- Figure R6 and related discussion is added as a new section S14 in the supplementary information of our revised manuscript.

Comment#5: Line 150 reads “... are used to obtain the results presented in Fig. 2(c)”. However, Fig. 2 does not have subpanel (c). I suspect “Fig. 2(c)” should be “Fig. 2(b)”.

Authors' reply: Thank you for pointing out this discrepancy. We have corrected this error in the revised manuscript.

Reviewer #3

Remarks by the Reviewer: The authors present an experimental study of the THz emission from photoexcited ferromagnetic (FM)/non-magnetic metal (NM) heterostructures in which they seek to disentangle emission driven by the inverse orbital Hall effect (IOHE) and the inverse spin Hall effect (ISHE). Based on the temperature dependence of the THz emission amplitudes, the dominant emission process in NiFe/Nb is attributed to the inverse orbital Hall effect (IOHE), whereas THz emission from Fe/Ta and CoFeB/Pt bilayers is predominantly attributed to the inverse spin Hall effect (ISHE). These results are extracted from calculated values of the effective intrinsic Hall conductivities of each structure.

To my knowledge, the disentanglement of IOHE and ISHE mechanisms using temperature dependent THz time-domain spectroscopy has not previously been reported. The subject matter sits well within the scope of work published in Nature Communications. Many of the papers referenced in the manuscript have been published in journals within the Nature stable, including Nature Communications, Nature Photonics, Nature Electronics, Nature Materials, Nature Nanotechnology, and in other quality journals including Advanced Optical Materials, Physical Review B, Physical Review Letters, and Applied Physics Letters. As such, the work is likely to be of interest and relevance to readers of Nature Communications.

Comments on the analysis presented.

Authors' response: We thank the referee for appreciating and recommending our work. We are grateful for his/her time spent reviewing our manuscript and its assessment. To the specific comments by the referee, we have provided our detailed responses below.

Comment#1: Lines 99-101

The statement made that “.....hence the conversion to orbital current is pronounced” does not appear to be well supported by reference 52 cited by the authors. The referenced paper investigates angular momentum dissipation following femtosecond laser excitation of ferromagnetic Ni. This paper states:

"We also show that electron orbits do not act as a reservoir for angular momentum" and "In 3d transition metals, sum rules relate the integral L3 XMCD signal to a linear combination of spin, S, and orbital, L, angular momentum components along the magnetization direction as $S+3/2L$The temporal evolution of $S+3/2L$ in Fig. 3b represents the first quantitative demonstration that S is transferred to the lattice and not to L on a 100 fs timescale.....This excludes L as a reservoir for S."

Authors' reply: We are sorry for this slip while citing an article from the literature. We agree with the reviewer that the ref. 52 talks about the NIR femtosecond laser excitation induced spin angular momentum quenching and relaxation via spin-lattice relaxation channel. In the other paper {C. Stamm et al., *Physical Review B*, 81, 104425, (2010)}, soft x-ray femtosecond pulses induced transient enhancement in SOC was studied. We submit that we are unable to provide such

experimental evidence to the transient enhancement of SOC in our sample and, it is beyond the scope of our study.

We have revised the corresponding statement and citation to the previous work from literature in this regard. The revised text in para 3 of page 2 reads as follows:

“Therefore, an ultrafast optically induced orbital current sets in,^{48,52} which possess similar symmetry properties to the spin current but can exhibit relatively different transport dynamics.^{9,53} Furthermore, as the ultrafast excitation of spin and orbital magnetization has been reported^{50,54-56} to exhibit a similar evolution, the emergence of orbital current can be comprehended through the analogy with the already established spin current formation.⁵² ”

Comment#2: Temperature dependent transmission of quartz?

The THz emission data is collected in transmission mode. The samples are pumped on the substrate side so the 800nm excitation pulses pass through the quartz substrates before reaching the film. In analysing the temperature dependence of the emission, has the temperature dependence of the pump beam transmission through quartz been checked? Different excitation powers will clearly impact the THz emission amplitudes, so ruling this out as a potential factor would be a useful addition to the Supporting Information. Alternatively, the authors might consider recollecting this data in reflection mode.

Authors' reply: We thank the referee for this insightful comment. The effect of temperature on the transmission of an 800 nm femtosecond laser pulse through a quartz substrate can depend on several factors, such as the temperature range, laser intensity, pulse duration, and material properties, etc. At fixed and reasonable pump fluence values, the optical factors are minimized, however, the effects due to material and temperature need to be carefully investigated.

Figure R7

We have performed additional experiments to check the above and recorded the optical transmission through the quartz substrate with the varying sample temperature and other parameters fixed as in the original experiments. The corresponding result is shown in Fig. R7. There is hardly any change in the transmission, it is less than a percent while going from the lowest temperature to the room temperature. We measured a negligible (<2%) change in the THz emission efficiency from our sample corresponding to the variation in the excitation power mentioned above. Such an insignificant excitation power variation does not affect the large temperature dependence of the THz signal as measured by us in the manuscript.

Regarding the other point by the referee, we had in fact collected data initially in the reflection mode to compare the behavior with that in the transmission mode. We found a similar kind of behavioral change in the two cases. Due to consistent similarity in the data from the two, we continued all the experiments only in the transmission mode and by excitation the sample from the substrate side.

From the above discussion, we have added the following in our revised manuscript.

- We have added a line in para 1 of page 4: “The substrate side optical excitation geometry does not create any temperature dependent contributions (see Supplementary Section S15).”
- Figure R7 and related discussion is added as a new section S15 in the supplementary information of our revised manuscript.

Comment#3: Fig.2b and related analysis

The linear fit presented is clearly not a good fit to the data. The reliability of the extracted experimental value of $(\sigma_{\text{SOH}}^{\text{int}})_{\text{NM}}^{\text{eff}} = +281 \text{ } \Omega\text{-1cm-1}$ (line 175) is thus called into question. A strong and robust justification for the use of a linear fit here needs to be made by the authors. The equivalent theoretically calculated value of $+140 \text{ } \Omega\text{-1cm-1}$ (line 177) does not derive from the parameters stated in lines 175-176. The quoted figures would appear to suggest a value of $+170 \text{ } \Omega\text{-1cm-1}$. Also, given that the claimed experimental value is twice the size of the theoretical value, I am not sure I can agree with the statement (line 177) that these “compare quite well”.

Authors' reply: We thank the reviewer for pointing out an error in the calculated numerical value. We are sorry for the typo error (line 177 of the old manuscript), as the calculation suggests a number $+170 \text{ } \Omega\text{-1cm-1}$ instead of $+140 \text{ } \Omega\text{-1cm-1}$. We have corrected this in our revised manuscript. In regard to the point by the referee that the calculated value differs significantly from the

experimentally obtained value of $\sim +280 \text{ } \Omega^{-1} \text{ cm}^{-1}$, we have modified the corresponding text in our revised manuscript as, “Although the experimental value of $(\sigma_{SOH}^{int})_{NM}^{eff}$ is slightly higher than the calculated value, they can still be considered to be in good agreement,....” in para 1 of page 5.

Within the currently available theoretical framework {*Y. Tian et al., Physical Review Letters, 103, 087206, (2009)*} for the spin Hall resistivity relation with the longitudinal resistivity, which include contributions from both the intrinsic and extrinsic mechanisms, has to follow a linear scaling relation. {*M. Isasa et al., Physical Review B, 91, 024402, (2015)*; *E. Sagasta et al., Physical Review B, 94, 060412, (2016)*; *E. Sagasta et al., Physical Review B, 98, 060410, (2018)*; *H. Gamou et al., Physical Review B, 99, 184408, (2019)*} Therefore, Hence, we have adhered to this model and attempted to extract relevant parameters accordingly. Below, we would like to again justify the theoretical background to use of the existing linear scaling relation for the analysis our results.

The formalism of the spin Hall resistivity includes both the intrinsic and extrinsic mechanisms contributing to the spin-charge interconversion-based effects like AHE, SHE, etc., has to follow a linear scaling relation {*Y. Tian et al., Physical Review Letters, 103, 087206, (2009)*} due to the following reasons. First, according to Karplus and Luttinger, {*R. Karplus et al., Physical Review, 95, 1154-1160, (1954)*} AHE is related to SOC and it is responsible for any transverse velocity component in between scattering events. This mechanism is identified as the intrinsic contribution to AHE that relies on the band structure of the metal and describe by the Berry curvature. From the latter, the intrinsic anomalous Hall conductivity, σ_{AHE}^{int} , can be calculated {*T. Jungwirth et al., Physical Review Letters, 88, 207208, (2002)*} via the relation, $\rho_{AHE} = \sigma_{AHE}^{int} \cdot \rho_{xx}^2$ between the anomalous Hall resistivity, ρ_{AHE} , and longitudinal resistivity, ρ_{xx} . The extrinsic contributions, namely, skew-scattering and side-jump scattering from impurities, were proposed by Smit {*J. Smit, Physica, 24, 39-51, (1958)*} and Berger {*L. Berger, Physical Review B, 2, 4559-4566, (1970)*}, respectively. In both of these cases, a transverse displacement is generated during the scattering with impurities. Skew scattering contribution arises due to spin-dependent scattering caused by effective SOC imparted by the impurities in the lattice. The corresponding contribution to ρ_{AHE} is a linear function of the residual resistivity, $\rho_{0,xx}$, i.e., $\rho_{AHE} = \alpha_{SS} \cdot \rho_{0,xx}$, α_{SS} being the skew-scattering angle. Side-jump scattering results into the deflection of the electron velocity in opposite direction for the different spin states due to the opposite electric field they experience

when approaching an impurity. The corresponding contribution to ρ_{AHE} is given as $\rho_{AHE} = \sigma_{SJ} \cdot \rho_{0,xx}^2$, σ_{SJ} being the side-jump anomalous Hall conductivity. Finally, the overall behaviour of ρ_{AHE} must include both the above mentioned intrinsic and extrinsic contributions for AHE, which, according to Tian et al., {*Y. Tian et al., Physical Review Letters, 103, 087206, (2009)*} can be put together in the form of the following equation,

$$\rho_{AHE} = \sigma_{AHE}^{int} \cdot \rho_{xx}^2 + \sigma_{SJ} \cdot \rho_{0,xx}^2 + \alpha_{SS} \cdot \rho_{0,xx}$$

Phenomenologically, a similar linear fitting equation (Eq. 5 in the manuscript) is also employed for SHE and has been widely used {*M. Isasa et al., Physical Review B, 91, 024402, (2015)*; *E. Sagasta et al., Physical Review B, 94, 060412, (2016)*; *E. Sagasta et al., Physical Review B, 98, 060410, (2018)*; *H. Gamou et al., Physical Review B, 99, 184408, (2019)*} to determine the intrinsic and extrinsic contributions in nonmagnetic materials.

Comment#4: Fig.4e

Again a linear fit is clearly not appropriate here. The data seems to follow a curve of decreasing negative gradient. This calls into question the reliability of the extracted value of $(\sigma_{SOH}^{int})^{eff} = -3256$ (\hbar/e) Ω -1cm-1 (line 296).

Authors' reply: Thank you for the comment. From the response to the previous comment, we reiterate that the use of the linear scaling relation in analyzing our results is the most appropriate one within the currently existing theoretical framework for the quantification of weightage of intrinsic and extrinsic mechanisms. We agree that the linear fit to some of our data does not appear to be the best fit in Fig. 4(e), however, it succeeds in capturing an overall trend in the data. We hope that the referee will agree with the justification given in the response of the previous comment.

Comment#5: Comparison between emission from CoFeB/Ta and CoFeB/W/Ta

In the second half of the manuscript, further results are presented in which the authors demonstrate a ten-fold enhancement of the THz emission from CoFeB/Ta arising from the insertion of a W interlayer. It is claimed that the enhancement seen from the CoFeB/W/Ta heterostructure arises from the enhanced orbital transport in W. This section of the work raises questions which are not currently addressed by the manuscript. It is not clear how the authors can be confident that the W is acting as an interlayer and not as the key ISHE interface for the generation of THz. I would strongly suggest that a further comparison with a CoFeB/W bilayer is required in addition to the comparison between CoFeB/Ta and CoFeB/W/Ta presented in the current submission.

Authors' reply: Thank you for your valuable feedback. We appreciate your concern regarding the role of the W interlayer in the enhancement of THz emission from CoFeB/Ta heterostructures. We agree that a comparison between CoFeB/W, CoFeB/Ta bilayers, and CoFeB/W/Ta trilayer would be helpful in clarifying the role of the insertion layer. We have performed additional experiments on CoFeB/W bilayer system to address this concern.

Before getting to our detailed reply, we would like to bring to the attention of the referee that another study on arXiv{*P. Wang et al., arXiv:2305.05830, (2023)*}, that has appeared a couple of days ago, also presents role of W-insertion layer in a Co/W/Ti and Co/W/Mn trilayer heterostructures. The introduction of a W interlayer has been found to significantly enhance (more than an order) the orbitronic terahertz emission. This enhancement is attributed to the generation of additional large orbital current within the W-insertion layer.

Figure R8 presents our experimental results on CoFeB/W bilayer. The results on the CoFeB/Ta and CoFeB/W/Ta are already included in our original manuscript. From Fig. R8, following observations are made: (i) THz signal magnitude emitted from CoFeB/W(2nm) bilayer sample is ~ 2.5 times higher than that from CoFeB/Ta(2nm) whereas their polarities are same. (ii) THz signal from CoFeB/W bilayer decreases if the W layer thickness is changed from 2 to 3 nm. (iii) CoFeB/W(2)/Ta(2) sample emits ~ 10 and ~ 4 times higher THz signal as compared to that from CoFeB/Ta(2) and CoFeB/W(2) samples, respectively. These observations clearly suggest a different origin than ISHE for the THz emission here.

Figure R8

Based on the above observations, the following points are made.

Analogous to CoFeB/Ta(2nm) bilayers, ISHE is the origin of THz emission from CoFeB/W(2nm). The similar polarity and different THz amplitudes are quite consistent with the sign and magnitude of the spin Hall angles in W and Ta.

- (1) Analogous to CoFeB/Ta(2nm) bilayers, ISHE is the origin of THz emission from CoFeB/W(2nm). The similar polarity and different THz amplitudes are quite consistent with the sign and magnitude of the spin Hall angles in W and Ta.
- (2) If the THz emission in the CoFeB/W(2)/Ta(2) trilayer was solely due to ISHE, one would expect THz signal amplitude to be nearly equal to the sum of the THz signals emitted from the individual bilayer counterparts, namely CoFeB/Ta(2) and CoFeB/W(2). On the other hand, we measure THz signal from the CoFeB/W(2)/Ta(2) to be nearly three times stronger than the total sum of the signals from the two bilayers, individually. Therefore, we conclude that THz generation from the CoFeB/W(2)/Ta(2) is not solely due to ISHE.
- (3) In the case of CoFeB/W bilayer, THz signal strength decreases while increasing the W-layer thickness from 2 to 3 nm. This clearly indicates that the spin diffusion length in the W-layer, and hence it's optimum thickness for the THz emission via ISHE, is about 2-3 nm. Such a value of the spin diffusion length in W-layer matches well with the literature. {*T.-C. Wang et al., Physical Review Materials, 2, 014403, (2018)*}
- (4) Continuing with the point 2 above, the presence of such a thick W-insertion layer ($W = 2\text{nm}$) would have significantly reduced the spin current being transmitted to the adjacent Ta layer in the CoFeB/W/Ta trilayer structure. Therefore, complete spin to charge conversion within the W-layer or in parts within the W-layer and the Ta-layer of the heterostructure, one cannot explain the nearly 10 times [4 times] THz signal enhancement in CoFeB/W(2)/Ta(2) as compared with the CoFeB/Ta(2) [CoFeB/W(2)] from the ISHE as the source. Additionally, enhancement in the spin current transparency due to an insertion layer is only possible if a very small thickness of it, just a few Angstroms, {*Q. Lu et al., Physical Review Applied, 12, 064035, (2019); Y. Li et al., ACS Nano, 16, 11852-11861, (2022)*} is used. For thick (a few nm) insertion layers, like the case in our experiments, the spin transparency reduces significantly. {*J. Hawecker et al., Advanced Optical Materials, 9, 2100412, (2021)*}

From the above points, we therefore conclude that THz enhancement from CoFeB/W(2)/Ta(2) as compared to the CoFeB/W(2) and CoFeB/Ta(2) is not due to either an

increased spin Hall angle or the spin transparency as required for the ISHE to play the dominant role in the THz emission. However, due to the presence of W-interlayer in CoFeB/W(2)/Ta(2) trilayer heterostructure, the spin current gets converted to charge current due to the ISHE as well as into an additionally large orbital current as per the spin-orbital correlation in W. Now, this orbital current is efficiently converted to charge current through IOHE in the Ta layer. Coherent addition of all the in-phase charge currents from ISHE in W layer and IOHE in the Ta layer results in much enhanced THz radiation from the trilayer. Hence, the role of the W-insertion layer as an orbital transport channel rather than simply as an ISHE interface must be considered as demonstrated in the very recent arXiv {*P. Wang et al., arXiv:2305.05830, (2023)*} report as well. This understanding elucidates the observations presented in our manuscript.

We believe that the additional data will further reinforce and solidify the conclusions drawn in our study. From the above clarifications and discussion, we have added the following in our revised manuscript.

- We have added a line in para 1 of page 8: “While the THz amplitude for $t = 2$ nm is ~ 10 times stronger than that from $t = 0$ nm, the same is nearly 4 times higher than from the CoFeB(2)/W(2) bilayer (see Supplementary Section S14)..”
- Figure R8 and related discussion is added as a new section S14 in the supplementary information of our revised manuscript.

Referencing and Optional/minor work recommended to authors:

Referencing:

1. A large number of non-peer reviewed arXiv works are currently cited by the authors, particularly relating to the injection of orbital currents into neighboring heavy metal layers. While this might be expected for recently emerging research directions such as this, it would be good to see these references converted to peer reviewed journal articles as this manuscript progresses towards publication.

Authors' response:

Thank you for your comment. Many of the cited arXiv works have been published in different journals. Accordingly, we have updated them in our revised manuscript.

Other referencing queries:

2. Line 193. *Is the cited reference 8 correct here? I am unable to locate a reference to a negligible spin-orbit correlation factor for CoFeB and Fe in the cited paper.*

Authors' response: Figure 2(b) of the cited paper [8] compares the spin-orbit correlation factor values calculated from first principle results on various FM materials, including, Fe, CoFe, Co, and Ni. In another paper, {*Y. Xu et al., arXiv:2208.01866, (2022)*} an explicit statement has been made, i.e., "...CoFeB cannot generate a noticeable orbital current...", further confirming the fact that spin-orbit correlation factor in CoFeB is negligible. We have now added this additional reference in our revised manuscript as ref. [48].

3. Line 269. *Is the cited reference 39 the most appropriate here? The cited paper did not consider pure W, as used here, as an interlayer material.*

Authors' response: It is true that reference [39] did not explicitly consider pure W as the interlayer material, rather an alloy of W, i.e., Au_{0.85}W_{0.15}. The study did investigate the effects of different materials with the conclusion that the interface passivation would give to a common trend, irrespective of the choice in the material selection. Specifically, they have concluded that "The investigation of Co/X(d)/Pt trilayer with X = Ti (weak SOC material), Au and Au_{0.85}W_{0.15} (high SOC materials) shows that in all cases, an interlayer reduces the spin-to-charge conversion". W is also a high spin-orbit coupling materials. Therefore, we believe that reference 39 is still an appropriate citation in that context.

Other general comments:

4. *The technical content is generally well set out and easy to follow, however a number of abbreviations appear in the manuscript without definition and there appears to be inconsistency between the symbols used in the main text and in Table 1:*

Authors' response: We thank the referee for bringing these anomalies to our attention. We have now taken care of them appropriately in our revised manuscript.

5. Line 91: "*<L.S>*". *Neither L nor S are defined. The reviewer has taken these to be orbital angular momentum (L) and spin angular momentum (S) but this needs to be made explicit within the document.*

Authors' response: Thank you for your comment. We apologize for any confusion caused by the lack of clarity regarding the symbols L and S. Indeed, these symbols represent, the orbital angular momentum and the spin angular momentum, respectively.

We have now defined them in para 1 of page 1 in our revised manuscript.

6. Line 102: “OHC and SHC”. These abbreviations are first introduced here but not defined. These may stand for orbital Hall conductivity (OHC) and spin Hall conductivity (SHC), however these same terms are given different symbols in Table 1, where σ_{SH} is used to represent spin Hall conductivity and σ_{OH} is used to represent orbital Hall conductivity. It is suggested that the authors might consider additions and amendments here to achieve greater clarity and consistency.

Authors’ response: We apologize for the inconsistency in the use of abbreviations and would like to thank referee for bringing this to our notice. We have corrected this issue by considering σ_{SH} to represent spin Hall conductivity and σ_{OH} to represent orbital Hall conductivity, everywhere in our revised manuscript.

7. Experimental methods are clearly described, with an adequate level of detail that would enable the work to be reproduced by the reader. The experimental work is based on standard, well-tested and robust techniques. The figures generally support the content well, with appropriate use of Supplementary Material to provide useful additional images and data.

Authors’ response: We thank the reviewer for appreciating and providing positive feedback on the experimental methods and figures presented in the manuscript.

8. An annotated copy of the manuscript is attached. In addition to the comments above, small typing errors and queries regarding sentence construction are highlighted within.

Authors’ response: We greatly appreciate the help by the referee in improving our manuscript. We have gone through all the inputs by the referee in the annotated copy and have tried out best to correct them as appropriately as possible in the revised manuscript.

Reviewers' Comments:

Reviewer #1:

Remarks to the Author:

In the revised manuscript, the authors addressed the issues about the AHE contribution and the temperature rise due to femtosecond laser excitation in the cryostat. They provide additional experimental results, such as the terahertz emission from single FM layers and the estimation of temperature rise upon femtosecond laser excitation. Those revisions show that the AHE contribution and the temperature rise in the cryostat do not play an essential role during the terahertz emission.

Eqn 5 was extended from a similar equation that has been used to interpret the terahertz emission from the inverse spin Hall effect. I raised the issue about the validity of Eqn 5 in the previous report because one assumed the orbital diffusion length is the same as the spin diffusion length in the derivation of Eqn 5. In the revised manuscript, the authors tried to show the validity of Eqn 5 by claiming the orbital and spin diffusion length were 25 nm for Nb. I disagree with those arguments because of the following:

1. The extraction of the orbital diffusion length of Nb in Figure S12 was not clear. The decay above 25 nm is likely due to the decrease in the impedance of the thin film. The orbital diffusion length seems to be much higher than 25 nm. The problem of the orbital diffusion length of Nb would be interesting for the community, but a systematic investigation is lacking.

2. The authors circumvented my concern and reformulated my question into a different problem about the orbital diffusion length of Nb. Even if the spin diffusion length and the orbital diffusion length were the same for Nb, such equality could not be generalized to other materials. Therefore, simple extension from inverse spin Hall effect to the case of orbital current like Eqn 5 is risky. The validity of Eqn 5 is highly questionable.

In the response letter, the author referred to Wang et al., arXiv:2305.05830 for the role of the W insertion layer. In fact, this paper by Wang et al. has been published in Npj Quantum Materials (Wang et al., Npj Quantum Materials (2023)). It was brought to my attention that the manuscript under review shows much resemblance to the paper of Wang et al. Wang et al. also reported the terahertz emission from a bilayer and trilayers with an insertion layer of W. Moreover, Wang et al. also showed that the inserted W layer significantly enhances the terahertz emission. The main message conveyed by the manuscript, especially for trilayers with an insertion layer of W, are more or less similar. In other words, the manuscript under review would not bring much new knowledge to the community.

In conclusion, I believe the quality of this manuscript did not meet the high criteria of Nature Communications. The manuscript would be more suitable for another journal specializing in materials.

Reviewer #2:

Remarks to the Author:

The authors carried out additional experiments, which support the validity of their interpretation of the inverse orbital Hall effect. In particular, their new data indicates that the spin and orbital relaxation lengths are similar in Nb, which makes their analysis [Eq. (5)] more reasonable. I am satisfied with the revised manuscript, and I recommend the publication of the manuscript in Nature Communications after the following revision; Figure 4(a) shows that the inverse orbital Hall effect and the inverse spin Hall effect generate the charge current in opposite directions. But according to Eq. (6), both the inverse orbital Hall effect and the inverse spin Hall effect should generate the charge current along the same direction.

Reviewer #3:

Remarks to the Author:

Nature Comms paper: Kumar and Kumar, “Ultrafast THz probing of nonlocal orbital current in transverse multilayer metallic heterostructures”

Review of updated manuscript

The additional work carried out since the initial review has satisfactorily addressed many of the concerns raised. In particular, the addition results on the temperature dependence of the pump transmission through the quartz substrate and the added comparison between CoFeB/W, CoFeB/Ta and CoFeB/W/Ta are useful additions to the work and provide much stronger evidence for the claims within.

All issues raised regarding referencing, abbreviations etc. have been adequately addressed by the authors.

My only remaining concern is the linear fit to the graphs in Figs 2(b) and 4(e) and the parameters subsequently extracted from these fit lines. While I am grateful for the authors' detailed justification for a theoretical linear relationship between the plotted variables, the fact remains that the experimental data does not appear to be linear. I would suggest a more cautious approach in the discussion of these results. Fig 2(b) for example, may show some linearity over limited regions. The fit line and the gradient of $281 \Omega^{-1}\text{cm}^{-1}$ currently quoted in the paper would seem somewhat arbitrary. There is a degree of linearity above $\rho_{\text{Nb}}^2 = 0.5 \times 10^3 (\mu\Omega.\text{cm})^2$ with gradient approximately $190 \Omega^{-1}\text{cm}^{-1}$ (which incidentally is then in reasonable agreement to the theoretical quoted value of $170 \Omega^{-1}\text{cm}^{-1}$); conversely, a linear fit to the data from $\rho_{\text{Nb}}^2 = 0.450 - 0.475 \times 10^3 (\mu\Omega.\text{cm})^2$ gives a gradient exceeding $1500 \Omega^{-1}\text{cm}^{-1}$ (which is clearly in very poor agreement with the theoretical value presented) . The change here is interesting but at present unexplained.

The revised wording in the manuscript states that “the experimental value is slightly higher than the calculated value”. While there is an order of magnitude agreement, I would suggest the removal of the word “slightly” as an increase of over 65% cannot be accurately described as slight.

Similar comments apply to Fig.4(e). Again I would urge greater caution in analysis of the data here. For example in line 305 the statement “Clearly, the data fits for a negative slope.....” might be better expressed as “the data shows a degree of linearity”, since the gradient from $\rho_{\text{W/Ta}}^2 = 2.80-2.95 \times 10^3 (\mu\Omega.\text{cm})^2$ would appear somewhat different to the gradient beyond $\rho_{\text{W/Ta}}^2 = 3.0 \times 10^3 (\mu\Omega.\text{cm})^2$. Again this deviation from the expected theoretical behaviour is interesting but currently unexplained.

Replies to referee comments (Manuscript ID: NCOMMS-23-03062A)

The comments from the reviewers are reproduced verbatim as below (black color). We have tried to answer all of them in our best possible manner (the responses are given in blue color). As per the comments by reviewer 3, we have made a few minor revisions arising from the two-segmented linear fit of the data in Figs. 2b and 4e. We are thankful for the insightful comments from all the referees.

The revisions in the main manuscript and the supplementary information are highlighted in red color.

Reviewer #1

In the revised manuscript, the authors addressed the issues about the AHE contribution and the temperature rise due to femtosecond laser excitation in the cryostat. They provide additional experimental results, such as the terahertz emission from single FM layers and the estimation of temperature rise upon femtosecond laser excitation. Those revisions show that the AHE contribution and the temperature rise in the cryostat do not play an essential role during the terahertz emission. Eqn 5 was extended from a similar equation that has been used to interpret the terahertz emission from the inverse spin Hall effect. I raised the issue about the validity of Eqn 5 in the previous report because one assumed the orbital diffusion length is the same as the spin diffusion length in the derivation of Eqn 5. In the revised manuscript, the authors tried to show the validity of Eqn 5 by claiming the orbital and spin diffusion length were 25 nm for Nb. I disagree with those arguments because of the following:

1. The extraction of the orbital diffusion length of Nb in Figure S12 was not clear. The decay above 25 nm is likely due to the decrease in the impedance of the thin film. The orbital diffusion length seems to be much higher than 25 nm. The problem of the orbital diffusion length of Nb would be interesting for the community, but a systematic investigation is lacking.
2. The authors circumvented my concern and reformulated my question into a different problem about the orbital diffusion length of Nb. Even if the spin diffusion length and the orbital diffusion length were the same for Nb, such equality could not be generalized to other materials. Therefore, simple extension from inverse spin Hall effect to the case of orbital current like Eqn 5 is risky. The validity of Eqn 5 is highly questionable.

In the response letter, the author referred to Wang et al., arXiv:2305.05830 for the role of the W insertion layer. In fact, this paper by Wang et al. has been published in Npj Quantum Materials (Wang et al., Npj Quantum Materials (2023)). It was brought to my attention that the manuscript under review shows much resemblance to the paper of Wang et al. Wang et al. also reported the terahertz emission from a bilayer and trilayers with an insertion layer of W. Moreover, Wang et al. also showed that the inserted W layer significantly enhances the terahertz emission. The main message conveyed by the manuscript, especially for trilayers with an insertion layer of W, are more or less similar. In other words, the manuscript under review would not bring much new knowledge to the community. In conclusion, I believe the quality of this manuscript did not meet the high criteria of Nature Communications. The manuscript would be more suitable for another journal specializing in materials.

Authors' response: The reviewer's concern is about the use of Eq. (5) and its extension to the orbital Hall effect. Here, we have clarified referee's two explicit points and justified the use of

Eq.5 and related formalism in our analysis. We respectfully disagree with the reviewer's assessment that our manuscript does not meet the criteria of Nat. Communications. The reviewer made this judgment as she/he feels that our results on W-inserted trilayer sample resemble with the recently published paper on arXiv which, we had brought to his/her notice earlier. We would like to highlight that our original submission in January 2023 predates the publication date of the arXiv/npj paper, and therefore, any similarity is coincidental. We have provided a detailed explanation to this fact at the end of our below responses, i.e., after the response to the reviewers' two explicit points.

(1) In the previous revision, the estimation of orbital diffusion length in Nb was provided by normalizing the THz signal amplitude with the optical absorbance. The inclusion of the THz impedance does not affect much our results in Fig. S12 of the supplementary information. For completeness, we have added this information in the revised supplementary section S13 (highlighted in red color). Details of the same are provided below for the reviewer's convenience.

Part (a) in the above figure shows the recorded THz signals transmitted through different NiFe/Nb samples of varying Nb thickness, including the reference signals, by using a dual color air plasma-based THz radiation source. The THz pulses are incident on the samples from the substrate side. The THz impedance was calculated using the standard procedure^{1,2} for each bilayer structure. Further, these values of the THz impedance were used to normalize the THz emission signals from all the NiFe/Nb samples. In part (b) of the above figure, peak-to-peak (E_{pp}) value of the THz signal is normalized by the THz impedance and the results are plotted (blue data points, R2) as a function of the Nb thickness. It is seen that the normalized data behavior (R2) does not differ much from the earlier one (R1) when THz impedance was not considered. Therefore, the estimated value of the orbital diffusion length in Nb can be safely considered to be ~25 nm as mentioned in our manuscript.

(2) Regarding the extension of spin Hall-related effects to the orbital case via Eq. 5 in our manuscript, we would like to highlight a few facts here. In general, the THz field is directly related to the material parameters via the relation³, $E(\omega)_{\text{THz}} = Ze \int_0^d dz. \theta_{\text{SH}} \cdot J_S$. The ultrafast optically injected spin current, J_S , has a spin diffusion length, λ_{SD} , which gives an effective spin current, J_S^* , smaller than J_S in the nonmagnetic layer. Likewise, the ultrafast optically injected orbital current, J_L , has an orbital diffusion length, λ_{OD} , which gives an effective orbital current, J_L^* , smaller than J_L . It should be noted that J_L and J_S share a common dynamic origin,^{1, 4-6} but their transport properties are specific to the materials and heterostructures involved, which are reflected in their respective diffusion lengths. A generalized equation³ that relates the effective ultrafast current to the diffusion length is expressed as $\frac{J^*}{J} = \frac{\lambda}{\delta} \tanh\left(\frac{\delta}{2\lambda}\right)$, where δ represents the thickness of the nonmagnetic layer. The value of λ for either spin or orbital current varies depending on the type

of materials but the ratio $\frac{\partial \lambda(J^*/J)}{(J^*/J)}$ is approximately a constant. For optimal THz emission from a heterostructure, δ is kept close to λ . In the equality situation when $\lambda_{OD} = \lambda_{SD}$, the expression yields a value of ~ 0.02 (or 2%) for $\lambda_{OD} = \lambda_{SD} = \delta = 3\text{nm}$. For the other situation with an inequality, i.e., $\lambda_{OD} \neq \lambda_{SD}$, either $\lambda_{OD} > \lambda_{SD}$ or $\lambda_{OD} < \lambda_{SD}$ is possible. Mostly, we have used material thicknesses $\delta < \lambda$. A larger value of λ , for example, $\lambda = 8\text{ nm}$ and $\delta = 3\text{ nm}$, the above expression yields a value of approximately 0.5% relative to previous case (i.e., $\lambda_{OD} = \lambda_{SD}$). It implies that the variations in λ can be neglected in Eq. 5, thereby, justifying its use for orbital currents even if λ value has a significant variation for the spin and orbital cases. Another fact that strongly supports extension of spin Hall-related effects to the OHE case, is the origin of this effect itself. Similar to the spin-related effects like AHE⁷⁻¹⁰ and SHE¹¹, the OHE is also expected¹⁵ to have intrinsic and extrinsic contributions. It is indeed reflected in the outcomes for our OHE systems, i.e., NiFe/Nb, where we have distinguished intrinsically and extrinsically dominated resistivity regions analogous to the effects like, AHE⁷⁻¹⁰ and SHE¹¹ in other systems. Therefore, to the best of our knowledge and current understanding of the phenomena, the OHE formalism as given in our paper is appropriate.

The last point by the reviewer is on the suitability of our paper for Nature Communications due to similarities between our results on CoFeB/W/Ta heterostructures with those in a recent arXiv paper (arXiv:2305.05830, now published in npj quantum materials), where THz enhancement due to the W-insertion layer is much the same, though different FM and NM material layers are used in the later. During the revision of our paper, we brought this study to the notice of the referee. The similarity between the two is just coincidental. In fact, we were the first to report such results and hence the subsequent study in the above arXiv paper only strengthens further our original arguments that orbital transport due to W-insertion layer enhances the THz emission. The reviewer seems to have ignored these facts to which we politely disagree. Please note that we submitted our original manuscript on 20th Jan, 2023 for which we received request for first revision on 29th March, 2023. To this, we submitted our revised version on 22nd May, 2023. Meanwhile, the arXiv paper first appeared on 10th May, 2023. Hence, we included this particular article in our responses and also brought to notice of the referee. Our manuscript uniquely brings out several new details and understanding as summarized here again for the reference of the reviewer. **(a)** The paper by Wang et al., (arXiv:2305.05830) investigates IOHE in Ti and Mn materials via THz emission studies performed at the room temperature ONLY. The insertion of a W-layer enhances THz emission due to an orbital to charge current conversion. The fact in our paper that W-insertion layer enhances the orbital transport and hence the THz emission from the respective multilayer heterostructures, is only strengthened further by the observations in that paper. Please note that different sample configurations, i.e., Co/W/Ti and Co/W/Mn, were used in their study. In our case, we used CoFeB/W/Ta, for this particular fact. **(b)** The unique and the notable aspect, which distinguishes our work from the previous studies, is our experimental approach of investigating the emerging orbital-based phenomena. We conducted comprehensive and complex low-temperature dependent THz emission measurements to determine the IOHE and ISHE phenomena systematically in various bilayer heterostructures, where either IOHE or ISHE is the main contributor. The insights gained from temperature dependent measurements were then utilized to upscale the orbital transport induced THz emission. W-insertion layer assisted pronounced THz emission via orbitronic effect was demonstrated in CoFeB/W/Ta and it is analysed in the whole temperature range. Moreover, the determination of orbital diffusion length in Nb adds significant value to our study. **(c)** The low-temperature experiments also help us, for the first time, to reveal

the signature of extrinsic contribution to OHE. By employing two-segmented linear fitting across the entire range of resistivity data, in case of the NiFe/Nb and the CoFeB/W/Ta heterostructures, we are able to distinguish dominating extrinsic and intrinsic contributions to OHE in different resistivity regions. Thus obtained parameters compare even better with the theoretically calculated values.

Reviewer #2

The authors carried out additional experiments, which support the validity of their interpretation of the inverse orbital Hall effect. In particular, their new data indicates that the spin and orbital relaxation lengths are similar in Nb, which makes their analysis [Eq. (5)] more reasonable. I am satisfied with the revised manuscript, and I recommend the publication of the manuscript in Nature Communications after the following revision; Figure 4(a) shows that the inverse orbital Hall effect and the inverse spin Hall effect generate the charge current in opposite directions. But according to Eq. (6), both the inverse orbital Hall effect and the inverse spin Hall effect should generate the charge current along the same direction.

Authors' response: We thank the reviewer for appreciating and recommending our work for publication in Nature communication. We are grateful for his/her time spent reviewing our manuscript and its assessment.

We are sorry for the error in the depiction of the orbitals' flow in the schematic of Figure 4(a), though the arrows representing the charge currents due to ISHE (J_{CS}) and IOHE (J_{CL}) were correctly shown as per the sign conventions of Eq. 6. In the revised manuscript, we have appropriately rectified this oversight. We greatly appreciate the reviewer's careful reading of our manuscript.

Reviewer #3

The additional work carried out since the initial review has satisfactorily addressed many of the concerns raised. In particular, the addition results on the temperature dependence of the pump transmission through the quartz substrate and the added comparison between CoFeB/W, CoFeB/Ta and CoFeB/W/Ta are useful additions to the work and provide much stronger evidence for the claims within. All issues raised regarding referencing, abbreviations etc. have been adequately addressed by the authors.

My only remaining concern is the linear fit to the graphs in Figs 2(b) and 4(e) and the parameters subsequently extracted from these fit lines. While I am grateful for the authors' detailed justification for a theoretical linear relationship between the plotted variables, the fact remains that the experimental data does not appear to be linear. I would suggest a more cautious approach in the discussion of these results. Fig 2(b) for example, may show some linearity over limited regions. The fit line and the gradient of $281 \Omega^{-1} \text{ cm}^{-1}$ currently quoted in the paper would seem somewhat arbitrary. There is a degree of linearity above $\rho_{\text{Nb}2} = 0.5 \times 10^3 (\mu\Omega.\text{cm})^2$ with gradient approximately $190 \Omega^{-1} \text{ cm}^{-1}$ (which incidentally is then in reasonable agreement to the theoretical quoted value of $170 \Omega^{-1} \text{ cm}^{-1}$); conversely, a linear fit to the data from $\rho_{\text{Nb}2} = 0.450 - 0.475 \times 10^3 (\mu\Omega.\text{cm})^2$ gives a gradient exceeding $1500 \Omega^{-1} \text{ cm}^{-1}$ (which is clearly in very poor agreement with the theoretical value presented). The change here is interesting but at present unexplained.

The revised wording in the manuscript states that “the experimental value is slightly higher than the calculated value”. While there is an order of magnitude agreement, I would suggest the removal of the word “slightly” as an increase of over 65% cannot be accurately described as slight.

Similar comments apply to Fig.4(e). Again I would urge greater caution in analysis of the data here. For example in line 305 the statement “Clearly, the data fits for a negative slope.....” might be better expressed as “the data shows a degree of linearity”, since the gradient from $\rho_{W/Ta} = 2.80-2.95 \times 10^3 (\mu\Omega.cm)^2$ would appear somewhat different to the gradient beyond $\rho_{W/Ta} = 3.0 \times 10^3 (\mu\Omega.cm)^2$. Again this deviation from the expected theoretical behaviour is interesting but currently unexplained.

Authors' response: We are thankful and greatly appreciate the reviewer’s constructive feedback on our work.

Regarding the comment by the reviewer: My only remaining concern is the linear fit to the graphs The change here is interesting but at present unexplained.

As per the suggestion by the reviewer, we have relooked at the linear fitting of the data in Figs. 2(b) for NiFe/Nb and 4(e) for CoFeB/W/Ta. First, in Fig. 2(b), a linear fit to the data above $\rho_{Nb}^2 = 0.5 \times 10^3 (\mu\Omega.cm)^2$ indeed provided value of the gradient to be, $(\sigma_{SOH}^{int})^{eff.} = \sim + 195 \Omega^{-1}cm^{-1}$, which is remarkably close to the theoretically calculated value of $\sim + 170 \Omega^{-1}cm^{-1}$. Therefore, we agree that a segmented linear fit to the data would be more appropriate here. The linear fittings are redone in Figs. 2(b) and 4(e), and the same are reproduced below for the convenience of the reviewer.

An excellent agreement between the experimental value and the one calculated from theoretical values of intrinsic parameters suggests that the particular resistivity regime, i.e. $\rho_{Nb}^2 > 0.5 \times 10^3 (\mu\Omega.cm)^2$ is dominated by the intrinsic contribution to OHE. On the other hand, at lower values of $\rho_{Nb}^2 < 0.5 \times 10^3 (\mu\Omega.cm)^2$, linear fit to the data in Fig. 2(b) produces a large slope value of $\sim + 1210 \Omega^{-1}cm^{-1}$, which cannot be justified by considering only the intrinsic contribution to OHE. This signifies the selective dominance of intrinsic and extrinsic contributions to OHE within the studied resistivity range. Similar phenomenon was reported earlier for AHE⁷⁻¹⁰ and SHE.¹¹ in certain other systems, where, in the low resistivity region, referred to as the clean metal region, the extrinsic mechanism is found to be the dominant factor, whereas in the high resistivity region or the dirty and moderately dirty region, the intrinsic effect is the main contributor. Consequently, different linear scaling relations are followed in each region. Thus, as predicted in a recent report¹⁵ that the resistivity dependent analysis could be helpful in revealing the extrinsic contribution to

the OHE, our study provides experimental evidence of extrinsic and intrinsic contributions to the OHE, which has not been reported hitherto.

Regarding the comment by the reviewer: The revised wording in the manuscript states that “the experimental increase of over 65% cannot be accurately described as slight.

In view of the corrections made in the above comment, this point is now taken care of. We have now used two-segment linear fit across the entire resistivity range.

Regarding the comment by the reviewer: Similar comments apply to Fig.4(e).theoretical behaviour is interesting but currently unexplained.

In concurrence with the above discussion, here in Fig. 4 (e) also, we have opted a two-segment linear fitting approach to analyze the experimental data. By doing so, the degree of linearity is much improved in comparison to the single linear fitting in the whole range. In the low and high resistivity regions across $\rho_{W/Ta}^2 = 2.9 \times 10^3 (\mu\Omega.cm)^2$, we obtain very good fits though the disparity with the single linear curve fitting in the entire range is not as pronounced as in the previous case (Figure 2(b)). After incorporating the above fitting procedure, we have revised the corresponding text in our manuscript. Through temperature dependent IOHE induced THz emission studies, we are able to capture the intrinsically and extrinsically dominated resistivity regions by employing the two-segment linear fitting approach in NiFe/Nb and CoFeB/W/Ta systems.

In light of the above discussion, we have made a few changes in Fig. 2(b) and Fig. 4(e); and revised related text (highlighted in red color) in para 2 of page 4 and in para 3 of page 8, respectively, in our revised manuscript.

References

1. Seifert, T. S.; Go, D.; Hayashi, H.; *et al.*, 2023; arXiv:2301.00747.
2. Wang, P.; Feng, Z.; Yang, Y.; *et al.* *npj Quantum Materials* **2023**, 8, (1), 28.
3. Seifert, T.; Jaiswal, S.; Martens, U.; *et al.* *Nature Photonics* **2016**, 10, 483.
4. Boeglin, C.; Beaupaire, E.; *et al.* *Nature* **2010**, 465, (7297), 458-461.
5. Stamm, C.; Pontius, N.; Kachel, T.; Wietstruk, M.; Dürr, H. A. *Physical Review B* **2010**, 81, (10), 104425.
6. Hennecke, M.; Radu, I.; Abrudan, R.; *et al.* *Physical Review Letters* **2019**, 122, (15), 157202.
7. Sangiao, S.; Morellon, L.; Simon, G.; *et al.* *Physical Review B* **2009**, 79, (1), 014431.
8. Onoda, S.; Sugimoto, N.; Nagaosa, N. *Physical Review B* **2008**, 77, (16), 165103.
9. Onoda, S.; Sugimoto, N.; Nagaosa, N. *Physical Review Letters* **2006**, 97, (12), 126602.
10. Tian, Y.; Ye, L.; Jin, X. *Physical Review Letters* **2009**, 103, (8), 087206.
11. Sagasta, E.; Omori, Y.; Isasa, M.; *et al.* *Physical Review B* **2016**, 94, (6), 060412.
12. Go, D.; Jo, D.; Lee, H.-W.; Kläui, M.; Mokrousov, Y. *Europhysics Letters* **2021**, 135, (3), 37001.
13. Kontani, H.; Tanaka, T.; Hirashima, D. S.; Yamada, K.; Inoue, J. *Physical Review Letters* **2009**, 102, (1), 016601.
14. Tanaka, T.; Kontani, H.; *et al.* *Physical Review B* **2008**, 77, (16), 165117.
15. Sala, G.; Gambardella, P. *Physical Review Research* **2022**, 4, (3), 033037.
16. McHugh, O. L. W.; Goh, W. F.; Gradhand, M.; Stewart, D. A. *Physical Review Materials* **2020**, 4, (9), 094404.
17. Kumar, A.; Bansal, R.; Chaudhary, S.; Muduli, P. K. *Physical Review B* **2018**, 98, (10), 104403.

Reviewers' Comments:

Reviewer #1:

Remarks to the Author:

The authors addressed my question about the orbital diffusion length of Nb in their reply. They generated terahertz emission from a dual-color air plasma terahertz source and recorded the terahertz transmission through NiFe/Nb samples. From the comparison of terahertz generation from NiFe/Nb samples and terahertz transmission through NiFe/Nb samples (Part b), I think a characteristic Nb thickness of ~ 25 nm is reasonable.

The authors proposed that the generalized equation³ that relates the effective ultrafast current to the diffusion length of orbit or spin. Such an equation has been derived previously for spin transport by considering the exponential decay of spin currents and the reflection of spin currents at the boundary. This equation, originally derived for spin currents, was extended directly to the case of orbital currents by the authors, which is not scientifically sound. Actually, this issue has been addressed by Sala et al. in Phys. Rev. Research 4, 033037 (2022). Sala et al. showed in Eqn. (6-9) that the interconversion between spin and orbital needs to be considered to describe properly of the orbital transport.

Moreover, judging from Eqn. 5 and the equation in the response, it appears to me that the authors believed that the spin currents and orbital currents are independent of each other. However, such an oversimplified picture leads to an inconsistency between the results of the W-insertion layer and their explanation. Both this manuscript and the paper by Wang et al. (Wang et al., NPJ Quantum Materials (2023)) highlighted the enhancement of THz emission due to the W insertion layer. The role of the W layer cannot be understood following the author's assumption that spin currents and orbit currents belong to two independent channels. The role of W needs to be related to the interconversion between spin and orbit currents, and it could be interpreted more naturally in the framework proposed by Sala et al.

More generally, I disagree with the assertion of authors that " J_L and J_S share a common dynamic origin." The authors mentioned that AHE, OHE and SHE shared similar origins, which is well-accepted in the community. Just as pointed out by Go et al in his paper Phys. Rev. Lett. 130, 246701 (2023), the orbital angular momentum and the spin angular momentum are fundamentally different objects. The two effects do not possess the same transport properties. The message raised by the authors is controversial and even misleading for the community.

To summarize, I feel that the authors did not improve much the quality of their manuscript compared to the previous version. I believe the quality of this manuscript did not meet the high criteria of Nature Communications.

Reviewer #3:

Remarks to the Author:

In the latest revision of the manuscript, the authors have satisfactorily addressed the issues raised relating to the line fitting on Figures 2(b) and 4(c).

I am satisfied with the revised manuscript, and have no further material objections to publication of this work in Nature Communications.

Replies to Referee Comments

(Manuscript ID: NCOMMS-23-03062B)

We had received further remarks/comments on our revised manuscript from Reviewers 1 and 3. The remarks and comments are reproduced verbatim in the following, where we have also replied point by point to them.

Reviewer #3

Remarks to the Author:

In the latest revision of the manuscript, the authors have satisfactorily addressed the issues raised relating to the line fitting on Figures 2(b) and 4(c).

I am satisfied with the revised manuscript and have no further material objections to publication of this work in Nature Communications.

Authors' response:

We are thankful to the Reviewer for recommending our revised manuscript for publication in Nature Communications.

Reviewer #1

Remarks to the Author:

The authors addressed my question about the orbital diffusion length of Nb in their reply. They generated terahertz emission from a dual-color air plasma terahertz source and recorded the terahertz transmission through NiFe/Nb samples. From the comparison of terahertz generation from NiFe/Nb samples and terahertz transmission through NiFe/Nb samples (Part b), I think a characteristic Nb thickness of ~ 25 nm is reasonable.

The authors proposed that the generalized equation³ that relates the effective ultrafast current to the diffusion length of orbit or spin. Such an equation has been derived previously for spin transport by considering the exponential decay of spin currents and the reflection of spin currents at the boundary. This equation, originally derived for spin currents, was extended directly to the case of orbital currents by the authors, which is not scientifically sound. Actually, this issue has been addressed by Sala et al. in Phys. Rev. Research 4, 033037 (2022). Sala et al. showed in Eqn. (6-9) that the interconversion between spin and orbital needs to be considered to describe properly of the orbital transport.

Moreover, judging from Eqn. 5 and the equation in the response, it appears to me that the authors believed that the spin currents and orbital currents are independent of each other. However, such an oversimplified picture leads to an inconsistency between the results of the W-insertion layer and their explanation. Both this manuscript and the paper by Wang et al. (Wang et al., NPJ Quantum Materials (2023)) highlighted the enhancement of THz emission due to the W insertion layer. The role of the W layer cannot be understood following the author's assumption that spin currents and orbit currents belong to two independent channels. The role of W needs to be related to the interconversion between spin and orbit currents, and it could be interpreted more naturally in the framework proposed by Sala et al.

More generally, I disagree with the assertion of authors that “ J_L and J_S share a common dynamic origin.” The authors mentioned that AHE, OHE and SHE shared similar origins, which is well-accepted in the community. Just as pointed out by Go et al in his paper Phys. Rev. Lett. 130, 246701 (2023), the orbital angular momentum and the spin angular momentum are fundamentally different objects. The two effects do not possess the same transport properties. The message raised by the authors is controversial and even misleading for the community.

To summarize, I feel that the authors did not improve much the quality of their manuscript compared to the previous version. I believe the quality of this manuscript did not meet the high criteria of Nature Communications.

Authors’ response:

We are thankful that the Reviewer is satisfied with our extended experiments on the NiFe/Nb samples for demonstrating the characteristic length of Nb. From other specific details given by the referee in the comments, we are now able to see the main contention of his/her. We submit that for all experimental results in our manuscript, role of spin-orbit conversion is indeed included through the correlation factor. With regards to the three points raised by the referee above, we have tried to address them in the following in our best possible manner. Please note that all the revisions which were required to enhance clarity and our presentation are indicated in red color in the revised manuscript and the Supporting Information document.

Regarding the comment by the referee, “The authors proposed that the generalized equation³ that relates the effective ultrafast current to the diffusion length of orbit or spin. Such an equation has been derived previously for spin transport by considering the exponential decay of spin currents and the reflection of spin currents at the boundary. This equation, originally derived for spin currents, was extended directly to the case of orbital currents by the authors, which is not scientifically sound. Actually, this issue has been addressed by Sala et al. in Phys. Rev. Research 4, 033037 (2022). Sala et al. showed in Eqn. (6-9) that the interconversion between spin and orbital needs to be considered to describe properly of the orbital transport.”

Sala et al. (Phys. Rev. Research 4, 033037 (2022)) noted that the orbital and spin currents can compete or assist each other in determining the net spin-orbit torques acting on the magnetic layer via the OHE and SHE. Calling it a single or two-channel current phenomenon can still be debated but ultimately there is one charge current which produced both the orbital and the spin currents. Sala et al. also noted that in light and heavy metals, OHE is more efficient than SHE. Due to Onsager reciprocity, same has to be true for the inverse effects, i.e., IOHE is intrinsically more efficient for orbit-charge conversion than the ISHE for spin-charge conversion. However, appropriate conditions are necessary for their direct observations. In our case, femtosecond laser excitation perturbs the magnetization in the FM layer which induces an instantaneous spin current. Subsequently, spin-orbit conversion due to the correlation factor η_{L-S} leads to production of an ultrafast orbital current. While the interconversion is ON between them, both the spin and orbital currents diffuse through the thickness of the material (limited by the respective characteristic diffusion length and transport properties) before they produce charge current in tandem in the NM layer via ISHE and IOHE, respectively. The net transient charge current is thus written as Eq. (1) in the manuscript for the case of NiFe/Nb, for example,

$$J_C = \theta_{SH} \cdot J_S + \theta_{OH} \cdot J_L = \theta_{SH}^{Nb} \cdot J_S + \theta_{OH}^{Nb} \cdot \eta_{L-S}^{NiFe} \cdot J_S \quad (1)$$

To satisfy the reviewer’s concern, we have gone through the entire discussion related to Eq. 3 to Eq. 5 of our manuscript again. Please also see the revised text in Section S10 of the

Supplementary information, some of which we have rewritten for better clarity. There, up to Eq. S5, the discussion is for spin current and its conversion to charge current via ISHE to generate E_{THz} . Now to account for the spin-orbit conversion in the FM layer while both the spin and the orbital currents are transported across the FM layer thickness and produce charge current in the NM layer, an effective spin-orbit diffusion length (Sala et al., Phys. Rev. Research 4, 033037 (2022)), λ_{LS} has been used. We are sorry for missing out this point earlier where λ_S was used instead, in the relation between effective spin-orbital Hall resistivity and the THz signal magnitude. Please see particularly, the modifications made in the text from Eq. (3) to just below Eq. (5) on pages 4 and 5 of the revised manuscript.

The modifications in the Supplementary Information are reproduced below for the convenience of the referee. The modifications in the main manuscript are also indicated in red color.

For the case of THz emission by IOHE and ISHE, where both the spin current to charge current and orbital current to charge current conversions take place simultaneously, the above relation needs to be modified. IOHE becomes relevant if spin-orbit conversion takes place in the material layer via the spin-orbit correlation factor, η_{L-S} . Hence, for this case, the above relation gets modified to the following one,

$$E_{THz} = \frac{eJ_S\lambda_{LS}}{(d/\rho_{FM/NM})} \left(\frac{\rho_{SH}^{NM}}{\rho_{xx}^{NM}} + \frac{\rho_{OH}^{NM}}{\rho_{xx}^{NM}} \cdot \eta_{L-S} \right) \quad (S6)$$

Notice that, J_S is retained in the above relation because it represents the maximum instantaneous spin current produced in the FM layer by the ultrafast optical excitation, however, the spin relaxation length has been substituted by the effective spin-orbit diffusion length,¹⁵ $\lambda_{LS} = \sqrt{\lambda_L\lambda_S}$ (λ_L and λ_S being the orbital and spin diffusion lengths) and θ_{SH} has been substituted by an effective spin-orbit Hall angle¹⁶, $\theta_{SOH}^{eff} = \left(\frac{\rho_{SH}^{NM}}{\rho_{xx}^{NM}} + \frac{\rho_{OH}^{NM}}{\rho_{xx}^{NM}} \cdot \eta_{L-S} \right)$ to take into account the orbital Hall resistivity, ρ_{OH}^{NM} of the NM layer. The θ_{SOH}^{eff} effectively takes care of the overall effect of spin and orbital currents, and their interconversion on the ultimate charge current and hence THz radiation produced.^{15, 16} Equation (S6) is strictly valid for unit interfacial transparency,^{16, 17} which can be rearranged to obtain a relation between the effective spin-orbit Hall resistivity, ρ_{SOH}^{eff} and the measured THz signal as

$$\rho_{SOH}^{eff} = (\rho_{SH}^{NM} + \rho_{OH}^{NM} \cdot \eta_{L-S}) = E_{THz} \left(\frac{\rho_{xx}^{NM}}{\rho_{FM/NM}} \right) \left(\frac{d}{\lambda_{LS}} \right) \frac{1}{eJ_S} \quad (S7)$$

Now, at each sample temperature T , the value of the experimentally measured THz signal $E_{THz}(T)$ in V/cm from Eq. (S1) can be used to calculate the effective spin-orbit Hall resistivity $\rho_{SOH}^{eff}(T)$ in the units of $\mu\Omega \cdot cm$ using Eq. (S7) for the complete temperature-dependent analysis.

Regarding the comment by the referee, “Moreover, judging from Eqn. 5 and the equation in the response, it appears to me that the authors believed that the spin currents and orbital currents are independent of each other. However, such an oversimplified picture leads to an inconsistency between the results of the W-insertion layer and their explanation. Both this manuscript and the paper by Wang et al. (Wang et al., NPJ Quantum Materials (2023)) highlighted the enhancement of THz emission due to the W insertion layer. The role of the W layer cannot be understood following the author's assumption that spin currents and orbit currents belong to two independent channels. The role of W needs to be related to the interconversion between spin and orbit currents, and it could be interpreted more naturally in the framework proposed by Sala et al.”

The equation mentioned in our response earlier is same as Eq. (S5) of the Supplementary Information and it was used as Eq. (5) in the main manuscript. You will notice that it has been corrected now. From the beginning in our manuscript, spin-orbit conversion was considered via the spin-orbit correlation factor η_{L-S} . Following the ultrafast optical excitation of the FM layer, an instantaneous spin current J_S is produced, which through η_{L-S} converts to orbital current partially and this interconversion between the two continues during their transport in the layer due to which

the effective spin-orbit diffusion length $\lambda_{LS} = \sqrt{\lambda_L \lambda_S}$ shall be used. Since spin and orbit currents are transported across the thickness of the FM layer in the case of NiFe/Nb sample and the CoFeB and W-insertion layer in the case of CoFeB/W/Ta, an effective spin-orbit diffusion length $\lambda_{LS} = \sqrt{\lambda_L \lambda_S}$ has been used in the relationship between the effective spin-orbit Hall resistivity and the THz field magnitude for these samples. We have added a new section, S16 in the revised Supplementary information document to elaborate on the enhancement in the orbital current and hence THz emission from the CoFeB/W/Ta trilayer due to the W-insertion layer in it. Thicker the W-insertion layer, more is the enhancement in the orbital current and hence in the THz emission from the trilayer. This is reproduced below for the convenience of the referee.

As shown in Fig. 4 (c) of our manuscript, the THz signal from CoFeB/W(t)/Ta trilayer heterostructure increases with the increasing thickness (t) of the W-insertion layer. This observation is consistent with the fact that heavy metal W possesses large negative valued spin-orbit correlation factor owing to which efficient spin-orbit conversion occurs in it. Long diffusion length for the orbital current and high orbital Hall conductivity in the adjacent Ta layer profuse in tandem to generate stronger THz signal via IOHE from CoFeB/W/Ta trilayer and efficiency increases with the varying thickness of W-insertion layer. Following the assertions of spin-orbit interconversions by Sala et al.,¹⁵ and the associated coupled differential equations for the chemical potentials and current densities related to the spin and orbital degrees, the thickness dependent enhancement in the OHE or the ultrafast IOHE in our case can be analyzed. Figure S15 presents the response of the orbital current as a function of the W-insertion layer calculated using the phenomenological model of Sala et al.¹⁵ for the orbital current given by the relation,

$$J_L(z_{NM}) = - \left(\frac{\sigma_S \mp \frac{\sigma_L}{\lambda_{LS}^2 \gamma_2}}{1 - \frac{\gamma_1}{\gamma_2}} \right) \frac{E}{2} \text{Sech}^2 \left(\frac{z_{NM}}{2\lambda_1} \right) - \left(\frac{\sigma_S \mp \frac{\sigma_L}{\lambda_{LS}^2 \gamma_1}}{1 - \frac{\gamma_1}{\gamma_2}} \right) \frac{E}{2} \text{Sech}^2 \left(\frac{z_{NM}}{2\lambda_2} \right) + \sigma_L E \quad (\text{S8})$$

Here, different parameters have meaning as given in the original paper,¹⁵ E is the applied external field in typical OHE settings, z_{NM} is the thickness of the heavy metal layer and so on. In generating the qualitative result of Fig. S15, we have used the fact that $\sigma_S < 0$ and $\sigma_L > 0$ for the heavy metal W.^{31, 32} Clearly, a larger thickness of the W-insertion layer supports larger spin-orbital conversion. Hence, stronger orbital current is injected into the adjacent Ta layer which possesses much stronger orbital Hall conductivity than the spin Hall conductivity, thereby, resulting into much stronger orbit-charge conversion via IOHE and hence stronger THz emission from the trilayer with thicker W-insertion layer. We believe that our extensive temperature-dependent experiments for the THz emission via IOHE in CoFeB/W/Ta have much scope for further theoretical exploration and more experiments studies on such systems in future.

Figure S15. Increase in the orbital current with the increasing thickness of W-insertion layer in CoFeB/W/Ta trilayer.

Regarding the comment by the referee, “More generally, I disagree with the assertion of authors that “ J_L and J_S share a common dynamic origin.” The authors mentioned that AHE, OHE and SHE shared similar origins, which is well-accepted in the community. Just as pointed out by Go et al in his paper Phys. Rev. Lett. 130, 246701 (2023), the orbital angular momentum and the spin angular momentum are fundamentally different objects. The two effects do not possess the same transport

properties. The message raised by the authors is controversial and even misleading for the community.”

We are sorry for the confusion. We feel the referee got it wrong from only part of the sentence in our previous response. The complete statement in our response earlier (previous revision) was, “It should be noted that J_L and J_S share a common dynamic origin,^{1,4,6,7} but their transport properties are specific to the materials and heterostructures involved, which are reflected in their respective diffusion lengths.”

From the “common dynamic origin”, we meant to say that the source for the generation of both the J_L and J_S initially is the same ultrafast laser pulse. Of course, their transport properties are specific to the materials and heterostructures involved, which are reflected in their respective diffusion lengths. Such an assertion is fundamentally valid and consistent with the notes from previous work such as by Go et al., Phys. Rev. Lett. 130, 246701 (2023) and others.^{1,4,6,7}

The paper by Sala et al., Phys. Rev. Research 4, 033037 (2022) was already included in our manuscript. In addition, we have appropriately cited two new references in our revised manuscript:

1. Wang, P. et al. Inverse orbital Hall effect and orbitronic terahertz emission observed in the materials with weak spin-orbit coupling. npj Quantum Materials 8, 28, (2023).
2. Go, D. et al. Long-Range Orbital Torque by Momentum-Space Hotspots. Physical Review Letters 130, 246701, (2023).

References

1. Seifert, T. S.; Go, D.; Hayashi, H.; Rouzegar, R.; Freimuth, F.; Ando, K.; Mokrousov, Y.; Kampfrath, T. *Nature Nanotechnology* **2023**.
2. Xu, Y.; Zhang, F.; Fert, A.; Jaffres, H.-Y.; Liu, Y.; Xu, R.; Jiang, Y.; Cheng, H.; Zhao, W. J. a. e.-p., Orbitronics: Light-induced Orbit Currents in Terahertz Emission Experiments. 2023; p arXiv:2307.03490.
3. Go, D.; Freimuth, F.; Hanke, J.-P.; Xue, F.; Gomonay, O.; Lee, K.-J.; Blügel, S.; Haney, P. M.; Lee, H.-W.; Mokrousov, Y. *Physical Review Research* **2020**, 2, (3), 033401.
4. Stamm, C.; Pontius, N.; Kachel, T.; Wietstruk, M.; Dürr, H. A. *Physical Review B* **2010**, 81, (10), 104425.
5. Sala, G.; Gambardella, P. *Physical Review Research* **2022**, 4, (3), 033037.
6. Boeglin, C.; Beaupaire, E.; Halté, V.; López-Flores, V.; Stamm, C.; Pontius, N.; Dürr, H. A.; Bigot, J. Y. *Nature* **2010**, 465, (7297), 458-461.
7. Hennecke, M.; Radu, I.; Abrudan, R.; Kachel, T.; Holldack, K.; Mitzner, R.; Tsukamoto, A.; Eisebitt, S. *Physical Review Letters* **2019**, 122, (15), 157202.

Reviewers' Comments:

Reviewer #1:

Remarks to the Author:

In the latest revision of the manuscript, the authors have addressed my concerns. I have no further objections to publication of this work in Nature Communications.

Replies to Referee Comments

(Manuscript ID: NCOMMS-23-03062C)

Reviewer #1

Remarks to the Author:

In the latest revision of the manuscript, the authors have addressed my concerns. I have no further objections to publication of this work in Nature Communications.

Authors' response:

We are thankful to the Reviewer for recommending our revised manuscript for publication in Nature Communications.